# PROVABLY EFFICIENT POLICY OPTIMIZATION WITH RARE POLICY SWITCHES

## ABSTRACT

While policy optimization algorithms have demonstrated remarkable empirical success in reinforcement learning (RL) tasks, their theoretical analysis is limited compared to those of value-based algorithms. In this paper, we address the gap by proposing a new provably efficient policy optimization algorithm that incorporates optimistic value estimation and rare policy switches. For linear Markov decision processes (MDPs), our algorithm achieves a regret bound of $\tilde{O}(d^2 H^2 \sqrt{T})$, which is the sharpest regret bound of a policy optimization algorithm for linear MDPs. Furthermore, we extend our algorithm to general function approximation and establish a regret bound of $\tilde{O}(\sqrt{T})$. To our best knowledge, this is the first regret guarantee of a policy optimization algorithm with general function approximation. Numerical experiments demonstrate that our algorithm has competitive regret performances compared to the existing RL algorithms while also being computationally efficient, supporting our theoretical claims.

## 1 INTRODUCTION

Policy optimization algorithms have garnered substantial attention across diverse RL applications, from board/video games to large language models (Silver et al., 2016; Schulman et al., 2015; 2017; Mnih et al., 2016; Fujimoto et al., 2018; Ouyang et al., 2022). However, despite its wide applicability, in contrast to their value-based counterparts (Jiang et al., 2017; Agrawal & Jia, 2017; Jin et al., 2018), the theoretical understandings of policy optimization methods remain relatively under-explored. (Kakade & Langford, 2002; Bagnell et al., 2003; Bhandari & Russo, 2019)

This evident dichotomy between value-based and policy optimization methods becomes increasingly pronounced concerning linear function approximation and the associated regret guarantees. For the value-based methods, the literature on linear function approximation has seen rapid expansion, with the linear Markov decision processes (MDP) often being the central focus (Jiang et al., 2017; Du et al., 2019; Wang et al., 2019; Sun et al., 2019; Yang & Wang, 2019; Zanette et al., 2020b; Modi et al., 2020; Jin et al., 2020; Cai et al., 2020; Jia et al., 2020; Ayoub et al., 2020; Zanette et al., 2020a; Ishfaq et al., 2021; Zhou et al., 2021; Zhou & Gu, 2022; He et al., 2023). Many of these value-based methods have been shown to achieve $O(\sqrt{T})$ regret upper bounds, matching the $\Omega(\sqrt{T})$ lower bound for linear MDPs established in Zhou et al. (2021) in terms of the total time-step $T$. On the other hand, there have been very few existing works on policy optimization algorithms with $O(\sqrt{T})$ regret under the linear MDP. Hence, the following research question arises:

> **Q**: *Can we design a provably efficient and practical policy optimization algorithm with $O(\sqrt{T})$ regret bound for linear MDPs?*

We answer this question affirmatively by proposing a policy optimization algorithm, *Optimistic Policy Optimization with Rare Switches* (OPORS). For our proposed algorithm, we establish $\tilde{O}(d^2 H^2 \sqrt{T})$ regret bound under linear MDPs. To our best knowledge, Zhong & Zhang (2023) and Sherman et al. (2023) are the only existing results for policy optimization with linear MDPs. Zhong & Zhang (2023) show a $\tilde{O}(T^{3/4})$ regret bound which is known to be sup-optimal. A concurrent work by Sherman et al. (2023) proves a regret of $\tilde{O}(d^2 H^{5/2} \sqrt{T \log |\mathcal{A}|})$. Thus, the regret of our proposed method shows an improvement by a factor of $\sqrt{H \log |\mathcal{A}|}$. Hence, our algorithm has the sharpest regret bound known for policy optimization algorithms under linear MDPs. Such an

improvement is made possible by a new policy update rule inspired by the proximal policy optimization (PPO) (Schulman et al., 2017). Furthermore, the implementation of the algorithm in Sherman et al. (2023) is much more complicated than that of our method. Hence, our method is one of very few policy optimization methods for linear MDPs that achieve both provably efficiency and practicality. In addition, we extend our algorithm and regret analysis to general function approximation, which, to our best knowledge, is the first regret guarantees of a policy optimization algorithm for general function approximation. Our main contributions are summarized as follows:

- We propose a policy optimization algorithm OPORS for linear MDPs, which incorporates our novel policy update rule and a rare policy switching technique of $O(\log T)$ described in Section 3.1 We establish $\tilde{O}(d^2 H^2 \sqrt{T})$ regret bound (in Theorem 3.1) for our proposed algorithm, where $d$ is the dimension of linear features, $H$ is the episode length, and $T$ is the total time-steps. To our best knowledge, Theorem 3.1 establishes the sharpest known regret bound proven for linear MDPs, improving the regret bound of a concurrent work (Sherman et al., 2023). Along with Sherman et al. (2023), our result shows the first policy-based algorithm matching the $O(\sqrt{T})$ of the regret lower bound (Zhou et al., 2021).

- We extend our proposed algorithm to general function approximation and achieve $\tilde{O}(d_g^3 H^2 \sqrt{T})$ regret bound where $d_g$ represents the complexity of the function class measured by covering number or the eluder dimension (Russo & Van Roy, 2013). To our best knowledge, this is the first policy optimization algorithm with regret guarantees for general function approximation.

- We numerically evaluate our algorithm and show that it shows the state-of-the-art performances compared with the existing provably efficient RL methods, with far reduced computational cost.

## 2 PRELIMINARIES

**Notations.** We denote $[n] := \{1, 2, \ldots, n\}$ for $n \in \mathbb{N}$. For $x, y \in \mathbb{R}^d$, $\langle x, y \rangle$ denotes the inner product of $x$ and $y$. For a positive definite matrix $\boldsymbol{A}$, we denote $\|x\|_{\boldsymbol{A}} := \sqrt{x^T \boldsymbol{A} x}$. The notation $f \lesssim g$ means that there exists some constant $C > 0$ satisfying $f \le Cg$. For any function $f$ and numbers $a < b$, we denote $[f]_{[a,b]} := \max\{\min\{f, b\}, a\}$. For a function class $\mathcal{F}$, $\mathcal{C}(\mathcal{F}, \varepsilon)$ and $\mathcal{N}(\mathcal{F}, \varepsilon)$ denote the $\varepsilon$-cover and $\varepsilon$-covering number of $\mathcal{F}$.

**Problem Formulation.** We consider episodic MDPs $(\mathcal{S}, \mathcal{A}, H, \{P_h\}_{h \in [H]}, \{r_h\}_{h \in [H]})$, where $\mathcal{S}$ and $\mathcal{A}$ are the state space and the action space, $H$ is the length of each episode, $\mathbb{P} = \{\mathbb{P}_h\}_{h \in [H]}$ is the transition probability distributions, $r = \{r_h\}_{h \in [H]}$ is the reward functions. We assume that each episode starts at some initial state $s_1$ [1], and the episode ends after $H$ steps. For each step $h \in [H]$, the agent observes state $s_h$, then takes an action $a_h$. The agent receives a reward $r_h(s_h, a_h)$ determined by $(s_h, a_h)$, and transit to the next state $s_{h+1}$ according to the transition probability $\mathbb{P}_h(\cdot \mid s_h, a_h)$.

The agent takes actions based on its policy $\pi = \{\pi_h\}_{h \in [H]}$, where $\pi_h(\cdot \mid s)$ is a probability distribution over $\mathcal{A}$. The value function ($V$-function) and the action-value function ($Q$-function) of policy $\pi$ are the expected sum of rewards up to termination, starting from $s_h = s$ and $(s_h, a_h) = (s, a)$ respectively, following the policy $\pi$. Formally, they are defined as

$$V_h^\pi(s) := \mathbb{E}_\pi \left[ \sum_{h'=h}^H r_h(s_{h'}, a_{h'}) \mid s_h = s \right], \ Q_h^\pi := \mathbb{E}_\pi \left[ \sum_{h'=h}^H r_h(s_{h'}, a_{h'}) \mid s_h = s, a_h = a \right].$$

To simplify the notation, we use $P_h V_{h+1}$ to denote $\mathbb{E}_{s' \sim \mathbb{P}(\cdot|s,a)}[V_{h+1}(s')]$. We denote the optimal value function and the optimal action-value function by $V_h^*(s) := \max_\pi V_h^\pi(s)$ and $Q_h^*(s, a) := \max_\pi Q_h^\pi(s, a)$. There exists an optimal policy $\pi^*$ such that $V_h^{\pi^*}(s) = V_h^*(s)$ and $Q_h^{\pi^*}(s, a) = Q_h^*(s, a)$ for all $h \in [H], (s, a) \in \mathcal{S} \times \mathcal{A}$. For any $\pi$, the Bellman equation relates $Q^\pi$ to $V^\pi$ as

$$Q_h^\pi(s, a) = (r + \mathbb{P}V_{h+1}^\pi)(s, a), \ V_h^\pi(s) = \langle Q_h^\pi(s, \cdot), \pi(\cdot \mid s) \rangle, \ V_{H+1}^\pi(s) = 0.$$

When $\pi^k$ is the policy executed in the $k$-th episodes, the suboptimality of $\pi^k$ compared to the optimal policy $\pi^*$ is represented by $V_1^*(s_1^k) - V_1^{\pi^k}(s_1^k)$, where $s_1^k$ is the initial state in the $k$-th

---

[1] Our result is valid for the general case with an initial distribution $\rho(\cdot)$. We can modify the MDP by setting a fixed initial state $s_1$ and $\mathbb{P}_1(\cdot \mid s_1, a) = \rho(\cdot)$ for all $a \in \mathcal{A}$.

---

**Algorithm 1** Optimistic Policy Optimization with Rare Switches (OPORS)

---

1: **Input:** Failure probability $\delta \in (0,1)$, stepsize $\alpha > 0$, confidence radius $\beta > 0$, regularization parameter $\lambda > 0$, policy switching parameter $\eta > 1$
2: **Initialize:** Set $\{\pi_h^0(\cdot \mid \cdot)\}_{h \in [H]}$ as uniform policy on $\mathcal{A}$, $\Lambda_h^0 \leftarrow \lambda I$ for $\forall h \in [H]$, $\bar{k} = 0$
3: **for** episode $k = 1, \cdots, K$ **do**
4:     $\Lambda_h^k \leftarrow \sum_{i=1}^{k-1} \phi(s_h^i, a_h^i) \phi(s_h^i, a_h^i)^T + \lambda I$
5:     **if** $k = 1$ or $\exists h \in [H]$ $\det\left(\Lambda_h^k\right) \geq \eta \det\left(\Lambda_h^{\bar{k}}\right)$ **then**
6:         **for** step $h = H, \ldots, 1$ **do**
7:             $\hat{w}_h^k \leftarrow (\Lambda_h^k)^{-1} \sum_{i=1}^{k-1} \phi(s_h^i, a_h^i) \left[r_h(s_h^i, a_h^i) + V_{h+1}^k(s_{h+1}^i)\right]$
8:             $b_h^k(s,a) \leftarrow \beta \left\|\phi(s,a)\right\|_{(\Lambda_h^k)^{-1}}$
9:             $Q_h^k(s,a) \leftarrow [\phi(s,a)^T \hat{w}_h^k + b_h^k(s,a)]_{[0,H-h+1]}$
10:           Update $\pi_h^k(a \mid s) \propto \pi_h^{k-1}(a \mid s) \exp(\alpha Q_h^k(s,a))$
11:           $V_h^k(s) \leftarrow \langle Q_h^k(s,\cdot), \pi_h^k(\cdot \mid s)\rangle$
12:         **end for**
13:         $\bar{k} \leftarrow k$
14:     **else**
15:         $Q_h^k(s,a) \leftarrow Q_h^{k-1}(s,a), \pi_h^k(a \mid s) \leftarrow \pi_h^{k-1}(a \mid s), V_h^k(s) \leftarrow V_h^{k-1}(s)$ for all $h \in [H]$
16:     **end if**
17:     **for** step $h = 1, \ldots, H$ **do**
18:         Take an action $a_h^k \sim \pi_h^k(\cdot \mid s_h^k)$ and observe $s_{h+1}^k$
19:     **end for**
20: **end for**

---

episode. Summing them up for $k \in [K]$, we get the cumulative regret of $\pi$ over $K$ episodes:

$$\text{Regret}(K) := \sum_{k=1}^{K} V_1^*(s_1^k) - V_1^{\pi^k}(s_1^k).$$

The goal of the agent is to maximize the sum of rewards over $K$ episodes, or equivalently, to minimize the cumulative regret over $K$ episodes.

## 3   LINEAR FUNCTION APPROXIMATION

In this section, we present our proposed algorithm, named *Optimistic Policy Optimization with Rare Switches* (OPORS), a policy optimization algorithm with linear function approximation. The algorithm is based on the standard linear MDP (Yang & Wang, 2019; Jin et al., 2020) as defined below.

**Assumption 1** (Linear MDP). *An MDP $(\mathcal{S}, \mathcal{A}, H, \{P_h\}_{h \in [H]}, \{r_h\}_{h \in [H]})$ is a linear MDP with a feature map $\phi : \mathcal{S} \times \mathcal{A} \to \mathbb{R}^d$ if for all $h \in [H]$, there exists $d$ unknown (signed) measures $\boldsymbol{\mu}_h = (\mu_h^{(1)}, \ldots, \mu_h^{(d)})$ over $\mathcal{S}$ and an unknown vector $\boldsymbol{\theta}_h \in \mathbb{R}^d$, such that for all $(s,a) \in \mathcal{S} \times \mathcal{A}$,*

$$P_h(\cdot \mid s,a) = \langle \phi(s,a), \boldsymbol{\mu}_h\rangle, \quad r_h(s,a) = \langle \phi(s,a), \boldsymbol{\theta}_h\rangle.$$

*We assume $\|\phi(s,a)\|_2 \leq 1$ for all $(s,a) \in \mathcal{S} \times \mathcal{A}$, and $\max\{\|\boldsymbol{\mu}_h(\mathcal{S})\|_2, \|\boldsymbol{\theta}_h\|_2\} \leq \sqrt{d}$ for all $h \in [H]$.*

### 3.1   ALGORITHM: OPORS

In this section, we illustrate each component of our algorithm OPORS in Algorithm 1.

**Rare Policy Switching.** Infrequent policy switching (update) is a pivotal factor in enhancing both the regret bound and computational efficiency. OPORS accomplishes this by adhering to a rare-switching strategy, wherein policy updates are triggered when the determinant of the Gram matrix $\Lambda_h^k$ increases by a factor of $\eta$ since the last update (Line 5 of Algorithm 1). This rare-switching technique was introduced by Wang et al. (2021) for value-based methods and has been demonstrated to be computationally efficient. We can guarantee that when $\eta$ remains an absolute constant, the number of policy switches enforced by OPORS is upper-bounded by $O(dH \log T)$.

**Value Function Estimation.** Lines 6-12 of Algorithm 1 involve estimating the value function of the current policy. Starting with $V_{H+1}^k = 0$, for $h = H, \ldots, 1$, we compute the $Q$-values by minimizing the empirical squared Bellman error. The estimator is given by

$$\hat{w}_h^k = (\Lambda_h^k)^{-1} \sum_{i=1}^{k-1} \phi(s_h^i, a_h^i) \left[ r_h(s_h^i, a_h^i) + V_{h+1}^k(s_{h+1}^i) \right],$$

where $\Lambda_h^k = \sum_{i=1}^{k-1} \phi(s_h^i, a_h^i) \phi(s_h^i, a_h^i)^T + \lambda I$. Subsequently, we construct the optimistic $Q$-value $Q_h^k(s, a) = [\phi(s, a)^T \hat{w}_h^k + b_h^k(s, a)]_{[0, H-h+1]}$, where the bonus function is defined as $b_h^k(s, a) := \beta \|\phi(s, a)\|_{(\Lambda_h^k)^{-1}}$. Based on the optimistic $Q$-value $Q_h^k$ and the policy $\pi_h^k$, the optimistic $V$-value is determined as $\langle Q_h^k(s, \cdot), \pi_h^k(\cdot \mid s) \rangle$.

**Policy Update.** Our policy update rule is inspired by the proximal policy optimization (PPO) (Schulman et al., 2017). It is important to note that OPORS update the policy $\pi_h^k$ based on the previous policy $\pi_h^{k-1}$ and the *current* $Q$-value $Q_h^k$ (Line 10 of Algorithm 1). Consequently, the policy update and value function estimation steps alternate. In essence, each update in Line 10 follows a stepwise PPO approach, optimizing the policy at step $h$ in the order of $h = H, \ldots, 1$. In the $k$-th episode, during the step $h$ of the inner-loop (Lines 6-12), we have $\{Q_{h'}^k\}_{h' \geq h}$ and $\{\pi_{h'}^k\}_{h' \geq h+1}$. Consider an auxiliary policy $\tilde{\pi}^{k,h} := \{\pi_{h'}^{k-1}\}_{h' \leq h} \cup \{\pi_{h'}^k\}_{h' \geq h+1}$. Then, the PPO objective is formulated as:

$$\text{maximize}_\pi \left\{ L_h^{k-1}(\pi) - \alpha^{-1} \cdot \mathbb{E}_{\tilde{\pi}^{k,h}} \left[ \sum_{h'=1}^H D_{KL} \left( \pi_{h'}(\cdot \mid s_{h'}) \| \tilde{\pi}_{h'}^{k,h}(\cdot \mid s_{h'}) \right) \right] \right\}. \quad (1)$$

Here, $L_h^{k-1}$ is the local approximator for the expected return under $\tilde{\pi}^{k,h}$, which is defined as

$$L_h^{k-1}(\pi) := V_1^{\tilde{\pi}^{k,h}}(s_1) + \mathbb{E}_{\tilde{\pi}^{k,h}} \left[ \sum_{h'=1}^H \langle Q_{h'}^{\tilde{\pi}_h^k}(s_{h'}, \cdot), \pi_{h'}(\cdot \mid s_{h'}) - \tilde{\pi}_{h'}^{k,h}(\cdot \mid s_{h'}) \rangle \right].$$

Rearranging the optimization objective (1), we have an equivalent objective:

$$\text{maximize}_\pi \left\{ \mathbb{E}_{\tilde{\pi}^{k,h}} \left[ \sum_{h'=1}^H \left( \langle Q_{h'}^{\tilde{\pi}_h^k}(s_{h'}, \cdot), \pi_{h'}(\cdot \mid s_{h'}) \rangle - \alpha^{-1} D_{KL} \left( \pi_{h'}(\cdot \mid s_{h'}) \| \tilde{\pi}_{h'}^{k,h}(\cdot \mid s_{h'}) \right) \right) \right] \right\}.$$

A simple calculation leads to the closed-form solution:

$$\pi_{h'}(\cdot \mid s) \propto \tilde{\pi}_{h'}^{k,h}(\cdot \mid s) \cdot \exp(\alpha Q_{h'}^{\tilde{\pi}_h^k}(s, \cdot)).$$

Instead of optimizing $\{\pi_{h'}\}_{h' \in [H]}$ for all steps, we optimize $\pi_h$ for current step and set the optimized policy as $\pi_h^k$. Since true $Q_h^{\tilde{\pi}_h^k}$ is unknown, we substitute it with the estimated $Q$-value $Q_h^k$. Considering the definition of the auxiliary policy $\tilde{\pi}^{k,h}$, our update rule is expressed as:

$$\pi_h^k(\cdot \mid s) \propto \pi_h^{k-1}(\cdot \mid s) \cdot \exp(\alpha Q_h^k(s, \cdot)).$$

Our update rule distinguishes itself from the previous provably efficient policy optimization algorithms (Agarwal et al., 2020a; Cai et al., 2020; Zanette et al., 2021; Feng et al., 2021; Li et al., 2023; Liu et al., 2023; Zhong & Zhang, 2023; Li et al., 2023; Sherman et al., 2023), which employ the update rule $\pi_h^k(\cdot \mid s) \propto \pi_h^{k-1}(\cdot \mid s) \cdot \exp(\alpha Q_h^{k-1}(s, \cdot))$. As shown in Section 3.3, the new policy update rule enables us to incorporate $O(\log T)$ rare-switching techniques into our algorithm, which plays a crucial role in reducing regret bound, as well as reducing the computational cost.

### 3.2 REGRET ANALYSIS

We present the regret bound for OPORS under the linear MDP assumption.

**Theorem 3.1** (Regret bound of OPORS). *Suppose Assumption 1 holds. There exists a constant $C_l > 0$ such that, if we set $\lambda = 1$, $\alpha = \text{poly}(T, 1/\delta, \log|\mathcal{A}|, d) \geq \Omega(\sqrt{K} \log|\mathcal{A}|)$, and $\beta = C_l \cdot d^{3/2} H^{3/2} \chi_l$ where $\chi_l = \frac{1}{\log \eta} \sqrt{\log \left(1 + \frac{K}{\lambda d}\right) \log \left(\frac{dT \log|\mathcal{A}|}{\delta \lambda \log \eta}\right)}$, then with probability at least $1 - \delta$, the regret of Algorithm 1 is upper bounded by*

$$\text{Regret}(K) \leq \tilde{O} \left( \frac{\eta}{\log \eta} d^2 H^2 \sqrt{T} \right).$$

**Discussion of Theorem 3.1.** Theorem 3.1 establishes $\tilde{O}(d^2 H^2 \sqrt{T})$ regret bound. There exists a concurrent work (Sherman et al., 2023) which proves $\tilde{O}(d^2 H^{5/2} \sqrt{T \log |\mathcal{A}|})$. Theorem 3.1 shows an improved regret bound by a factor of $\sqrt{H \log |\mathcal{A}|}$. Hence, our regret bound is the sharpest regret bound known for policy optimization algorithms under linear MDPs. Sherman et al. (2023) utilize a reward-free exploration algorithm adapted from Wagenmaker et al. (2022) to bound optimistic $Q$-values, then conduct policy updates without clipping $Q$-values. In contrast, we integrate our new policy update rule and the rare-switching technique, without an initial exploration phase.

## 3.3 Proof Sketch

In this section, we give an outline of the proof of Theorem 3.1 and the key lemmas for the proof. We begin with the regret decomposition in Cai et al. (2020):

$$\text{Regret}(K) = \sum_{k=1}^{K} \sum_{h=1}^{H} \mathbb{E}_{\pi^*}[\langle Q_h^k(s_h, \cdot), \pi_h^*(\cdot \mid s_h) - \pi_h^k(\cdot \mid s_h)\rangle \mid s_1 = s_1^k] \tag{2}$$

$$+ \sum_{k=1}^{K} \sum_{h=1}^{H} \left( P_h[V_{h+1}^k - V_{h+1}^{\pi^k}](s_h^k, a_h^k) - [V_{h+1}^k - V_{h+1}^{\pi^k}](s_{h+1}^k) \right) \tag{3}$$

$$+ \sum_{k=1}^{K} \sum_{h=1}^{H} \left( \langle [Q_h^k - Q_h^{\pi^k}](s_h^k, \cdot), \pi_h^k(\cdot \mid s_h^k)\rangle - [Q_h^k - Q_h^{\pi^k}](s_h^k, a_h^k) \right) \tag{4}$$

$$+ \sum_{k=1}^{K} \sum_{h=1}^{H} \left( \mathbb{E}_{\pi^*}[\xi_h^k(s_h, a_h) \mid s_1 = s_1^k] - \xi_h^k(s_h, a_h) \right) \tag{5}$$

where $\xi_h^k(\cdot, \cdot) := r_h(\cdot, \cdot) + P_h V_{h+1}^k(\cdot, \cdot) - Q_h^k(\cdot, \cdot)$. Note that (3) and (4) are the sums of uniformly bounded martingale difference sequences. Hence, with high probability $1 - \delta$, the sum of the martingale difference sequences is easily bounded by $O(\sqrt{H^2 T \log(1/\delta)})$.

The statistical error term (5) represents the error arising from the least-square value iteration step (Lines 6-12 of Algorithm 1). The technical challenge in bounding this statistical error lies in the fact that the log covering number of the $V$-funciton class is directly related to the number of policy switches, as Lemma D.8 implies. Consequently, the confidence radius of the estimator $\hat{w}_h^k$ grows accordingly. In extreme cases where the policy is updated every episode, this leads to a trivial linear regret bound. To mitigate this issue, OPORS effectively controls the covering number by infrequently switching the policy. The following lemma from Wang et al. (2021) ensures that the number of policy switches is bounded.

**Lemma 3.2** (Policy switches). *The number of policy switches in Algorithm 1 is upper bounded by*

$$N_l = dH / \log \eta \cdot \log (1 + H/\lambda d).$$

With Lemma 3.2, it can be shown that the log covering number of $V$-function class is $\tilde{O}(d^3 H)$. Utilizing this covering number, we can prove that $\xi_h^k(\cdot, \cdot)$ is bounded.

**Lemma 3.3** (Concentration of value functions). *With probability at least $1 - \delta/2$, for all $k \in [K], h \in [H], (s, a) \in \mathcal{S} \times \mathcal{A}$, it holds that $-2\beta \|\phi(s, a)\|_{(\Lambda_h^{\bar{k}})^{-1}} \leq \xi_h^k(s, a) \leq 0$, where $\bar{k}$ is the largest index $k' \leq k$ on which the policy is switched, and $\beta$ is defined in Theorem 1.*

Lemma 3.3 implies that the statistical error is bounded by the sum of bonus functions, evaluated at the trajectories. The policy optimization error (2) is of significance for policy optimization algorithms, unlike value-based algorithms that employ the argmax policy with respect to the $Q$-value. The distinction arises because $\langle Q_h^k(s_h, \cdot), \pi_h^*(\cdot \mid s_h) - \pi_h^k(\cdot \mid s_h)\rangle$ could be positive for general stochastic policy $\pi_h^k$. Previous approaches, which utilize the update rule $\pi_h^k(\cdot \mid s) \propto \pi_h^{k-1}(\cdot \mid s) \cdot \exp(\alpha Q_h^{k-1}(s, \cdot))$ bound the policy optimization error by $\sqrt{H^3 T \log |\mathcal{A}|}$, using a well-known result of online mirror descent (see Lemma 3.3 in Cai et al. 2020). However, the bound is valid only when policy updates are equally spaced. In contrast, our update rule $\pi_h^k(\cdot \mid s) \propto \pi_h^{k-1}(\cdot \mid s) \cdot \exp(\alpha Q_h^k(s, \cdot))$ guarantees that the policy optimization error is bounded, regardless of the specific update schedule. Lemma 3.4 formally states this property.

---

**Algorithm 2** $\mathcal{F}$-OPORS

---

1: **Input:** Failure probability $\delta \in (0, 1)$, stepsize $\alpha > 0$, confidence radius $\beta > 0$
2: **Initialize:** Set $\{\pi_h^0(\cdot \mid \cdot)\}_{h \in [H]}$ as uniform policy on $\mathcal{A}$, $\hat{\mathcal{Z}}_h^1 \leftarrow \{\}$ for all $h \in [H]$
3: **for** episode $k = 1, \cdots, K$ **do**
4:     **for** step $h = H, \ldots, 1$ **do**
5:         $\hat{\mathcal{Z}}_h^k \leftarrow \mathbf{Sample}\left(\mathcal{F}, \hat{\mathcal{Z}}_h^{k-1}, (s_h^{k-1}, a_h^{k-1}), \delta\right)$ (if $k \leq 2$)
6:     **end for**
7:     **if** $k = 1$ **or** $\exists h \in [H]$ $\hat{\mathcal{Z}}_h^k \neq \hat{\mathcal{Z}}_h^{k-1}$ **then**
8:         **for** step $h = H, \ldots, 1$ **do**
9:             $\mathcal{D}_h^k \leftarrow \{(s_h^\tau, a_h^\tau, r_h^\tau + V_{h+1}^k(s_{h+1}^\tau)\}_{\tau \in [k-1]}$
10:           $f_h^k \leftarrow \arg\min_{f \in \mathcal{F}} \|f\|_{\mathcal{D}_h^k}^2$
11:           $b_h^k(\cdot, \cdot) \leftarrow \sup_{f_1, f_2 \in \mathcal{F}, \|f_1 - f_2\|_{\hat{\mathcal{Z}}_h^k}^2 \leq \beta} |f_1(\cdot, \cdot) - f_2(\cdot, \cdot)|$
12:           $Q_h^k(\cdot, \cdot) \leftarrow \min\left\{f_h^k(\cdot, \cdot) + b_h^k(\cdot, \cdot), H\right\}$
13:           Update $\pi_h^k(\cdot \mid \cdot) \propto \pi_h^{k-1}(\cdot \mid \cdot) \exp(\alpha Q_h^k(\cdot, \cdot))$
14:           $V_h^k(\cdot) \leftarrow \langle Q_h^k(\cdot, \cdot), \pi_h^k(\cdot \mid \cdot)\rangle$
15:     **end for**
16:     **else**
17:         $Q_h^k(s, a) \leftarrow Q_h^{k-1}(s, a), \pi_h^k(a \mid s) \leftarrow \pi_h^{k-1}(a \mid s), V_h^k(s) \leftarrow V_h^{k-1}(s)$ for all $h \in [H]$
18:     **end if**
19:     **for** step $h = 1, \ldots, H$ **do**
20:         Take an action $a_h^k \sim \pi_h^k(\cdot \mid s_h^k)$ and observe $s_{h+1}^k$
21:     **end for**
22: **end for**

---

**Algorithm 3** $\mathbf{Sample}(\mathcal{F}, \hat{Z}, z, \delta)$

---

1: **Input:** Function class $\mathcal{F}$, current sub-sampled dataset $\hat{\mathcal{Z}} \subseteq \mathcal{S} \times \mathcal{A}$, new stat-action pair $z$, failure probability $\delta$
2: Let $p_z$ be the smallest real number such that $1/p_z$ is an integer and

$$p_z \geq \min\left\{1, C \cdot \text{sensitivity}_{\hat{\mathcal{Z}}, \mathcal{F}}(z) \cdot \log\left(T\mathcal{N}(\mathcal{F}, \sqrt{\delta/64T^3})/\delta\right)\right\}$$

3: Let $\hat{z} \in \mathcal{C}(\mathcal{S} \times \mathcal{A}, 1/(16\sqrt{64T^3/\delta}))$ such that $\sup_{f \in \mathcal{F}} |f(z) - f(\hat{z})| \leq 1/(16\sqrt{64T^3/\delta})$
4: Add $1/p_z$ copies of $\hat{z}$ into $\hat{\mathcal{Z}}$ with probability $p_z$
5: **return** $\hat{\mathcal{Z}}$

---

**Lemma 3.4** (Policy optimization error). *It holds that*

$$\sum_{k=1}^K \sum_{h=1}^H \mathbb{E}_{\pi^*}\left[\langle Q_h^k(s_h, \cdot), \pi_h^*(\cdot \mid s_h) - \pi_h^k(\cdot \mid s_h)\rangle \mid s_1 = s_1^k\right] \leq \alpha^{-1} HK \log|\mathcal{A}|$$

We emphasize that Lemma 3.4 places no restrictions on the sequence of $Q$-values, except the boundedness $|Q_h^k| \leq H$. We can bound the policy optimization error corresponding to any sequence $\{Q_h^k\}_{(k,h) \in [K] \times [H]}$, with carefully chosen $\alpha$.

Since $Q_h^k(s, \cdot)$ are $|\mathcal{A}|$-dimensional vectors and $\pi_h^k(\cdot \mid s)$ are unit vectors in the same space, each term $\langle Q_h^k(s_h, \cdot), \pi_h^*(\cdot \mid s_h) - \pi_h^k(\cdot \mid s_h)\rangle$ can be understood as the difference between two inner products, $\langle Q_h^k(s, \cdot), \pi_h^*(\cdot \mid s)\rangle$ and $\langle Q_h^k(s, \cdot), \pi_h^k(\cdot \mid s)\rangle$. Hence, the policy optimization error decreases if $\pi_h^k(\cdot \mid s)$ aligns more closely with $Q_h^k(s, \cdot)$ than $\pi_h^*(\cdot \mid s)$ does. This is the intuition behind why our update rule yields a smaller policy optimization error. Unlike the previous update rule, our update rule directly incorporates $Q_h^k$ into $\pi_h^k$.

With these lemmas at hand, we can bound the cumulative regret of Algorithm 1. The detailed proofs are provided in Appendix C.

## 4 GENERAL FUNCTION APPROXIMATION

In this section, we extend our rare-switching optimistic policy optimization method to general function approximation. The extended algorithm, $\mathcal{F}$-OPORS is described in Algorithm 2.

### 4.1 ALGORITHM: $\mathcal{F}$-OPORS

The overall structure of $\mathcal{F}$-OPORS is similar to our first algorithm OPORS, except for the rare-switching technique. In the following, we describe each component of the algorithm.

**Sampling.** Utilizing online sentitivity sampling (Line 5 of Algorithm 2) introduced by Kong et al. (2021) effectively reduces policy switching. During the collection of trajectories into the dataset $\mathcal{Z}_h^k := \{(s_h^\tau, a_h^\tau, r_h^\tau, s_{h+1}^\tau)\}_{\tau \in [k-1]}$, we concurrently maintain a subsampled dataset $\{\hat{\mathcal{Z}}_h^k\}_{h \in [H]}$ that approximates the original dataset with reduced complexity. In the $k$-th episode, we calculate the online sensitivity score based on the current subsampled dataset $\{\hat{\mathcal{Z}}_h^{k-1}\}_{h \in [H]}$ and the trajectory from the previous episode $\{(s_h^{k-1}, a_h^{k-1})\}_{h \in [H]}$, as follows:

$$\text{sensitivity}_{\hat{\mathcal{Z}}_h^{k-1}, \mathcal{F}}(z_h^{k-1}) := \min \left\{ \sup_{f_1, f_2 \in \mathcal{F}} \frac{(f_1(z_h^{k-1}) - f_2(z_h^{k-1}))^2}{\min\{\|f_1 - f_2\|_{\hat{\mathcal{Z}}_h^{k-1}}, T(H+1)^2\} + \beta}, 1 \right\}.$$

Each $z_h^{k-1} := (s_h^{k-1}, a_h^{k-1})$ is stochastically sampled into $\hat{\mathcal{Z}}_h^k$ with a probability proportional to the sensitivity score. Additionally, the sampled $z_h^{k-1}$ is rounded to belong to a finite cover of $\mathcal{S} \times \mathcal{A}$ to bound the complexity of $\hat{\mathcal{Z}}_h^k$. After the sampling procedure, we evaluate and update the policy only when any of the $z_h^{k-1}$ is sampled. Otherwise, we proceed without changing the value function estimate and the policy. By implementing this strategy, the number of policy switches remains less than or equal to the number of distinct elements in $\{\hat{\mathcal{Z}}_h^K\}_{h \in [H]}$, which is bounded by $\tilde{O}(d_g^2 H)$ with high probability.

**Value Function Estimation and Policy Update.** In the $k$-th episode, if any of the $z_h^{k-1}$ instances is sampled during the sampling procedure, we proceed to compute the optimistic $Q$-values using the least-square value iteration, as described in Lines 9-13 of Algorithm 2. For each $h = H, \ldots, 1$, given the dataset $\mathcal{D}_h^k := \{(s_h^\tau, a_h^\tau, r_h^\tau + V_{h+1}^k(s_{h+1}^\tau))\}_{\tau \in [k-1]}$, we solve the regression problem:

$$f_h^k = \arg\min_{f \in \mathcal{F}} \|f\|_{\mathcal{D}_h^k}^2 = \arg\min_{f \in \mathcal{F}} \sum_{\tau=1}^{k-1} \left( f(s_h^\tau) - \left[ r_h^\tau + V_{h+1}^k(s_{h+1}^\tau) \right] \right)^2.$$

Subsequently, we set the optimistic $Q$-value $Q_h^k(\cdot, \cdot) = \min \left\{ f_h^k(\cdot, \cdot) + b_h^k(\cdot, \cdot), H \right\}$, where the bonus function is defined as follows:

$$b_h^k(\cdot, \cdot) := \sup_{f_1, f_2 \in \mathcal{F}, \|f_1 - f_2\|_{\hat{\mathcal{Z}}_h^k}^2 \le \beta} |f_1(\cdot, \cdot) - f_2(\cdot, \cdot)|.$$

Intuitively, the bonus function reflects the level of confidence in the value estimation, thereby promoting the exploration of rarely observed state-action pairs. The policy update process mirrors that of OPORS. After updating $\pi_h^k$, the optimistic $V$-value is defined as $V_h^k(\cdot) = \langle Q_h^k(\cdot, \cdot), \pi_h^k(\cdot \mid \cdot) \rangle$.

### 4.2 REGRET ANALYSIS

In this section, we present the regret bound for $\mathcal{F}$-OPORS. The following standard assumptions (Wang et al., 2020) are required for our analysis. To measure the complexity of a function class, we use the eluder dimension (Russo & Van Roy, 2013).

**Assumption 2** (Value closedness). *For any $V \to [0, H]$ and $h \in [H]$, $r_h + P_h V \in \mathcal{F}$.*

**Assumption 3** (Covering number). *For any $\varepsilon > 0$, the function class $\mathcal{F}$ satisfies the following.*

1. *There exists an $\varepsilon$-cover $\mathcal{C}(\mathcal{F}, \varepsilon) \subseteq \mathcal{F}$ with size $|\mathcal{C}(\mathcal{F}, \varepsilon)| \le \mathcal{N}(\mathcal{F}, \varepsilon)$, such that for any $f \in \mathcal{F}$, there exists $f' \in \mathcal{F}$ with $\|f - f'\|_\infty \le \varepsilon$.*

2. *There exists an $\varepsilon$-cover $\mathcal{C}(\mathcal{S} \times \mathcal{A}, \varepsilon) \subseteq \mathcal{S} \times \mathcal{A}$ with size $|\mathcal{C}(\mathcal{S} \times \mathcal{A}, \varepsilon)| \le \mathcal{N}(\mathcal{S} \times \mathcal{A}, \varepsilon)$, such that for any $(s, a) \in \mathcal{S} \times \mathcal{A}$, there exists $(s', a') \in \mathcal{S} \times \mathcal{A}$ with $\sup_{f \in \mathcal{F}} |f(s, a) - f(s', a')| \le \varepsilon$.*

**Definition 1** (Eluder dimension). *Let $\varepsilon > 0$ and $\mathcal{Z} = \{(s_i, a_i)\}_{i \in [n]}$.*

1. *A state-action pair $(s, a)\mathcal{S} \times \mathcal{A}$ is $\varepsilon$-dependent on $\mathcal{Z}$ with respect to $\mathcal{F}$ if any $f, f' \in \mathcal{F}$ satisfying $\|f - f'\|_{\mathcal{Z}} \leq \varepsilon$ also satisfies $|f(s, a) - f'(s, a)| \leq \varepsilon$.*

2. *An $(s, a)$ is $\varepsilon$-independent of $\mathcal{Z}$ with respect to $\mathcal{F}$ if $(s, a)$ is not $\varepsilon$-dependent on $\mathcal{Z}$.*

3. *The $\varepsilon$-eluder dimension $dim_E(\mathcal{F}, \varepsilon)$ is a function class $\mathcal{F}$ is the length of the longest sequence of elements in $\mathcal{S} \times \mathcal{A}$ such that, for some $\varepsilon' \geq \varepsilon$, every element is $\varepsilon'$-independent of its predecessors.*

When $\mathcal{F} = \{\theta^T \phi(s, a) : \theta \in \Theta \subseteq \mathbb{R}^d\}$ is the class of linear functions with a feature map $\phi : \mathcal{S} \times \mathcal{A} \to \mathbb{R}^d$, it is known that $dim_E(\mathcal{F}, \varepsilon) = O(d \log(1/\varepsilon))$ and $\log \mathcal{N}(\mathcal{S} \times \mathcal{A}, \epsilon) = \tilde{O}(d)$ (Russo & Van Roy, 2013). Thus, we can roughly understand that $dim_E(\mathcal{F}, \cdot) \approx \mathcal{N}(\mathcal{S} \times \mathcal{A}, \cdot)$ measure the complexity of a given function class. Equipped with these concepts, we state the regret bound.

**Theorem 4.1** (Regret bound of $\mathcal{F}$-OPORS ). *Suppose Assumption 2 and Assumption 3 hold. There exist absolute constants $C_g, C_s, C_{sw}$ such that, if we set $\alpha \geq \Omega(\sqrt{T} \log |\mathcal{A}|)$,*

$$\beta = C_g \cdot H^3 \cdot \log(T\mathcal{N}(\mathcal{F}, \delta/T^2)/\delta) \cdot dim_E(\mathcal{F}, 1/T) \cdot \log^2 T \cdot [\log \mathcal{N}(\mathcal{F}, 1/(16\alpha N_g T^2 H^3))$$
$$+ \log(T\mathcal{N}(\mathcal{F}, \delta/T^2)/\delta) \cdot dim_E(\mathcal{F}, 1/T) \cdot \log^2 T \cdot \log(\mathcal{N}(\mathcal{S} \times \mathcal{A}, \delta/T^2) \cdot T/\delta)],$$

*in Algorithm 2 and $C = C_s$ in Algorithm 3, then with probability at least $1 - \delta$, the regret of Algorithm 2 is upper bounded by*

$$\text{Regret}(K) \leq O\left(H^2 \sqrt{\iota \cdot T}\right)$$

*where* $\iota = \log(T\mathcal{N}(\mathcal{F}, \delta/T^2)/\delta) \cdot dim_E(\mathcal{F}, 1/T)^2 \cdot \log^2 T \cdot [\log \mathcal{N}(\mathcal{F}, 1/(16\alpha N_g T^2 H^3))$
$$+ \log(T\mathcal{N}(\mathcal{F}, \delta/T^2)/\delta) \cdot dim_E(\mathcal{F}, 1/T) \cdot \log^2 T \cdot \log(\mathcal{N}(\mathcal{S} \times \mathcal{A}, \delta/T^2) \cdot T/\delta)],$$
$$N_g = C_{sw} \cdot \log(T\mathcal{N}(\mathcal{F}, \sqrt{\delta/64T^3}/\delta)) \cdot dim_E(\mathcal{F}, 1/T) \cdot \log^2 T.$$

**Discussion of Theorem 4.1** This is the first regret guarantee for a policy optimization algorithm with general function approximation. When $\mathcal{F}$-OPORS is applied to the class of $d$-dimensional linear functions, Theorem 2 implies $\tilde{O}(d^2 H)$ policy switches and $\tilde{O}(d^3 H^2 \sqrt{T})$ regret bound. However, as Kong et al. (2021) pointed out, the spectral approximation algorithm in Cohen et al. (2016) can be used to construct a subsampled dataset. In Appendix E, we present a policy optimization algorithm using the techniques in Cohen et al. (2016), that achieves $\tilde{O}(dH)$ policy switches and $\tilde{O}(d^2 H^2 \sqrt{T})$ regret bound.

### 4.3 PROOF SKETCH

The proof of Theorem 4.1 shares the same outline with Theorem 3.1. Using the same regret decomposition, we bound (2) with Lemma 3.4, and bound (3), (4) in the same way. The difference lies in the rare-switching technique. Specifically, the following lemma from Kong et al. (2021) provides a polylog($T$) bound on the number of policy switching.

**Lemma 4.2** (Policy switches). *For $\beta$ specified in Theorem 4.1, with probability at least $1 - \delta/8$, for all $(k, h) \in [K] \times [H]$, the number of distinct elements in $\hat{\mathcal{Z}}_h^K$ is bounded by*

$$N_g \leq O\left(\cdot \log(T\mathcal{N}(\mathcal{F}, \sqrt{\delta/64T^3}/\delta)) \cdot dim_E(\mathcal{F}, 1/T) \cdot \log^2 T\right).$$

*Therefore, the number of policy switches in Algorithm 2 is bounded by $H \cdot N_g$.*

Therefore, we can bound the covering number of value functions. The bound is applied to a uniform concentration inequality and then gives the following lemma.

**Lemma 4.3** (Concentration of value function). *With probability at least $1 - \delta/8$, for all $(k, h) \in [K] \times [H]$ and for all $(s, a) \in \mathcal{S} \times \mathcal{A}$, it holds that $-2b_h^k(s, a) \leq \xi_h^k(s, a) \leq 0$.*

Therefore, the statistical error term (5) is bounded by the sum of bonus functions, evaluated at the trajectories. The proof is complete by bounding the sum of bonuses using the property of the eluder dimension. The detailed proof is deferred to Appendix D.

## 5 NUMERICAL EXPERIMENTS

In this section, we evaluate our algorithm on RiverSwim (Osband et al., 2013) and DeepSea (Osband et al., 2019) environments. These environments are considered to be hard exploration problems that require deep, strategic exploration. To compare the performance of our algorithm with other provably efficient RL algorithms for linear MDPs, we choose value-based algorithms, `LSVI-UCB` (Jin et al., 2020), `LSVI-PHE` (Ishfaq et al., 2021) and a policy-based algorithm `OPPO+` (Zhong & Zhang, 2023) for comparisons.[2] For each algorithm, we performed a grid search to set the hyperparameters. We run 20 independent instances for each experiment and plot the average performance with the standard deviation. For more details, please refer to Appendix G.

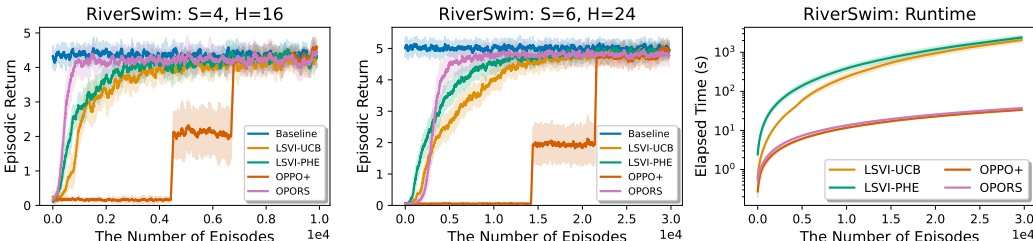

Figure 1: Results on RiverSwim. (left) Episodic return on $S = 4, H = 16$, (center) Episodic return on $S = 6, H = 24$, (right) Runtime of the algorithms

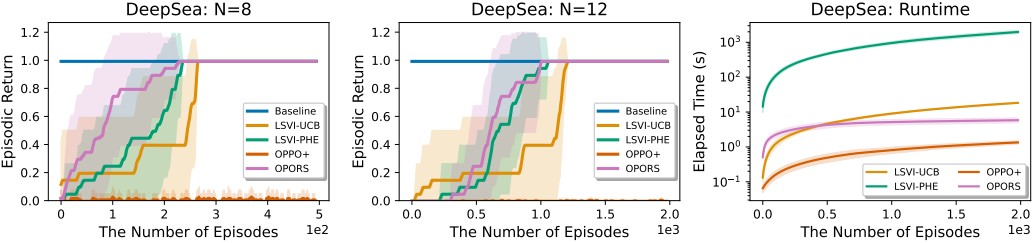

Figure 2: Results on DeepSea. (left) Episodic return on $N = 8$, (center) Episodic return on $N = 12$, (right) Runtime of the algorithms

We performed experiments on the two RiverSwim instances, $|\mathcal{S}| = 4$ and $|\mathcal{S}| = 6$. The episode lengths are set to be $H = 4|\mathcal{S}|$. Figure 1 shows that `OPORS` outperforms the value-based methods, and `OPPO+` lags behind the others since the batch size suggested by Zhong & Zhang (2023) is too large. Note that `OPORS` achieves high performance at much less computational cost, as the runtime of `OPORS` over 30000 episodes is around two orders of magnitude shorter than that of `LSVI-UCB` and `LSVI-PHE`. Such efficiency comes from rare policy switching, as `OPORS` requires only $\tilde{O}(dH \log T)$ value estimations and policy updates. The experiments on two DeepSea instances, $N = 8$ and $N = 12$, shown in Figure 2 exhibit consistent results. Again, `OPORS` is statistically efficient compared to `LSVI-UCB` and `LSVI-PHE` while being far more computationally efficient.

## 6 CONCLUSION

In this paper, we propose optimistic policy optimization algorithms with rare policy switching: `OPORS` for linear MDPs and $\mathcal{F}$-`OPORS` for general function approximation. For both algorithms, we prove regret bounds of $\tilde{O}(d^2 H^2 \sqrt{T})$ and $\tilde{O}(d_g^3 H^2 \sqrt{T})$, respectively. Therefore, our algorithms are the first policy optimization method establishing $\tilde{O}(\sqrt{T})$ regret bound for both settings with the sharpest result for linear MDPs. Our empirical evaluations further highlight the competitive performance and reduced computational cost of our proposed method.

---

[2]Since the algorithm suggested by Sherman et al. (2023) is impractical due to a large number of hyperparameters and subroutines, we did not include it in our experiments.

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

# A  ADDITIONAL RELATED WORKS

**RL with Linear Function Approximation.**  There exist extensive works on RL with linear function approximation (Jiang et al., 2017; Du et al., 2019; Wang et al., 2019; Sun et al., 2019; Yang & Wang, 2019; Zanette et al., 2020b; Modi et al., 2020; Jin et al., 2020; Cai et al., 2020; Jia et al., 2020; Ayoub et al., 2020; Zanette et al., 2020a; Ishfaq et al., 2021; Zhou et al., 2021; Zhou & Gu, 2022; Hwang & Oh, 2023; He et al., 2023). The most related works to ours is the linear MDP model introduced by Yang & Wang (2019) and Jin et al. (2020), which assumes that rewards and transition probabilities are linear functions of some $d$-dimensional feature mapping $\phi(s, a)$. Based on the analysis from linear contextual bandit literature (Chu et al., 2011; Abbasi-Yadkori et al., 2011; Agrawal & Goyal, 2013; Abeille & Lazaric, 2017), Jin et al. (2020) establish $\tilde{O}(\sqrt{d^3 H^3 T})$ regret bound. Zanette et al. (2020a) develop a randomized least-square value iteration algorithm and prove a frequentist regret bound of $\tilde{O}(\sqrt{d^4 H^4 T})$. Ishfaq et al. (2021) also proposed a randomized algorithm with $\tilde{O}(\sqrt{d^3 H^3 T})$, which achieves optimism by taking a maximum value over a number of samples. Recently, He et al. (2023) prove a nearly minimax optimal regret bound $\tilde{O}(dH\sqrt{T})$ using a Bernstein-type concentration (Zhou & Gu, 2022) and monotonic value estimation.

**RL with General Function Approximation.**  Since many successful applications of RL rely on complex function approximation such as neural networks, an increasing number of works focus on the theoretical guarantee with general function approximation (Russo & Van Roy, 2013; Jiang et al., 2017; Du et al., 2021; Agarwal et al., 2020a; Feng et al., 2021; Wang et al., 2020; Jin et al., 2021; Kong et al., 2021; Agarwal et al., 2023). Each work defines a complexity measure for a general function class, with some assumptions required for theoretical guarantees. Among these complexity measures, we use the eluder dimension (Russo & Van Roy, 2013) to measure the complexity of general function classes. The eluder dimension was first introduced in Russo & Van Roy (2013), and they prove Bayesian regret bounds of Thompson sampling based bandit algorithms. Using the properties of the eluder dimension, Wang et al. (2020) propose a model-free value-based algorithm. They present the sampling method called sensitivity sampling to bound the complexity of bonus functions, and establish a regret bound $\tilde{O}(d_g^2 H\sqrt{T})$. Building upon this work, Kong et al. (2021) develop an improved sampling method that enables $O(\log T)$ rare policy switching while preserving the regret bound. Jin et al. (2021) proposed the Bellman Eluder dimension which includes the notion of Bellman rank and the eluder dimension, and prove $\tilde{O}(d_g H\sqrt{T})$ under the assumption that episodic return is bounded by 1. Recently, Agarwal et al. (2023) proposed an algorithm based on weighted least square and the general eluder dimension. They establish $\tilde{O}(d_g\sqrt{HT})$ regret bound with a sparse reward assumption, which is nearly minimax optimal in the linear MDP setting.

**Policy Optimization.**  Several works study the convergence guarantees for policy optimization algorithms (Kakade & Langford, 2002; Bagnell et al., 2003; Bhandari & Russo, 2019; Geist et al., 2019; Agarwal et al., 2020b; Shani et al., 2020; Cen et al., 2022), under the assumption that the starting state distribution well covers the state-action space. However, we focus on the setting without such assumptions, hence the agent should balance exploration and exploitation. Moreover, there exists a line of works that prove the sample complexity of policy optimization algorithms to learn a near-optimal policy. Agarwal et al. (2020a) develop a policy optimization algorithm for general function approximation, and establish $\tilde{O}(\frac{\text{poly}(d, 1/(1-\gamma))}{\epsilon^{11}})$ sample complexity bound in discounted linear MDPs. Feng et al. (2021) extend the idea for general function approximation, and prove $\tilde{O}(\frac{\text{poly}(d_g, 1/(1-\gamma))}{\epsilon^8})$ sample complexity bound. Agarwal et al. (2020a); Feng et al. (2021) are robust to the model misspecification in that the additional sample complexity due to the misspecification depends on the average model error, at the cost of large sample complexity. Zanette et al. (2021) improve the sample complexity to $\tilde{O}(\frac{d^3}{(1-\gamma)^{13}\epsilon^3})$ by using fewer samples for value estimation, but the result is specific to the linear function approximation. Building upon Feng et al. (2021), Li et al. (2023) achieve $\tilde{O}(\frac{d_g^3}{(1-\gamma)^8\epsilon^3})$ sample complexity bound using the sensitivity sampling introduced by Kong et al. (2021). In a slightly different setting of episodic MDP, Liu et al. (2023) proposed a simple on-policy batch policy optimization method with $\tilde{O}(\frac{d_g^2 H^6}{\epsilon^3})$ sample complexity bound. Another line of work seeks to prove regret bound for policy optimization algorithms. Cai et al. (2020) propose a policy update method that is equivalent to the proximal policy optimization (PPO), and

establish a regret bound of $\tilde{O}(\sqrt{d^2 H^3 T \log |\mathcal{A}|})$ in adversarial linear mixture MDPs. Subsequently, He et al. (2023) proves nearly minimax optimal regret in the same setting by adapting the Bernstein-type concentration inequality from Zhou et al. (2021). On the other hand, Shani et al. (2020) proves $\tilde{O}(\sqrt{S^2 A H^3 T})$ regret bound for tabular MDPs, where $S, A$ are the number of states and actions, respectively. In the same tabular setting, Wu et al. (2022) achieves nearly minimax optimal regret bound, with an assumption $S > H$. However, theoretical analyses in linear MDPs are relatively limited. Zhong & Zhang (2023) is the first to prove sub-linear regret $\tilde{O}(d^{3/4} H^{5/4} T^{3/4} \log |\mathcal{A}|)$ for a policy optimization algorithm, in adversarial linear MDPs. They propose a batch update method to reduce the number of policy switches. Recently, an independent and concurrent work Sherman et al. (2023) established a $\tilde{O}(d^2 H^{5/2} \sqrt{T \log |\mathcal{A}|})$ in adversarial linear MDPs, by controlling the size of $Q$-values via reward-free warm-up phase (Wagenmaker et al., 2022).

# B   PROOF OF LEMMA 3.4

*Proof.* Let $\{k_i\}_{i \in m}$ be the set of episodes on which the value function and the policy are updated. For convenience, define $k_0 = 0$ and $k_{m+1} = K + 1$. Note that the update rule is $\pi_h^{k_i}(\cdot \mid s) \propto \pi_h^{k_{i-1}}(\cdot \mid s) \exp\left(\alpha Q_h^{k_i}(s, \cdot)\right)$. To simplify the notation, we define $\gamma_h^{k_i}(\cdot) = \log\left(\sum_{a \in A} \pi_h^{k_i}(a \mid \cdot) \exp(\alpha Q_h^{k_i}(\cdot, a)\right)$. Then it holds that

$$
\begin{aligned}
&\langle \alpha Q_h^{k_i}(s_h, \cdot), \pi_h^*(\cdot \mid s_h) - \pi_h^{k_i}(\cdot \mid s_h)\rangle \\
&= \sum_a \alpha Q_h^{k_i}(s_h, a)\left(\pi_h^*(a \mid s_h) - \pi_h^{k_i}(a \mid s_h)\right) \\
&= \sum_a \left(\log \pi_h^{k_i}(a \mid s_h) - \log \pi_h^{k_{i-1}}(a \mid s_h) + \gamma_h^{k_i}(s_h)\right)\left(\pi_h^*(a \mid s_h) - \pi_h^{k_i}(a \mid s_h)\right) \\
&= \sum_a \left(\log \pi_h^{k_i}(a \mid s_h) - \log \pi_h^{k_{i-1}}(a \mid s_h)\right)\left(\pi_h^*(a \mid s_h) - \pi_h^{k_i}(a \mid s_h)\right) + \sum_a \gamma_h^{k_i}(s_h)\left(\pi_h^*(a \mid s_h) - \pi_h^{k_i}(a \mid s_h)\right) \\
&= \sum_a \left(\log \pi_h^{k_i}(a \mid s_h) - \log \pi_h^{k_{i-1}}(a \mid s_h)\right)\left(\pi_h^*(a \mid s_h) - \pi_h^{k_i}(a \mid s_h)\right)
\end{aligned}
$$

where the last equality holds because $\gamma(\cdot)$ is a function of states. Furthermore, it follows that

$$
\begin{aligned}
&\sum_a \left(\log \pi_h^{k_i}(a \mid s_h) - \log \pi_h^{k_{i-1}}(a \mid s_h)\right)\left(\pi_h^*(a \mid s_h) - \pi_h^{k_i}(a \mid s_h)\right) \\
&= \langle \log \pi_h^{k_i}(\cdot \mid s_h) - \log \pi_h^{k_{i-1}}(\cdot \mid s_h), \pi_h^*(\cdot \mid s_h) - \pi_h^{k_i}(\cdot \mid s_h)\rangle \\
&= \langle \log \frac{\pi_h^*(\cdot \mid s_h)}{\pi_h^{k_{i-1}}(\cdot \mid s_h)} - \log \frac{\pi_h^*(\cdot \mid s_h)}{\pi_h^{k_i}(\cdot \mid s_h)}, \pi_h^*(\cdot \mid s_h)\rangle - \langle \log \frac{\pi_h^{k_i}(\cdot \mid s_h)}{\pi_h^{k_{i-1}}(\cdot \mid s_h)}, \pi_h^{k_i}(\cdot \mid x_h)\rangle \\
&= D_{KL}(\pi_h^*(\cdot \mid s_h)\|\pi_h^{k_{i-1}}(\cdot \mid s_h)) - D_{KL}(\pi_h^*(\cdot \mid s_h)\|\pi_h^{k_i}(\cdot \mid s_h)) - D_{KL}(\pi_h^{k_i}(\cdot \mid s_h)\|\pi_h^{k_{i-1}}(\cdot \mid s_h)) \\
&\leq D_{KL}(\pi_h^*(\cdot \mid s_h)\|\pi_h^{k_{i-1}}(\cdot \mid s_h)) - D_{KL}(\pi_h^*(\cdot \mid s_h)\|\pi_h^{k_i}(\cdot \mid s_h))
\end{aligned}
$$

where the inequality holds because KL divergence is always non-negative. Since we only update $Q_h^k(\cdot, \cdot), V_h^k(\cdot)$, and $\pi_h^k(\cdot \mid \cdot)$ at $k_i$s, the term $\mathbb{E}_{\pi^*}[\langle Q_h^k(s_h, \cdot), \pi_h^*(\cdot \mid s_h) - \pi_h^k(\cdot \mid s_h)\rangle]$ is repeated

until the next update. Thus we have

$$\sum_{k=1}^{K}\sum_{h=1}^{H}\mathbb{E}_{\pi^*}\left[\langle Q_h^k(s_h,\cdot),\pi_h^*(\cdot\mid s_h)-\pi_h^k(\cdot\mid s_h)\rangle\right]$$

$$=\sum_{h=1}^{H}\sum_{i=1}^{m}(k_{i+1}-k_i)\mathbb{E}_{\pi^*}\left[\langle Q_h^{k_i}(s_h,\cdot),\pi_h^*(\cdot\mid s_h)-\pi_h^{k_i}(\cdot\mid s_h)\rangle\right]$$

$$\leq\alpha^{-1}\sum_{h=1}^{H}\sum_{i=1}^{m}(k_{i+1}-k_i)\mathbb{E}_{\pi^*}\left[D_{KL}(\pi_h^*(\cdot\mid s_h)\|\pi_h^{k_{i-1}}(\cdot\mid s_h))-D_{KL}(\pi_h^*(\cdot\mid s_h)\|\pi_h^{k_i}(\cdot\mid s_h))\right]$$

$$\leq\alpha^{-1}\sum_{h=1}^{H}\sum_{i=1}^{m}K\mathbb{E}_{\pi^*}\left[D_{KL}(\pi_h^*(\cdot\mid s_h)\|\pi_h^{k_{i-1}}(\cdot\mid s_h))-D_{KL}(\pi_h^*(\cdot\mid s_h)\|\pi_h^{k_i}(\cdot\mid s_h))\right]$$

$$=\alpha^{-1}\sum_{h=1}^{H}K\mathbb{E}_{\pi^*}\left[D_{KL}(\pi_h^*(\cdot\mid s_h)\|\pi_h^{0}(\cdot\mid s_h))-D_{KL}(\pi_h^*(\cdot\mid s_h)\|\pi_h^{k_m}(\cdot\mid s_h))\right]$$

$$\leq\alpha^{-1}HK\log|\mathcal{A}|$$

where the second inequality holds due to the fact that $k_i\in[K+1]$ for all $i\in[m+1]$, and the last inequality holds because $D_{KL}(\pi_h^*(\cdot\mid s)\|\pi_h^0(\cdot\mid s))=\sum_a\pi_h^*(a\mid s)\log\frac{\pi_h^*(a|s)}{\pi_h^0(a|s)}=\sum_a\left(\pi_h^*(a\mid s)\log\pi_h^*(a\mid s)+\pi_h^*(a\mid s)\log|\mathcal{A}|\right)\leq\log|\mathcal{A}|$.

$\square$

## C  ANALYSIS FOR THEOREM 3.1

In this section, we provide detailed proof of Theorem 3.1. First, define the following filtrations.

$$\mathbb{F}^k:=\sigma\left(\{(s_h^\tau,a_h^\tau,r_h^\tau)\}_{(\tau,h)\in[k]\times[H]}\right)$$
$$\mathbb{F}_{1,h}^k:=\sigma\left(\mathbb{F}^{k-1}\cup\{(s_j^k,a_j^k,r_j^k)\}_{j\in[h]}\right)$$
$$\mathbb{F}_{2,h}^k:=\sigma\left(\mathbb{F}^{k-1}\cup\{(s_j^k,a_j^k,r_j^k)\}_{j\in[h-1]}\cup s_h^k\right)$$

where $\sigma(A)$ is the $\sigma$-algebra generated by $A$.

### C.1  PROOF OF LEMMA 3.2

This is proved in Wang et al. (2021), and we include it for completeness.

*Proof of Lemma 3.2.* Let $\{k_i\}_{i\in m}$ be the set of episodes on which the value function and the policy are updated. For convenience, define $k_0=0$. Then according to Line 5 of Algorithm 1, for all $i\in[m]$, there exists $h\in[H]$ such that

$$\det\left(\Lambda_h^{k_i}\right)\geq\eta\det\left(\Lambda_h^{k_{i-1}}\right).$$

On the other hand, by the definition of $\Lambda_h^k$, we know $\Lambda_h^{k_i}\succeq\Lambda_h^{k_{i-1}}$ and $\det\left(\Lambda_h^{k_i}\right)\geq\det\left(\Lambda_h^{k_{i-1}}\right)$ for all $h\in[H]$. Thus, we have

$$\prod_{h=1}^{H}\det\left(\Lambda_h^{k_i}\right)\geq\eta\prod_{h=1}^{H}\det\left(\Lambda_h^{k_{i-1}}\right).$$

Repeating the inequality, we get

$$\prod_{h=1}^{H}\det\left(\Lambda_h^{k_m}\right)\geq\eta^m\prod_{h=1}^{H}\det\left(\Lambda_h^{k_0}\right)=\eta^m\lambda^{dH}$$

where we use $\Lambda_h^{k_0} = \lambda I$. Finally, since $\det\left(\Lambda_h^{k_m}\right) \leq \det\left(\Lambda_h^K\right)$ for all $h \in [H]$ and $\det\left(\Lambda_h^K\right) \leq (\lambda + K/d)^d$ by Lemma F.11, we have

$$\eta^m \lambda^{dH} \leq \prod_{h=1}^{H} \det\left(\Lambda_h^{k_m}\right) \leq \prod_{h=1}^{H} \det\left(\Lambda_h^K\right) \leq (\lambda + K/d)^{dH},$$

which immediately implies

$$m \leq \frac{dH}{\log \eta} \log\left(1 + \frac{K}{\lambda d}\right)$$

$\square$

### C.2    CONCENTRATION

The following lemma bound the error due to the least square value iteration.

**Lemma C.1.** *With probability at least $1 - \delta/2$, for all $k \in [K], h \in [H], (s, a) \in \mathcal{S} \times \mathcal{A}$, it holds that*

$$|\phi(s, a)^T \hat{w}_h^k - r_h(s, a) - P_h V_{h+1}^k(s, a)| \leq \beta \|\phi(s, a)\|_{(\Lambda_h^k)^{-1}}$$

*where $\beta$ is defined by $\beta = C_l d^{3/2} H^{3/2} \chi_l$ with $\chi_l = \frac{1}{\log \eta} \sqrt{\log\left(1 + \frac{K}{\lambda d}\right) \log\left(\frac{dT \log |\mathcal{A}|}{\delta \lambda \log \eta}\right)}$ for some constant $C_l$.*

*Proof.* By Lemma F.4, we can find $w_h^k$ such that $\phi(s, a)^T w_h^k = r_h(s, a) + P_h V_{h+1}^k(s, a)$ for all $(s, a) \in \mathcal{S} \times \mathcal{A}$. Therefore, we have

$$\phi(s, a)^T \hat{w}_h^k - r_h(s, a) - P_h V_{h+1}^k(s, a)$$

$$= \phi(s, a)^T (\Lambda_h^k)^{-1} \sum_{i=1}^{k-1} \phi(s_h^i, a_h^i)[r_h(s_h^i, a_h^i) + V_{h+1}^k(s_{h+1}^i)] - \phi(s, a)^T w_h^k$$

$$= \phi(s, a)^T (\Lambda_h^k)^{-1} \sum_{i=1}^{k-1} \phi(s_h^i, a_h^i)[r_h(s_h^i, a_h^i) + V_{h+1}^k(s_{h+1}^i)]$$

$$\quad - \phi(s, a)^T (\Lambda_h^k)^{-1} \left(\sum_{i=1}^{k-1} \phi(s_h^i, a_h^i)\phi(s_h^i, a_h^i)^T w_h^k + \lambda w_h^k\right)$$

$$= \phi(s, a)^T (\Lambda_h^k)^{-1} \sum_{i=1}^{k-1} \phi(s_h^i, a_h^i) \left[V_{h+1}^k(s_{h+1}^i) - P_h V_{h+1}^k(s_h^i, a_h^i)\right] - \lambda \phi(s, a)^T (\Lambda_h^k)^{-1} w_h^k$$

where the second last equality uses the definition of $\Lambda_h^k$.

Since Lemma 3.2 states that the number of policy switches is bounded by $N_l = \frac{dH}{\log \eta} \log\left(1 + K/\lambda d\right)$, we can bound the covering number of the class of $V$-functions. Applying Lemma F.12 with $M = N_l$, in conjunction with Lemma F.8 and Lemma F.6, we have

$$\log \mathcal{N}(\mathcal{V}, \varepsilon) \leq \log \mathcal{N}(\mathcal{Q}, \varepsilon/2) + \log \mathcal{N}(\Pi, \varepsilon/2H)$$

$$\leq \log \mathcal{N}(\mathcal{Q}, \varepsilon/2) + N_l \log \mathcal{N}(\mathcal{Q}, \varepsilon/(16\alpha M H^2))$$

$$\leq \left[d \log(1 + 16H\sqrt{dK/\lambda}/\varepsilon) + d^2 \log(1 + 32\sqrt{d}\beta^2/(\lambda \varepsilon^2))\right]$$

$$\quad + N_l \left[d \log(1 + 128\alpha M H^3 \sqrt{dK/\lambda}/\varepsilon) + d^2 \log(1 + 2048\alpha^2 \sqrt{d}\beta^2 M^2 H^4/(\lambda \varepsilon^2))\right]$$

$$\lesssim \frac{d^3 H}{\log \eta} \log\left(1 + \frac{K}{\lambda d}\right) \log\left(1 + \frac{\alpha \beta dH}{\lambda \varepsilon^2 \log \eta}\right).$$

where $\mathcal{V}$ denotes the class of $V$-functions. With this bound at hand, we use Lemma F.7 with $\varepsilon = \frac{dH}{K}$ and $\alpha = \text{poly}\left(T, 1/\delta, \log|\mathcal{A}|, d\right)$. Then it holds that

$$
|\phi(s,a)^T (\Lambda_h^k)^{-1} \sum_{i=1}^{k-1} \phi(s_h^i, a_h^i) \left[V_{h+1}^i(s_{h+1}^i) - P_h V_{h+1}^k(s_h^i, a_h^i)\right]|
$$

$$
\leq \left\|\sum_{i=1}^{k-1} \phi(s_h^i, a_h^i) \left[V_{h+1}^i(s_{h+1}^i) - P_h V_{h+1}^k(s_h^i, a_h^i)\right]\right\|_{(\Lambda_h^k)^{-1}} \|\phi(s,a)\|_{(\Lambda_h^k)^{-1}}
$$

$$
\lesssim c' \frac{d^{3/2} H^{3/2}}{\log\eta} \sqrt{\log\left(1 + \frac{K}{\lambda d}\right) \log\left(\frac{\beta dHK \log(|\mathcal{A}|)}{\delta\lambda\log\eta}\right)} \|\phi(s,a)\|_{(\Lambda_h^k)^{-1}}.
$$

On the other hand, by Cauchy-Schwarz inequality and Lemma F.4, it holds that

$$
|\lambda\phi(s,a)^T(\Lambda_h^k)^{-1} w_h^k| \leq \lambda \left\|w_h^k\right\|_{(\Lambda_h^k)^{-1}} \|\phi(s,a)\|_{(\Lambda_h^k)^{-1}} \leq 2H\sqrt{d}\,\|\phi(s,a)\|_{(\Lambda_h^k)^{-1}}
$$

where we used the fact that $(\Lambda_h^k)^{-1} \preceq \frac{1}{\lambda} I$. Combining the results above, it follows that

$$
|\phi(s,a)^T \hat{w}_h^k - r_h(s,a) - P_h V_{h+1}^k(s,a)|
$$

$$
\leq |\phi(s,a)^T (\Lambda_h^k)^{-1} \sum_{i=1}^{k-1} \phi(s_h^i, a_h^i) \left[V_{h+1}^i(s_{h+1}^i) - P_h V_{h+1}^k(s_h^i, a_h^i)\right]| + |\lambda\phi(s,a)^T(\Lambda_h^k)^{-1} w_h^k|
$$

$$
\leq c' \frac{d^{3/2} H^{3/2}}{\log\eta} \sqrt{\log\left(1 + \frac{K}{\lambda d}\right) \log\left(\frac{\beta dHK \log(|\mathcal{A}|)}{\delta\lambda\log\eta}\right)} \|\phi(s,a)\|_{(\Lambda_h^k)^{-1}}
$$

where $c'$ is some absolute constant. Now the proof is complete if the following inequality holds:

$$
c' \frac{d^{3/2} H^{3/2}}{\log\eta} \sqrt{\log\left(1 + \frac{K}{\lambda d}\right) \log\left(\frac{(C_l d^{3/2} H^{3/2}\chi_l) dHK \log(|\mathcal{A}|)}{\delta\lambda\log\eta}\right)} \leq \beta = C_l d^{3/2} H^{3/2}\chi_l
$$

Since $c'$ is independent of $C_l$, we can find an absolute constant $C_l$ satisfying this inequality. This completes the proof. $\qquad\square$

## C.3 PROOF OF LEMMA 3.3

*Proof of Lemma 3.3.* By Lemma C.1, with probability at least $1 - \delta/2$, for all $k \in [K], h \in [H]$, we have

$$
|\phi(s,a)^T \hat{w}_h^k - r_h(s,a) P_h V_{h+1}^k(s,a)| \leq \beta \|\phi(s,a)\|_{(\Lambda_h^k)^{-1}}.
$$

Therefore, for any $(s,a) \in \mathcal{S} \times \mathcal{A}$, it holds that

$$
\begin{aligned}
\xi_h^k(s,a) &= r_h(s,a) + P_h V_{h+1}^k(s,a) - Q_h^k(s,a) \\
&= r_h(s,a) + P_h V_{h+1}^{\bar{k}}(s,a) - Q_h^{\bar{k}}(s,a) \\
&= r_h(s,a) + P_h V_{h+1}^{\bar{k}}(s,a) - \left[\phi(s,a)^T \hat{w}_h^{\bar{k}} + \beta \|\phi(s,a)\|_{(\Lambda_h^{\bar{k}})^{-1}}\right]_{[0, H-h+1]} \\
&\leq r_h(s,a) + P_h V_{h+1}^{\bar{k}}(s,a) - \phi(s,a)^T \hat{w}_h^{\bar{k}} - \beta \|\phi(s,a)\|_{(\Lambda_h^{\bar{k}})^{-1}} \\
&\leq 0
\end{aligned}
$$

where the first equality uses the definition of $\bar{k}$, the first inequality uses the fact that $r_h(s,a) + P_h V_{h+1}^k(s,a) \in [0, H-h+1]$ for any $k \in [K], h \in [H]$ and any $(s,a) \in \mathcal{S} \times \mathcal{A}$. The other direction can be shown similarly:

$$
\begin{aligned}
-\xi_h^k(s,a) &= Q_h^k(s,a) - r_h(s,a) - P_h V_{h+1}^k(s,a) \\
&= Q_h^{\bar{k}}(s,a) - r_h(s,a) - P_h V_{h+1}^{\bar{k}}(s,a) \\
&= \left[\phi(s,a)^T \hat{w}_h^{\bar{k}} + \beta \|\phi(s,a)\|_{(\Lambda_h^{\bar{k}})^{-1}}\right]_{[0, H-h+1]} - r_h(s,a) - P_h V_{h+1}^{\bar{k}}(s,a) \\
&\leq \phi(s,a)^T \hat{w}_h^{\bar{k}} + \beta \|\phi(s,a)\|_{(\Lambda_h^{\bar{k}})^{-1}} - r_h(s,a) - P_h V_{h+1}^{\bar{k}}(s,a) \\
&\leq 2\beta \|\phi(s,a)\|_{(\Lambda_h^{\bar{k}})^{-1}}
\end{aligned}
$$

where the first inequality we again rely on the fact $r_h(s,a) + P_h V_{h+1}^k(s,a) \in [0, H-h+1]$ for any $k \in [K], h \in [H]$ and any $(s,a) \in \mathcal{S} \times \mathcal{A}$. □

## C.4  Proof of Theorem 3.1

We need the following lemma to bound the sum of martingale difference sequences that appears in the regret decomposition (Lemma F.3).

**Lemma C.2.** *It holds with probability at least $1 - \delta/2$ that*

$$\sum_{k=1}^{K}\sum_{h=1}^{H}\mathcal{M}_{1,h}^k + \sum_{k=1}^{K}\sum_{h=1}^{H}\mathcal{M}_{2,h}^k \leq 2\sqrt{2H^2 T \log(4/\delta)}$$

*where $\mathcal{M}_{1,h}^k$ and $\mathcal{M}_{2,h}^k$ are defined in Lemma F.3.*

*Proof.* Since $\mathcal{M}_{1,h}^k$ is adapted to $\mathbb{F}_{1,h+1}^k$ and $\mathbb{E}[\mathcal{M}_{1,h}^k \mid \mathbb{F}_{1,h}^k] = 0$, $\mathcal{M}_{1,h}^k$ is a martingale difference sequence bounded by $|\mathcal{D}_h^k| \leq 2H$. Hence, Azuma-Hoeffding inequality implies $\sum_{k=1}^{K}\sum_{h=1}^{H}\mathcal{M}_{1,h}^k \leq \sqrt{2H^2 T \log(4/\delta)}$ with probability at least $1 - \delta/4$. Similarly, $\mathcal{M}_{2,h}^k$ is adapted to $\mathbb{F}_{2,h+1}^k$. Thus $\mathbb{E}[\mathcal{M}_{2,h}^k \mid \mathcal{F}_{2,h}^k] = 0$ and $|\mathcal{M}_h^k| \leq 2H$ leads to $\sum_{k=1}^{K}\sum_{h=1}^{H}\mathcal{M}_{2,h}^k \leq \sqrt{2H^2 T \log(4/\delta)}$ with probability at least $1 - \delta/4$. Taking union bound completes the proof. □

Now we are ready to prove Theorem 3.1.

*Proof of Theorem 3.1.* By Lemma F.3, we have

$$\begin{aligned}
\mathrm{Regret}(K) &= \sum_{k=1}^{K} V_1^*(s_1^k) - V_1^{\pi^k}(s_1^k) \\
&= \sum_{k=1}^{K}\sum_{h=1}^{H}\mathbb{E}_{\pi^*}[\langle Q_h^k(s_h, \cdot), \pi_h^*(\cdot \mid s_h) - \pi_h^k(\cdot \mid s_h)\rangle \mid s_1 = s_1^k] \\
&\quad + \sum_{k=1}^{K}\sum_{h=1}^{H}\mathcal{M}_{1,h}^k + \sum_{k=1}^{K}\sum_{h=1}^{H}\mathcal{M}_{2,h}^k \\
&\quad + \sum_{k=1}^{K}\sum_{h=1}^{H}(\mathbb{E}_{\pi^*}[\xi_h^k(s_h, a_h) \mid s_1 = s_1^k] - \xi_h^k(s_h, a_h))
\end{aligned}$$

Lemma 3.4 with $\alpha = \Omega(\sqrt{K}\log(|\mathcal{A}|))$ gives

$$\sum_{k=1}^{K}\sum_{h=1}^{H}\mathbb{E}_{\pi^*}[\langle Q_h^k(s_h, \cdot), \pi_h^*(\cdot \mid s_h) - \pi_h^k(\cdot \mid s_h)\rangle \mid s_1 = s_1^k] \leq H\sqrt{K}.$$

By Lemma C.2, with probability at least $1 - \delta/2$, we have

$$\sum_{k=1}^{K}\sum_{h=1}^{H}\mathcal{M}_{1,h}^k + \sum_{k=1}^{K}\sum_{h=1}^{H}\mathcal{M}_{2,h}^k \leq 2\sqrt{2H^2 T \log(4/\delta)}.$$

On the other hand, Lemma 3.3 implies that, with probability at least $1 - \delta/2$, we have

$$\sum_{k=1}^{K}\sum_{h=1}^{H}(\mathbb{E}_{\pi^*}[\xi_h^k(s_h, a_h) \mid s_1 = s_1^k] - \xi_h^k(s_h, a_h))$$

$$\leq 0 + \sum_{k=1}^{K}\sum_{h=1}^{H} 2\beta \left\|\phi(s_h^k, a_h^k)\right\|_{(\Lambda_h^{\bar{k}})^{-1}} \leq \sum_{k=1}^{K}\sum_{h=1}^{H} 2\eta\beta \left\|\phi(s_h^k, a_h^k)\right\|_{(\Lambda_h^{k})^{-1}}$$

where the last inequality holds due to Lemma F.10 and the fact that $\det((\Lambda_h^{\bar{k}})^{-1}) < \eta \det((\Lambda_h^k)^{-1})$. Furthermore, it follows that

$$\sum_{k=1}^K \sum_{h=1}^H 2\eta\beta \left\|\phi(s_h^k, a_h^k)\right\|_{(\Lambda_h^k)^{-1}} \le 2\eta\beta \sum_{h=1}^H \sqrt{K} \sqrt{\sum_{k=1}^K \left\|\phi(s_h^k, a_h^k)\right\|_{(\Lambda_h^k)^{-1}}}$$

$$\le 2\eta\beta\sqrt{K} \sum_{h=1}^H \sqrt{2\log \frac{\det(\Lambda_h^K)}{\det(\Lambda_h^1)}} \le 2\eta\beta H\sqrt{K}\sqrt{2d\log\left(1 + \frac{K}{\lambda d}\right)}$$

where the first inequality uses the Cauchy-Schwartz inequality, the second inequality holds due to Lemma F.9, and the last inequality holds due to Lemma F.11. Combining the results above and taking a union bound, with probability at least $1 - \delta$, we have

$$Regret(K) \le H\sqrt{K} + 2\sqrt{2H^2 T \log(4/\delta)} + 2\eta\beta H\sqrt{K}\sqrt{2d\log\left(1 + \frac{K}{\lambda d}\right)}$$

$$\lesssim \frac{\eta d^2 H^2 \sqrt{T}}{\log \eta} \log\left(1 + \frac{K}{\lambda d}\right)\sqrt{\log\left(\frac{dT\log(|\mathcal{A}|)}{\delta\lambda\log\eta}\right)}$$

since we set $\beta = C_l d^{3/2} H^{3/2} \frac{1}{\log \eta}\sqrt{\log\left(1 + \frac{K}{\lambda d}\right)\log\left(\frac{dT\log|\mathcal{A}|}{\delta\lambda\log\eta}\right)}$.

$\square$

# D  ANALYSIS FOR THEOREM 4.1

In this section, we prove Theorem 4.1. We begin by defining the following filtrations.

$$\mathbb{G}^k := \sigma\left(\{(s_h^\tau, a_h^\tau, r_h^\tau)\}_{(\tau,h)\in[k]\times[H]} \cup \hat{\mathcal{Z}}_h^k\right)$$

$$\mathbb{G}_{1,h}^k := \sigma\left(\mathbb{G}^{k-1} \cup \hat{\mathcal{Z}}_h^k \cup \{(s_j^k, a_j^k, r_j^k)\}_{j\in[h]}\right)$$

$$\mathbb{G}_{2,h}^k := \sigma\left(\mathbb{G}^{k-1} \cup \hat{\mathcal{Z}}_h^k \cup \{(s_j^k, a_j^k, r_j^k)\}_{j\in[h-1]} \cup s_h^k\right)$$

where $\sigma(A)$ is the $\sigma$-algebra generated by $A$. For notational convenience, we use $\mathbb{E}_k$ to denote the conditional expectation with respect to $\mathbb{G}^k$, and we denote $\mathbb{P}(AB) := \mathbb{P}(A \cap B)$ for events $A, B$.

## D.1  PROOF OF LEMMA 4.2

We formally restate Lemma 4.2, which is established in Kong et al. (2021):

**Lemma D.1.** *For $\beta$ specified in Theorem 4.1, with probability at least $1 - \delta/8$, the following hold:*

1. *For all $(k, h) \in [K] \times [H]$, the number of distinct elements in $\hat{\mathcal{Z}}_h^K$ is bounded by*

$$N_g = C_{sw} \cdot \log(T\mathcal{N}(\mathcal{F}, \sqrt{\delta/64T^3}/\delta)) \cdot dim_E(\mathcal{F}, 1/T) \cdot \log^2 T.$$

   *Therefore, the number of policy switches in Algorithm 2 is bounded by $H \cdot N_g$.*

2. *For all $(k, h) \in [K] \times [H]$, $|\hat{\mathcal{Z}}_h^k| \le 64T^3/\delta$*

3. *For all $(k, h) \in [K] \times [H]$, it holds that*

$$\frac{\|f_1 - f_2\|_{\mathcal{Z}_h^k}^2}{10000} \le \min\{\|f_1 - f_2\|_{\hat{\mathcal{Z}}_h^k}^2, T(H+1)^2\} \le 10000\|f_1 - f_2\|_{\hat{\mathcal{Z}}_h^k}^2, \ \forall \|f_1 - f_2\|_{\mathcal{Z}_h^k}^2 > 100\beta$$

$$and \ \min\{\|f_1 - f_2\|_{\hat{\mathcal{Z}}_h^k}^2, T(H+1)^2\} \le 10000\beta, \ \forall \|f_1 - f_2\|_{\mathcal{Z}_h^k} \le 100\beta$$

We define sets and events required to prove Lemma D.1. For each $(k, h) \in [K] \times [H]$ and $\gamma \in [\beta, \infty)$, define

$$\underline{\mathcal{B}}_h^k(\gamma) := \{(f_1, f_2) \in \mathcal{F} \times \mathcal{F} : \|f_1 - f_2\|_{\mathcal{Z}_h^k}^2 \leq \gamma/100\}$$

$$\mathcal{B}_h^k(\gamma) := \{(f_1, f_2) \in \mathcal{F} \times \mathcal{F} : \min\{\|f_1 - f_2\|_{\hat{\mathcal{Z}}_h^k}^2, T(H+1)^2\} \leq \gamma\}$$

$$\overline{\mathcal{B}}_h^k(\gamma) := \{(f_1, f_2) \in \mathcal{F} \times \mathcal{F} : \|f_1 - f_2\|_{\mathcal{Z}_h^k}^2 \leq 100\gamma\}$$

Moreover, we define the following bonus functions with respect $\mathcal{Z}_h^k$.

$$\underline{b}_h^k(\cdot, \cdot) := \sup_{\|f_1 - f_2\|_{\mathcal{Z}_h^k}^2 \leq \beta/100} |f_1(\cdot, \cdot) - f_2(\cdot, \cdot)|$$

$$\overline{b}_h^k(\cdot, \cdot) := \sup_{\|f_1 - f_2\|_{\mathcal{Z}_h^k}^2 \leq 100\beta} |f_1(\cdot, \cdot) - f_2(\cdot, \cdot)|$$

For each $(k, h) \in [K] \times [H]$, we consider a good event $\mathcal{E}_h^k(\gamma)$ that

$$\underline{\mathcal{B}}_h^k(\gamma) \subseteq \mathcal{B}_h^k(\gamma) \subseteq \overline{\mathcal{B}}_h^k(\gamma)$$

and denote that

$$\mathcal{E}_h^k := \bigcap_{n=0}^{\infty} \mathcal{E}_h^k(100^n \gamma)$$

On the good event $\mathcal{E}_h^k$, we can show that $\hat{\mathcal{Z}}_h^k$ is a good approximation of $\mathcal{Z}_h^k$, i.e., $\|f_1 - f_2\|_{\hat{\mathcal{Z}}_h^k}$ is close to $\|f_1 - f_2\|_{\mathcal{Z}_h^k}$ up to some constant factor. Also, setting $\gamma = \beta$ directly implies

$$\underline{b}_h^k(\cdot, \cdot) \leq b_h^k(\cdot, \cdot) \leq \overline{b}_h^k(\cdot, \cdot)$$

on $\mathcal{E}_h^k$. The following lemma from Kong et al. (2021) formalizes the notion of good approximation.

**Lemma D.2.** *Conditioned on $\mathcal{E}_h^k$, it holds that*

$$\frac{\|f_1 - f_2\|_{\mathcal{Z}_h^k}^2}{10000} \leq \min\{\|f_1 - f_2\|_{\hat{\mathcal{Z}}_h^k}^2, T(H+1)^2\} \leq 10000 \|f_1 - f_2\|_{\mathcal{Z}_h^k}^2, \forall \|f_1 - f_2\|_{\mathcal{Z}_h^k}^2 > 100\beta$$

*and*

$$\min\{\|f_1 - f_2\|_{\hat{\mathcal{Z}}_h^k}^2, T(H+1)^2\} \leq 10000\beta, \ \forall \|f_1 - f_2\|_{\mathcal{Z}_h^k}^2 \leq 100\beta$$

*Proof.* If $\|f_1 - f_2\|_{\mathcal{Z}_h^k}^2 \leq 100\beta$, then $(f_1, f_2) \in \underline{\mathcal{B}}_h^k(10000\beta)$. Conditioned on $\mathcal{E}_h^k$, we have $(f_1, f_2) \in \mathcal{B}_h^k(10000\beta)$, which means $\min\{\|f_1 - f_2\|_{\hat{\mathcal{Z}}_h^k}^2, T(H+1)^2\} \leq 10000\beta$.

If $\|f_1 - f_2\|_{\mathcal{Z}_h^k}^2 > 100\beta$, assume $100^n \beta < \|f_1 - f_2\|_{\mathcal{Z}_h^k}^2 \leq 100^{n+1}\beta$ for some $n \in \mathbb{N} \cup \{0\}$. Then we have $(f_1, f_2) \notin \overline{\mathcal{B}}_h^k(100^{n-1}\beta)$, and $\mathcal{E}_h^k$ implies $(f_1, f_2) \notin \mathcal{B}_h^k(100^{n-1}\beta)$. Hence $\min\{\|f_1 - f_2\|_{\hat{\mathcal{Z}}_h^k}^2, T(H+1)^2\} \geq 100^{n-1}\beta \geq \frac{1}{10000}\|f_1 - f_2\|_{\mathcal{Z}_h^k}^2$. Similarly, we have $(f_1, f_2) \in \underline{\mathcal{B}}_h^k(100^{n+2}\beta)$, and $\mathcal{E}_h^k$ implies $(f_1, f_2) \in \mathcal{B}_h^k(100^{n+2}\beta)$. Thus it follows that $\min\{\|f_1 - f_2\|_{\hat{\mathcal{Z}}_h^k}^2, T(H+1)^2\} \leq 100^{n+2}\beta \leq 10000 \|f_1 - f_2\|_{\mathcal{Z}_h^k}^2$. $\qquad\square$

The following lemma from Kong et al. (2021) states that $\mathcal{E}_h^k$ happens with high probability.

**Lemma D.3.**

$$\mathbb{P}\left(\bigcap_{h=1}^{H} \bigcap_{k=1}^{K} \mathcal{E}_h^k\right) \geq 1 - \delta/32$$

Now we present some lemmas required to prove Lemma D.3. The following lemma states that the size of $\hat{\mathcal{Z}}_h^k$ is bounded with high probability. This is proved in Kong et al. (2021).

**Lemma D.4.** *With probability at least $1 - \delta/64T$, for all $(k, h) \in [K] \times [H]$, it holds that*

$$|\hat{\mathcal{Z}}_h^k| \leq 64T^3/\delta$$

*Proof.* For a fixed pair $(k, h) \in [K] \times [H]$, Markov's inequality implies

$$\mathbb{P}(|\hat{\mathcal{Z}}_h^k| > 64T^2|\mathcal{Z}_h^k|/\delta) \leq \mathbb{E}[|\hat{\mathcal{Z}}_h^k|]/(64T^2|\mathcal{Z}_h^k|/\delta) = \delta/64T^2$$

where we used the fact $\mathbb{E}[|\hat{\mathcal{Z}}_h^k|] = |\mathcal{Z}_h^k|$ which is the result of the sampling procedure. Taking a union bound completes the proof. $\square$

The following lemma is a key step for proving Lemma D.3, and this is established in Kong et al. (2021).

**Lemma D.5.** *For $\gamma \in [\beta, T(H + 1)^2]$ and any fixed $(k, h) \in [K] \times [H]$, it holds that*

$$\mathbb{P}\left(\left(\bigcap_{i=1}^{k-1} \mathcal{E}_h^i\right) \mathcal{E}_h^k(\gamma)^c\right) \leq \delta/32T^2$$

*Proof.* We use $\bar{\mathcal{Z}}_h^k$ to denote the dataset without rounding, i.e., we replace every element $\hat{z}$ with z in $\hat{\mathcal{Z}}_h^k$. Denote $\tilde{C} := C \log(T\mathcal{N}(\mathcal{F}, \sqrt{\delta/64T^3})/\delta)$ where $C$ is the parameter used in Algorithm 3.

Consider a fixed pair $(f_1, f_2) \in \mathcal{C}(\mathcal{F}, \sqrt{\delta/64T^3}) \times \mathcal{C}(\mathcal{F}, \sqrt{\delta/64T^3})$. For each $i \geq 2$, define

$$Z_i := \max\{\|f_1 - f_2\|_{\mathcal{Z}_h^i}^2, \min\{\|f_1 - f_2\|_{\hat{\mathcal{Z}}_h^{i-1}}^2, T(H + 1)^2\}\}$$

and

$$Y_i := \begin{cases} \frac{1}{p_{z_h^{i-1}}}(f_1(z_h^{i-1}) - f_2(z_h^{i-1}))^2 & z_h^{i-1} \text{ is added into } \bar{\mathcal{Z}}_h^i \text{ and } Z_i \leq 2000000\gamma \\ 0 & z_h^{i-1} \text{ is not added into } \bar{\mathcal{Z}}_h^i \text{ and } Z_i \leq 2000000\gamma \\ (f_1(z_h^{i-1}) - f_2(z_h^{i-1}))^2 & Z_i > 2000000\gamma \end{cases}$$

Note that $Y_i$ is adapted to $\mathbb{G}^i$ and $\mathbb{E}_{i-1}[Y_i] = (f_1(z_h^{i-1}) - f_2(z_h^{i-1}))^2$. In order to use Freedman's inequality (Lemma F.1), we need to bound $Y_i$ and its variance.

If $p_{z_h^{i-1}} = 1$ or $\min\{\|f_1 - f_2\|_{\hat{\mathcal{Z}}_h^{i-1}}^2, T(H + 1)^2\} > 2000000\gamma$, then $Y_i - \mathbb{E}_{i-1}[Y_i] = \text{Var}_{i-1}[Y_i - \mathbb{E}_{i-1}[Y_i]] = 0$. Otherwise, from the definition of $p_{z_h^{i-1}}$, we have

$$1 > C \cdot \text{sensitivity}_{\hat{\mathcal{Z}}_h^{i-1}, \mathcal{F}}(z_h^{i-1}) \cdot \log\left(T\mathcal{N}(\mathcal{F}, \sqrt{\delta/64T^3})/\delta\right) = \tilde{C} \cdot \text{sensitivity}_{\hat{\mathcal{Z}}_h^{i-1}, \mathcal{F}}(z_h^{i-1})$$

Note that we can pick sufficiently large absolute constant $C$, so that $\text{sensitivity}_{\hat{\mathcal{Z}}_h^{i-1}}(z_h^{i-1}) < 1$, which leads to

$$\text{sensitivity}_{\hat{\mathcal{Z}}_h^{i-1}}(z_h^{i-1}) = \min\left\{\sup_{\tilde{f}_1, \tilde{f}_2 \in \mathcal{F}} \frac{(\tilde{f}_1(z_h^{i-1}) - \tilde{f}_2(z_h^{i-1}))^2}{\min\left\{\left\|\tilde{f}_1 - \tilde{f}_2\right\|_{\hat{\mathcal{Z}}_h^{i-1}}^2, T(H + 1)^2\right\} + \beta}, 1\right\}$$

$$= \sup_{\tilde{f}_1, \tilde{f}_2 \in \mathcal{F}} \frac{(\tilde{f}_1(z_h^{i-1}) - \tilde{f}_2(z_h^{i-1}))^2}{\min\left\{\left\|\tilde{f}_1 - \tilde{f}_2\right\|_{\hat{\mathcal{Z}}_h^{i-1}}^2, T(H + 1)^2\right\} + \beta}.$$

In the case of $Y_i = \frac{1}{p_{z_h^{i-1}}}(f_1(z_h^{i-1}) - f_2(z_h^{i-1}))^2$, it follow that

$$
\begin{aligned}
|Y_i - \mathbb{E}_{i-1}[Y_i]| &\leq \frac{1}{p_{z_h^{i-1}}}(f_1(z_h^{i-1}) - f_2(z_h^{i-1}))^2 \\
&\leq \frac{(f_1(z_h^{i-1}) - f_2(z_h^{i-1}))^2}{\tilde{C} \cdot \text{sensitivity}_{\hat{\mathcal{Z}}_h^{i-1}}(z_h^{i-1})} \\
&\leq \frac{(f_1(z_h^{i-1}) - f_2(z_h^{i-1}))^2 (\min\{\left\|\tilde{f}_1 - \tilde{f}_2\right\|_{\hat{\mathcal{Z}}_h^{i-1}}^2, T(H+1)^2\} + \beta)}{\tilde{C}(f_1(z_h^{i-1}) - f_2(z_h^{i-1}))^2} \\
&\leq (2000000\gamma + \beta)/\tilde{C} \leq 3000000\gamma/\tilde{C}.
\end{aligned}
$$

If $Y_i = 0$, we have

$$
\begin{aligned}
|Y_i - \mathbb{E}_{i-1}[Y_i]| &= (f_1(z_h^{i-1}) - f_2(z_h^{i-1}))^2 \\
&\leq \frac{1}{p_{z_h^{i-1}}}(f_1(z_h^{i-1}) - f_2(z_h^{i-1}))^2 \leq 3000000\gamma/\tilde{C}
\end{aligned}
$$

In the last inequality, we apply the same steps derived above. Combining the two cases, we get

$$
|Y_i - \mathbb{E}_{i-1}[Y_i]| \leq 3000000\gamma/\tilde{C}.
$$

For the variance, we have

$$
\begin{aligned}
\text{Var}_{i-1}[Y_i - \mathbb{E}_{i-1}[Y_i]] &\leq \mathbb{E}_{i-1}[Y_i^2] \\
&= p_{z_h^{i-1}} \cdot \left(\frac{1}{p_{z_h^{i-1}}}(f_1(z_h^{i-1}) - f_2(z_h^{i-1}))^2\right)^2 + (1 - p_{z_h^{i-1}}) \cdot 0^2 \\
&= \frac{1}{p_{z_h^{i-1}}}(f_1(z_h^{i-1}) - f_2(z_h^{i-1}))^4 \\
&\leq (f_1(z_h^{i-1}) - f_2(z_h^{i-1}))^2 \cdot \frac{3000000\gamma}{\tilde{C}}
\end{aligned}
$$

where the last inequality holds due to the inequality we derived above. Let $\tilde{k} \leq k$ be the largest integer satisfying $Z_{\tilde{k}} < 2000000\gamma$. We know $\text{Var}_{i-1}[Y_i - \mathbb{E}_{i-1}[Y_i]] = 0$ for $i > \tilde{k}$. Therefore, we obtain

$$
\begin{aligned}
\sum_{i=2}^{k} \text{Var}_{i-1}[Y_i - \mathbb{E}_{i-1}[Y_i]] &= \sum_{i=2}^{\tilde{k}} \text{Var}_{i-1}[Y_i - \mathbb{E}_{i-1}[Y_i]] \\
&\leq \frac{3000000\gamma}{\tilde{C}} \sum_{i=2}^{\tilde{k}}(f_1(z_h^{i-1}) - f_2(z_h^{i-1}))^2 \\
&\leq \frac{3000000\gamma}{\tilde{C}} Z_{\tilde{k}} \leq (3000000\gamma)^2/\tilde{C}
\end{aligned}
$$

where the second last inequality uses the definition of $Z_i$. Applying Freedman's inequality(Lemma F.1) to the sequence $\{Y_i - \mathbb{E}_{i-1}[Y_i]\}$, we have that

$$
\begin{aligned}
\mathbb{P}\left(\left|\sum_{i=2}^{k}(Y_i - \mathbb{E}_{i-1}[Y_i])\right| \geq \gamma/100\right) &\leq 2\exp\left(-\frac{(\gamma/100)^2/2}{(3000000\gamma)^2/\tilde{C} + (3000000\gamma/\tilde{C})(\gamma/100)/3}\right) \\
&\leq 2\exp\left(-\frac{C\log(T\mathcal{N}(\mathcal{F}, \sqrt{\delta/64T^3})/\delta)}{20000 \cdot (3000000^2 + 10000)}\right) \\
&\leq (\delta/64T^2)/(\mathcal{N}(\mathcal{F}, \sqrt{\delta/64T^3}))^2
\end{aligned}
$$

for sufficiently large $C$. With a union bound over $\mathcal{C}(\mathcal{F}, \sqrt{\delta/64T^3}) \times \mathcal{C}(\mathcal{F}, \sqrt{\delta/64T^3})$, we know that with probability at least $1 - \delta/64T^2$, for any $(f_1, f_2) \in \mathcal{C}(\mathcal{F}, \sqrt{\delta/64T^3}) \times \mathcal{C}(\mathcal{F}, \sqrt{\delta/64T^3})$, the corresponding $\{Y_i\}$ satisfies

$$\left| \sum_{i=2}^{k} (Y_i - \mathbb{E}_{i-1}[Y_i]) \right| \leq \gamma/100$$

Now we condition on the above event (say $\mathcal{E}_1$) and the event defined in Lemma D.4 (say $\mathcal{E}_2$) for the rest of the proof. Since We know that $\mathbb{P}((\mathcal{E}_1\mathcal{E}_2)^c) \leq \delta/32T^2$ and

$$\mathbb{P}\left( \left( \bigcap_{i=1}^{k-1} \mathcal{E}_h^i \right) \mathcal{E}_h^k(\gamma)^c \right) \leq \mathbb{P}\left( \left( \bigcap_{i=1}^{k-1} \mathcal{E}_h^i \right) \mathcal{E}_h^k(\gamma)^c \mid \mathcal{E}_1\mathcal{E}_2 \right) \cdot \mathbb{P}(\mathcal{E}_1\mathcal{E}_2) + \mathbb{P}((\mathcal{E}_1\mathcal{E}_2)^c),$$

The proof is complete if we show that

$$\mathbb{P}\left( \left( \bigcap_{i=1}^{k-1} \mathcal{E}_h^i \right) \mathcal{E}_h^k(\gamma)^c \mid \mathcal{E}_1\mathcal{E}_2 \right) = 0.$$

Therefore, we condition on the event $\bigcap_{i=1}^{k-1} \mathcal{E}_h^i$, then show that $\underline{\mathcal{B}}_h^k(\gamma) \subseteq \mathcal{B}_h^k(\gamma) \subseteq \overline{\mathcal{B}}_h^k(\gamma)$ (i.e. $\mathcal{E}_h^k(\gamma)$) holds almost surely.

**Part 1:** $(\underline{\mathcal{B}}_h^k(\gamma) \subseteq \mathcal{B}_h^k(\gamma))$ Consider any pair $f_1, f_2 \in \mathcal{F}$ with $\|f_1 - f_2\|_{\mathcal{Z}_h^k}^2 \leq \gamma/100$. From the definition, we know that there exist $(\hat{f}_1, \hat{f}_2) \in \mathcal{C}(\mathcal{F}, \sqrt{\delta/64T^3}) \times \mathcal{C}(\mathcal{F}, \sqrt{\delta/64T^3})$ such that $\left\| \hat{f}_1 - f_1 \right\|_\infty, \left\| \hat{f}_2 - f_2 \right\|_\infty \leq \sqrt{\delta/64T^3}$. Then we have that

$$\left\| \hat{f}_1 - \hat{f}_2 \right\|_{\mathcal{Z}_h^k}^2 \leq (\|f_1 - f_2\|_{\mathcal{Z}_h^k} + \left\| \hat{f}_1 - f_1 \right\|_{\mathcal{Z}_h^k} + \left\| \hat{f}_2 - f_2 \right\|_{\mathcal{Z}_h^k})^2$$

$$\leq (\|f_1 - f_2\|_{\mathcal{Z}_h^k} + 2\sqrt{|\mathcal{Z}_h^k|}\sqrt{\delta/64T^3})^2$$

$$\leq (\|f_1 - f_2\|_{\mathcal{Z}_h^k} + 2\sqrt{\delta/64T})^2 \leq \gamma/50.$$

Consider the $\{Y_i\}$ corresponding to $\hat{f}_1$ and $\hat{f}_2$. Since $\left\| \hat{f}_1 - \hat{f}_2 \right\|_{\mathcal{Z}_h^{k-1}}^2 \leq \left\| \hat{f}_1 - \hat{f}_2 \right\|_{\mathcal{Z}_h^k}^2 \leq \gamma/100$, From the definition of $\mathcal{E}_h^{k-1}$ we know that $\min\{\left\| \hat{f}_1 - \hat{f}_2 \right\|_{\hat{\mathcal{Z}}_h^{k-1}}, T(H+1)^2\} \leq 2\gamma \leq 10000\gamma$. Then from the definition of $\{Y_i\}$, we have

$$\left\| \hat{f}_1 - \hat{f}_2 \right\|_{\bar{\mathcal{Z}}_h^k}^2 = \sum_{i=2}^{k} Y_i \leq \sum_{i=2}^{k} \mathbb{E}_{i-1}[Y_i] + \gamma/100$$

$$= \left\| \hat{f}_1 - \hat{f}_2 \right\|_{\mathcal{Z}_h^k}^2 + \gamma/100 \leq 3\gamma/100$$

where the first inequality used the concentration we derived above. Further, we can bound $\|f_1 - f_2\|_{\bar{\mathcal{Z}}_h^k}$ by

$$\|f_1 - f_2\|_{\bar{\mathcal{Z}}_h^k} \leq (\left\| \hat{f}_1 - \hat{f}_2 \right\|_{\bar{\mathcal{Z}}_h^k} + \left\| \hat{f}_1 - f_1 \right\|_{\bar{\mathcal{Z}}_h^k} + \left\| \hat{f}_2 - f_2 \right\|_{\bar{\mathcal{Z}}_h^k})^2$$

$$\leq (\left\| \hat{f}_1 - \hat{f}_2 \right\|_{\bar{\mathcal{Z}}_h^k} + 2\sqrt{|\bar{\mathcal{Z}}_h^k|}\sqrt{\delta/64T^3})^2$$

$$\leq (\left\| \hat{f}_1 - \hat{f}_2 \right\|_{\bar{\mathcal{Z}}_h^k} + 2)^2 \leq \gamma/25$$

where the second last inequality holds due to Lemma D.4, and the last inequality holds due to the fact $\gamma \geq \beta$. Finally, $\|f_1 - f_2\|_{\hat{\mathcal{Z}}_h^k}$ can be bounded:

$$\|f_1 = f_2\|_{\hat{\mathcal{Z}}_h^k}^2 \leq (\|f_1 - f_2\|_{\bar{\mathcal{Z}}_h^k} + \sqrt{|\hat{\mathcal{Z}}_h^k|}/16\sqrt{64T^3/\delta})^2$$

$$\leq (\|f_1 - f_2\|_{\bar{\mathcal{Z}}_h^k} + 1/16)^2 \leq \gamma$$

where the first inequality holds due to the rounding property of $\hat{\mathcal{Z}}_h^k$ and the second inequality holds due to Lemma D.4. Since it holds that any $f_1, f_2 \in \mathcal{F}$ with $\|f_1 - f_2\|_{\mathcal{Z}_h^k}^2 \leq \gamma/100$ satisfies $\|f_1 - f_2\|_{\hat{\mathcal{Z}}_h^k}^2 \leq \gamma$, we have $\underline{\mathcal{B}}_h^k(\gamma) \subseteq \mathcal{B}_h^k(\gamma)$.

**Part 2:** $(\mathcal{B}_h^k(\gamma) \subseteq \overline{\mathcal{B}}_h^k(\gamma))$   Consider any $f_1, f_2 \in \mathcal{F}$ with $\|f_1 - f_2\|_{\mathcal{Z}_h^k}^2 \geq 100\gamma$. From the definition, we know that there exist $(\hat{f}_1, \hat{f}_2) \in \mathcal{C}(\mathcal{F}, \sqrt{\delta/64T^3}) \times \mathcal{C}(\mathcal{F}, \sqrt{\delta/64T^3})$ such that $\left\|\hat{f}_1 - f_1\right\|_\infty, \left\|\hat{f}_2 - f_2\right\|_\infty \leq \sqrt{\delta/64T^3}$. Then we have that

$$
\begin{aligned}
\left\|\hat{f}_1 - \hat{f}_2\right\|_{\mathcal{Z}_h^k} &\leq (\|f_1 - f_2\|_{\mathcal{Z}_h^k} - \left\|\hat{f}_1 - f_1\right\|_{\mathcal{Z}_h^k} - \left\|\hat{f}_2 - f_2\right\|_{\mathcal{Z}_h^k})^2 \\
&\leq (\|f_1 - f_2\|_{\mathcal{Z}_h^k} - 2\sqrt{|\mathcal{Z}_h^k|}\sqrt{\delta/64T^3})^2 \\
&\leq (\|f_1 - f_2\|_{\mathcal{Z}_h^k} - 2\sqrt{\delta/64})^2 \\
&\leq 50\gamma.
\end{aligned}
$$

Now consider the $\{Y_i\}$ corresponding to $\hat{f}_1$ and $\hat{f}_2$.

**Case 1:** $(\left\|\hat{f}_1 - \hat{f}_2\right\|_{\mathcal{Z}_h^k} \leq 2000000\gamma)$   From the definition of $\{Y_i\}$, it holds that

$$
\begin{aligned}
\left\|\hat{f}_1 - \hat{f}_2\right\|_{\bar{\mathcal{Z}}_h^k} = \sum_{i=2}^k Y_i &\geq \sum_{i=2}^k \mathbb{E}_{i-1}[Y_i] - \gamma/100 \\
&= \left\|\hat{f}_1 - \hat{f}_2\right\|_{\mathcal{Z}_h^k}^2 - \gamma/100 > 49\gamma
\end{aligned}
$$

where the first inequality holds due to the concentration we derived above. Then we have that

$$
\begin{aligned}
\left\|\hat{f}_1 - \hat{f}_2\right\|_{\hat{\mathcal{Z}}_h^k} &\geq (\left\|\hat{f}_1 - \hat{f}_2\right\|_{\bar{\mathcal{Z}}_h^k} - \sqrt{|\hat{\mathcal{Z}}_h^k|}/16\sqrt{64T^3/\delta})^2 \\
&\geq (\left\|\hat{f}_1 - \hat{f}_2\right\|_{\bar{\mathcal{Z}}_h^k} - 1/16)^2 > 40\gamma
\end{aligned}
$$

where the second last inequality uses Lemma D.4.

**Case 2:** $(\left\|\hat{f}_1 - \hat{f}_2\right\|_{\mathcal{Z}_h^{k-1}} > 10000\gamma)$   Since we conditioned on $\mathcal{E}_h^{k-1}$, we have that

$$
\left\|\hat{f}_1 - \hat{f}_2\right\|_{\hat{\mathcal{Z}}_h^k} > 100\gamma > 40\gamma
$$

**Case 3:** $(\left\|\hat{f}_1 - \hat{f}_2\right\|_{\mathcal{Z}_h^k} > 2000000\gamma$ **and** $\left\|\hat{f}_1 - \hat{f}_2\right\|_{\mathcal{Z}_h^{k-1}} \leq 10000\gamma)$   It is clear that $(\hat{f}_1(z_h^{k-1}) - \hat{f}_2(z_h^{k-1}))^2 > 1990000\gamma$. Since $\left\|\hat{f}_1 - \hat{f}_2\right\|_{\mathcal{Z}_h^{k-1}} \leq 10000\gamma$, $\mathcal{E}_h^{k-1}$ guarantees that $\left\|\hat{f}_1 - \hat{f}_2\right\|_{\hat{\mathcal{Z}}_h^{k-1}} \leq 1000000\gamma$. Thus, from the definition of the sensitivity, sensitivity$_{\hat{\mathcal{Z}}_h^{k-1}}(z_h^{k-1}) = 1$ and $z_h^{k-1}$ is added to $\hat{\mathcal{Z}}_h^k$ almost surely. Hence, we have

$$
\left\|\hat{f}_1 - \hat{f}_2\right\|_{\hat{\mathcal{Z}}_h^k} \geq (\hat{f}_1(z_h^{k-1}) - \hat{f}_2(z_h^{k-1}))^2 > 40\gamma
$$

Combining the three cases, we conclude that $\left\|\hat{f}_1 - \hat{f}_2\right\|_{\hat{\mathcal{Z}}_h^k} > 40\gamma$. Finally we bound $\|f_1 - f_2\|_{\hat{\mathcal{Z}}_h^k}$:

$$
\begin{aligned}
\|f_1 - f_2\|_{\hat{\mathcal{Z}}_h^k} &\geq (\|f_1 - f_2\|_{\hat{\mathcal{Z}}_h^k} - 2\sqrt{|\hat{\mathcal{Z}}_h^k|}\sqrt{\delta/64T^3})^2 \\
&\geq (\|f_1 - f_2\|_{\hat{\mathcal{Z}}_h^k} - 2)^2 \geq \gamma
\end{aligned}
$$

where the second last inequality holds due to Lemma D.4. Since it holds that any $f_1, f_2 \in \mathcal{F}$ with $\|f_1 - f_2\|_{\mathcal{Z}_h^k}^2 > 100\gamma$ satisfies $\|f_1 - f_2\|_{\hat{\mathcal{Z}}_h^k}^2 > \gamma$, we have $\mathcal{B}_h^k(\gamma) \subseteq \overline{\mathcal{B}}_h^k(\gamma)$. $\qquad\square$

*Proof of Lemma D.3.* For any fixed $(k, h) \in [K] \times [H]$, we have

$$
\begin{aligned}
&\mathbb{P}(\mathcal{E}_h^1 \mathcal{E}_h^2 \ldots \mathcal{E}_h^{k-1}) - \mathbb{P}(\mathcal{E}_h^1 \mathcal{E}_h^2 \ldots \mathcal{E}_h^k) \\
&= \mathbb{P}\left(\mathcal{E}_h^1 \mathcal{E}_h^2 \ldots \mathcal{E}_h^{k-1} (\mathcal{E}_h^k)^c\right) \\
&= \mathbb{P}\left(\mathcal{E}_h^1 \mathcal{E}_h^2 \ldots \mathcal{E}_h^{k-1} \left(\bigcap_{n=0}^{\infty} \mathcal{E}_h^k(100^n \beta)\right)^c\right) \\
&= \mathbb{P}\left(\mathcal{E}_h^1 \mathcal{E}_h^2 \ldots \mathcal{E}_h^{k-1} \bigcup_{n=0}^{\infty} \mathcal{E}_h^k(100^n \beta)^c\right) \\
&\leq \sum_{n=0}^{\infty} \mathbb{P}\left(\mathcal{E}_h^1 \mathcal{E}_h^2 \ldots \mathcal{E}_h^{k-1} (\mathcal{E}_h^k(100^n \beta))^c\right) \\
&= \sum_{n \geq 0, 100^n \beta \leq T(H+1)^2} \mathbb{P}\left(\mathcal{E}_h^1 \mathcal{E}_h^2 \ldots \mathcal{E}_h^{k-1} (\mathcal{E}_h^k(100^n \beta))^c\right)
\end{aligned}
$$

where the last inequality is due to the fact that $\mathcal{E}_h^k(100^n \beta)$ always holds for $100^n \beta > T(H+1)^2$. Applying Lemma D.5, it holds that

$$
\mathbb{P}(\mathcal{E}_h^1 \mathcal{E}_h^2 \ldots \mathcal{E}_h^{k-1}) - \mathbb{P}(\mathcal{E}_h^1 \mathcal{E}_h^2 \ldots \mathcal{E}_h^k) \leq (\delta/32T^2) \cdot (\log(T(H+1)^2/\beta) + 2) \leq \delta/32T
$$

for sufficiently large $C_l$ (the absolute constant of $\beta$). Hence for all $h \in [H]$, we have

$$
\begin{aligned}
\mathbb{P}\left(\bigcap_{k=1}^{K} \mathcal{E}_h^k\right) &= 1 - \sum_{k=1}^{K} \left(\mathbb{P}(\mathcal{E}_h^1 \mathcal{E}_h^2 \ldots \mathcal{E}_h^{k-1}) - \mathbb{P}(\mathcal{E}_h^1 \mathcal{E}_h^2 \ldots \mathcal{E}_h^k)\right) \\
&\geq 1 - K \cdot (\delta/32T) = 1 - \delta/32H.
\end{aligned}
$$

Taking a union bound for all $h \in [H]$ completes the proof. $\qquad \square$

The following lemma bound the sum of online sensitivity scores with respect to $\mathcal{Z}_h^k$. This is proved in Kong et al. (2021).

**Lemma D.6.** *For all $h \in [H]$, it holds that*

$$
\sum_{k=1}^{K-1} sensitivity_{\mathcal{Z}_h^k, \mathcal{F}}(z_h^k) \leq 13 dim_E\left(\mathcal{F}, 1/T\right) \log(T(H+1)^2) \log T
$$

*Proof.* Since $|\mathcal{Z}_h^k| = k - 1 \leq T$, $\|f_1 - f_2\|_{\mathcal{Z}_h^k} \leq T(H+1)^2$ for all $f_1, f_2 \in \mathcal{F}$. Thus we have that

$$
\begin{aligned}
sensitivity_{\mathcal{Z}_h^k, \mathcal{F}}(z_h^k) &= \min\left\{\sup_{f_1, f_2 \in \mathcal{F}} \frac{(f_1(z_h^k) - f_2(z_h^k))^2}{\min\{\|f_1 - f_2\|_{\mathcal{Z}_h^k}^2, T(H+1)^2\} + \beta}, 1\right\} \\
&\leq \min\left\{\sup_{f_1, f_2 \in \mathcal{F}} \frac{(f_1(z_h^k) - f_2(z_h^k))^2}{\|f_1 - f_2\|_{\mathcal{Z}_h^k}^2 + 1}, 1\right\}.
\end{aligned}
$$

For each $k \in [K-1]$, let $f_1, f_2 \in \mathcal{F}$ be an arbitrary pair of functions such that

$$
\frac{(f_1(z_h^k) - f_2(z_h^k))^2}{\|f_1 - f_2\|_{\mathcal{Z}_h^k}^2 + 1}
$$

is maximized, and define $L(z_h^k) := (f_1(z_h^k) - f_2(z_h^k))^2$ for such $f_1, f_2$. Note that $0 \leq L(z_h^k) \leq (H+1)^2$. Let $\mathcal{Z}_h^K = \bigcup_{\alpha=0}^{\lfloor \log(T(H+1)^2)\rfloor - 1} Z^\alpha \cup Z^\infty$ be a dyadic decomposition with respect to $L(\cdot)$, where for each $0 \leq \alpha \leq \lfloor \log(T(H+1)^2)\rfloor - 1$, define

$$
Z^\alpha := \{z_h^k \in \mathcal{Z}_h^K : L(z_h^k) \in ((H+1)^2 \cdot 2^{-\alpha-1}, (H+1)^2 \cdot 2^{-\alpha}]\}
$$

and

$$
Z^\infty := \{z_h^k \in \mathcal{Z}_h^K : L(z_h^k) \leq (H+1)^2 \cdot 2^{-\lfloor \log(T(H+1)^2)\rfloor}\}.
$$

For any $z_h^k \in Z^\infty$, $\text{sensitivity}_{\mathcal{Z}_h^k, \mathcal{F}}(z_h^k) \leq (H+1)^2 \cdot 2^{-\lfloor \log(T(H+1)^2) \rfloor} \leq 2/T$ and we have

$$\sum_{z_h^k \in Z^\infty} \text{sensitivity}_{\mathcal{Z}_h^k, \mathcal{F}}(z_h^k) \leq |Z^\infty| \cdot 2/T \leq 2.$$

Now we bound $\sum_{z_h^k \in Z^\alpha} \text{sensitivity}_{\mathcal{Z}_h^k, \mathcal{F}}(z_h^k)$ for each $0 \leq \alpha \leq \lfloor \log(T(H+1)^2) \rfloor - 1$ separately. For each $\alpha$, let

$$N_\alpha := |Z^\alpha| / \dim_E \left( \mathcal{F}, (H+1)^2 \cdot 2^{-\alpha-1} \right)$$

and decompose $Z^\alpha$ into $N_\alpha + 1$ disjoint subsets, $Z^\alpha = \bigcup_{j=1}^{N_\alpha + 1} Z_j^\alpha$, using the following procedure:

Initialize $Z_j^\alpha = \{\}$ for all j and consider each $z_h^k \in Z^\alpha$ sequentially. For each $z_h^k \in Z^\alpha$, find the smallest $1 \leq j \leq N_\alpha$ such that $z_h^k$ is $(H+1)^2 \cdot 2^{-\alpha-1}$-independent of $Z_j^\alpha$ with respect to $\mathcal{F}$. If there is no $Z_j^\alpha$, we set $j = N_\alpha + 1$. We use $j(z_h^k)$ to denote such j, then add $z_h^k$ into $Z_{j(z_h^k)}^\alpha$.

Following the procedure, $z_h^k$ is $(H+1)^2 \cdot 2^{-\alpha-1}$-dependent of $Z_1^\alpha, \ldots, Z_{j(z_h^k)-1}^\alpha$. Recall that for each $z_h^k \in Z^\alpha$, we selected $f_1, f_2 \in \mathcal{F}$ such that

$$\frac{(f_1(z_h^k) - f_2(z_h^k))^2}{\|f_1 - f_2\|_{\mathcal{Z}_h^k}^2 + 1}$$

is maximized. Since $z_h^k \in Z^\alpha$, we have $(f_1(z_h^k) - f_2(z_h^k))^2 \leq (H+1)^2 \cdot 2^{-\alpha}$. Moreover, since $z_h^k$ is $(H+1)^2 \cdot 2^{-\alpha-1}$-dependent of $Z_1^\alpha, \ldots, Z_{j(z_h^k)-1}^\alpha$, it holds that

$$\|f_1 - f_2\|_{Z_t^\alpha} \geq (H+1)^2 \cdot 2^{-\alpha-1}, \ 1 \leq t \leq j(z_h^k).$$

Therefore, the online sensitivity score can be bounded by

$$\text{sensitivity}_{\mathcal{Z}_h^k, \mathcal{F}}(z_h^k) \leq \frac{(f_1(z_h^k) - f_2(z_h^k))^2}{\|f_1 - f_2\|_{\mathcal{Z}_h^k}^2 + 1} \leq \frac{(H+1)^2 \cdot 2^{-\alpha}}{\|f_1 - f_2\|_{\mathcal{Z}_h^k}^2}$$

$$\leq \frac{(H+1)^2 \cdot 2^{-\alpha}}{\sum_{t=1}^{j(z_h^k)-1} \|f_1 - f_2\|_{Z_t^\alpha}} \leq \frac{2}{j(z_h^k) - 1}.$$

By the definition of online sensitivity score, we have $\text{sensitivity}_{\mathcal{Z}_h^k, \mathcal{F}}(z_h^k) \leq 1$. Thus we obtain

$$\text{sensitivity}_{\mathcal{Z}_h^k, \mathcal{F}}(z_h^k) \leq \min \left\{ \frac{2}{j(z_h^k) - 1}, 1 \right\} \leq \frac{4}{j(z_h^k)}.$$

Moreover, by the definition of the eluder dimension, $|Z_j^\alpha| \leq \dim_E \left( \mathcal{F}, (H+1)^2 \cdot 2^{-\alpha-1} \right)$ for $1 \leq j \leq N_\alpha$. Therefore, it holds that

$$\sum_{z_h^k \in Z^\alpha} \text{sensitivity}_{\mathcal{Z}_h^k, \mathcal{F}}(z_h^k)$$

$$\leq \sum_{1 \leq j \leq N_\alpha} |Z_j^\alpha| \cdot 4/j + \sum_{z \in Z_{N_\alpha+1}^\alpha} 4/N_\alpha$$

$$\leq 4\dim_E \left( \mathcal{F}, (H+1)^2 \cdot 2^{-\alpha-1} \right) (\ln(N_\alpha) + 1) + |Z_\alpha| \cdot \frac{4\dim_E \left( \mathcal{F}, (H+1)^2 \cdot 2^{-\alpha-1} \right)}{|Z_\alpha|}$$

$$\leq 12\dim_E \left( \mathcal{F}, (H+1)^2 \cdot 2^{-\alpha-1} \right) \log T$$

where the second inequality holds due to the fact that $\sum_{i=1}^{n} 1/i \leq \ln n + 1$. Finally, combining the inequalities above and using the monotonicity of the eluder dimension, it follows that

$$
\sum_{k=1}^{K-1} \text{sensitivity}_{\mathcal{Z}_h^k, \mathcal{F}}(z_h^k)
$$

$$
\leq \sum_{\alpha=0}^{\lfloor \log(T(H+1)^2) \rfloor - 1} \sum_{z_h^k \in Z^\alpha} \text{sensitivity}_{\mathcal{Z}_h^k, \mathcal{F}}(z_h^k) + \sum_{z_h^k \in Z^\infty} \text{sensitivity}_{\mathcal{Z}_h^k, \mathcal{F}}(z_h^k)
$$

$$
\leq \sum_{\alpha=0}^{\lfloor \log(T(H+1)^2) \rfloor - 1} 12 \text{dim}_E \left( \mathcal{F}, (H+1)^2 \cdot 2^{-\alpha-1} \right) \log T + 2
$$

$$
\leq 12 \text{dim}_E \left( \mathcal{F}, 1/T \right) \log T \lfloor \log(T(H+1)^2) \rfloor + 2
$$

$$
\leq 13 \text{dim}_E \left( \mathcal{F}, 1/T \right) \log T \log(T(H+1)^2),
$$

which completes the proof. $\qquad\square$

*Proof of Lemma 4.2.* Note that in Algorithm 3, the sampling probability is bounded by

$$
p_z \lesssim \text{sensitivity}_{Z, \mathcal{F}}(z) \cdot \log(T\mathcal{N}(\mathcal{F}, \sqrt{\delta/64T^3})/\delta).
$$

On the other hand, we can bound $\text{sensitivity}_{\hat{\mathcal{Z}}_h^k, \mathcal{F}}(z_h^k)$ using Lemma D.2. Conditioned on $\bigcap_{h=1}^{H} \bigcap_{k=1}^{K} \mathcal{E}_h^k$, we have that

$$
\text{sensitivity}_{\hat{\mathcal{Z}}_h^k, \mathcal{F}}(z_h^k) = \min \left\{ \sup_{f_1, f_2 \in \mathcal{F}} \frac{(f_1(z_h^k) - f_2(z_h^k))^2}{\min\{\|f_1 - f_2\|_{\hat{\mathcal{Z}}_h^k}^2, T(H+1)^2\} + \beta}, 1 \right\}
$$

$$
\leq \min \left\{ \sup_{f_1, f_2 \in \mathcal{F}} \frac{(f_1(z_h^k) - f_2(z_h^k))^2}{\min\{\|f_1 - f_2\|_{\hat{\mathcal{Z}}_h^k}^2 / 10000, T(H+1)^2\} + \beta}, 1 \right\}
$$

$$
\leq 10000 \min \left\{ \sup_{f_1, f_2 \in \mathcal{F}} \frac{(f_1(z_h^k) - f_2(z_h^k))^2}{\min\{\|f_1 - f_2\|_{\mathcal{Z}_h^k}^2, T(H+1)^2\} + \beta}, 1 \right\}
$$

$$
= 10000 \cdot \text{sensitivity}_{\mathcal{Z}_h^k, \mathcal{F}}(z_h^k)
$$

Therefore, it follows that

$$
\sum_{k=1}^{K-1} \mathbf{1}\{\mathcal{E}_h^k\} \cdot p_{z_h^k} \lesssim \sum_{k=1}^{K-1} \mathbf{1}\{\mathcal{E}_h^k\} \cdot \text{sensitivity}_{\hat{\mathcal{Z}}_h^k, \mathcal{F}}(z_h^k) \cdot \log(T\mathcal{N}(\mathcal{F}, \sqrt{\delta/64T^3})/\delta)
$$

$$
\lesssim \sum_{k=1}^{K-1} \text{sensitivity}_{\mathcal{Z}_h^k, \mathcal{F}}(z_h^k) \cdot \log(T\mathcal{N}(\mathcal{F}, \sqrt{\delta/64T^3})/\delta)
$$

$$
\lesssim \log(T\mathcal{N}(\mathcal{F}, \sqrt{\delta/64T^3})/\delta) \text{dim}_E \left( \mathcal{F}, 1/T \right) \log^2 T
$$

for sufficiently large $T$. By adjusting the constant $C_{sw}$ in the statement of Lemma 4.2, we have that

$$
\sum_{k=1}^{K-1} \mathbf{1}\{\mathcal{E}_h^k\} \cdot p_{z_h^k} \leq N_g/3.
$$

For $2 \leq k \leq K$, let $X_h^k$ be a random variable defined as:

$$
X_h^k = \begin{cases} \mathbf{1}\{\mathcal{E}_h^{k-1}\} & \hat{z}_h^{k-1} \text{ is added into } \hat{\mathcal{Z}}_h^k \\ 0 & \text{otherwise} \end{cases}
$$

Then $X_h^k$ is adapted to $\mathbb{G}^k$. Simple calculation gives $\mathbb{E}_{k-1}[X_h^k] = p_{z_h^{k-1}} \cdot \mathbf{1}\{\mathcal{E}_h^{k-1}\}$ and $\mathbb{E}_{k-1}[(X_h^k - \mathbb{E}_{k-1}[X_h^k])^2] = p_{z_h^{k-1}}(1 - p_{z_h^{k-1}}) \cdot \mathbf{1}\{\mathcal{E}_h^{k-1}\}$. Since $X_h^k - \mathbb{E}_{k-1}[X_h^k]$ is a martingale difference

sequence satisfying

$$|X_h^k - \mathbb{E}_{k-1}[X_h^k]| \le 1,$$

$$\sum_{k=2}^{K} \mathbb{E}_{k-1}[(X_h^k - \mathbb{E}_{k-1}[X_h^k])^2] = \sum_{k=2}^{K} p_{z_h^{k-1}}(1 - p_{z_h^{k-1}}) \cdot \mathbf{1}\{\mathcal{E}_h^{k-1}\} \le \sum_{k=2}^{K} p_{z_h^{k-1}} \cdot \mathbf{1}\{\mathcal{E}_h^{k-1}\} \le N_g/3,$$

$$\sum_{k=2}^{K} \mathbb{E}_{k-1}[X_h^k] = p_{z_h^{k-1}} = \sum_{k=2}^{K} p_{z_h^{k-1}} \cdot \mathbf{1}\{\mathcal{E}_h^{k-1}\} \le N_g/3$$

By Freedman's inequality(Lemma F.1, it holds that

$$\mathbb{P}\left(\sum_{k=2}^{K} X_h^k \ge N_g\right)$$

$$\le \mathbb{P}\left(|\sum_{k=2}^{K}(X_h^k - \mathbb{E}_{k-1}[X_h^k])| \ge 2N_g/3\right)$$

$$\le 2\exp\left(-\frac{(2N_g/3)^2/2}{N_g/3 + 2N_g/9}\right)$$

$$\le 2\exp\left(-\frac{2}{5}C_{sw}\log(T\mathcal{N}(\mathcal{F}, \sqrt{\delta/64T^3}/\delta)) \cdot \dim_E(\mathcal{F}, 1/T) \cdot \log^2 T\right)$$

$$\le \delta/32T$$

where the first inequality uses the triangle inequality. Taking a union bound for all $h \in [H]$, with probability at least $1 - \delta/32$, it holds that

$$\sum_{k=2}^{K} X_h^k \le N_g, \ \forall h \in [H].$$

Since we've conditioned on $\bigcap_{h=1}^{H}\bigcap_{k=1}^{K} \mathcal{E}_h^k$, $\sum_{k=2}^{K} X_h^k$ is the number of distinct elements in $\hat{\mathcal{Z}}_h^K$. According to Line 7 of Algorithm 2, the number of policy switches is less than or equal to $\sum_{h=1}^{H} |\hat{\mathcal{Z}}_h^K|$. Hence we conclude that the number of policy switches is at most $H \cdot N_g$ times. This proves the first part of the statement. Note that the third part of Lemma 4.2 is proved bt Lemma D.2. Taking a union bound with the event in Lemma D.4, the second part is also proved. This completes the proof. □

### D.2 LEAST SQUARE VALUE ITERATION ERROR

**Lemma D.7.** *Conditioned on the event in Lemma D.1, for all $(k, h) \in [K] \times [H]$, $b_h^k(\cdot, \cdot) \in \mathcal{W}$ for a function class $\mathcal{W}$ such that*

$$\log|\mathcal{W}| \le C_{\mathcal{W}} \cdot \log(T\mathcal{N}(\mathcal{F}, \delta/T^2)/\delta) \cdot \dim_E(\mathcal{F}, 1/T) \cdot \log^2 T \cdot \log\left(\mathcal{N}(\mathcal{S} \times \mathcal{A}, \delta/T^2) \cdot T/\delta\right)$$

*for some absolute constant $C_{\mathcal{W}}$ and sufficiently large $T$.*

*Proof.* Note that the bonus function $b(\cdot, \cdot)$ is uniquely defined by the subsampled dataset $\hat{Z}$. Conditioned on the event in Lemma D.1, $|\hat{Z}| \le 64T^3/\delta$ and the number of distinct elements in $\hat{Z}$ is at most $N_g$. Since every element in $\hat{Z}$ belongs to $\mathcal{C}(\mathcal{S} \times \mathcal{A}, 1/16\sqrt{64T^3/\delta})$, it holds that

$$\log|\mathcal{W}|$$

$$\le N_g \cdot \log\left(\mathcal{N}(\mathcal{S} \times \mathcal{A}, 1/16\sqrt{64T^3/\delta}) \cdot 64T^3/\delta\right)$$

$$\lesssim \log(T\mathcal{N}(\mathcal{F}, \sqrt{\delta/64T^3}/\delta)) \cdot \dim_E(\mathcal{F}, 1/T) \cdot \log^2 T \cdot \log\left(\mathcal{N}(\mathcal{S} \times \mathcal{A}, 1/16\sqrt{64T^3/\delta}) \cdot 64T^3/\delta\right)$$

$$\lesssim \log(T\mathcal{N}(\mathcal{F}, \delta/T^2)/\delta) \cdot \dim_E(\mathcal{F}, 1/T) \cdot \log^2 T \cdot \log\left(\mathcal{N}(\mathcal{S} \times \mathcal{A}, \delta/T^2)T/\delta\right)$$

where the first inequality holds due to the following counting scheme: Construct $\hat{Z}$ by making $N_g$ choices, each of which determines what element $z \in \mathcal{N}(\mathcal{S} \times \mathcal{A}, 1/16\sqrt{64T^3/\delta})$ to add, and how many copies(at most $64T^3/\delta$) of $z$ to add. □

**Lemma D.8.** *Define the class of Q-functions*

$$\mathcal{Q} = \{\min\{f(\cdot,\cdot) + w(\cdot,\cdot), H\} : f \in \mathcal{F}, w \in \mathcal{W}\}$$

*where $\mathcal{W}$ is defined in Lemma D.7, the class of policies*

$$\Pi = \left\{ \pi(\cdot \mid \cdot) = \frac{\exp\left(\sum_{i=1}^{HN_g} \alpha Q_i(\cdot,\cdot)\right)}{\sum_{a \in \mathcal{A}} \exp\left(\sum_{i=1}^{HN_g} \alpha Q_i(\cdot, a)\right)} : Q_i \in \mathcal{Q} \text{ for } i \in [HN_g] \right\},$$

*and the class of $V$-functions*

$$\mathcal{V} = \{V(\cdot) = \langle Q(\cdot,\cdot), \pi(\cdot \mid \cdot) \rangle : Q \in \mathcal{Q}, \pi \in \Pi\}.$$

*Conditioned on the event in Lemma D.1, the functions generated by Algorithm 2 belong to these classes, i.e., for all $(k,h) \in [K] \times [H]$, $Q_h^k(\cdot,\cdot) \in \mathcal{Q}$, $\pi_h^k(\cdot \mid \cdot) \in \Pi$, and $V_h^k(\cdot) \in \mathcal{V}$. Moreover, for $\varepsilon > 0$, it holds that*

$$\log \mathcal{N}(\mathcal{Q}, \varepsilon) \leq \log \mathcal{N}(\mathcal{Q}, \varepsilon)$$
$$+ C_{\mathcal{W}} \log(T\mathcal{N}(\mathcal{F}, \delta/T^2)/\delta) \cdot dim_E(\mathcal{F}, 1/T) \cdot \log^2 T \cdot \log\left(\mathcal{N}(\mathcal{S} \times \mathcal{A}, \delta/T^2) \cdot T/\delta\right)$$

*and*

$$\log \mathcal{N}(\mathcal{V}, \varepsilon) \leq \log \mathcal{N}(\mathcal{Q}, \varepsilon/2) + HN_g \log \mathcal{N}(\mathcal{Q}, \varepsilon^2/(16\alpha N_g H^3))$$

*Proof.* Conditioned on the event in Lemma D.1, Lemma D.7 implies $Q_h^k(\cdot,\cdot) \in \mathcal{Q}$ for all $(k,h) \in [K] \times [H]$. Since the number of policy switches is bounded by $HN_g$, it follows that $\pi_h^k(\cdot \mid \cdot) \in \Pi$, and clearly $V_h^k(\cdot) \in \mathcal{V}$.

Now we show that the covering number of these classes are bounded. For any $q = \min\{f(\cdot,\cdot) + w(\cdot,\cdot), H\} \in \mathcal{Q}$, we can find $\hat{f} \in \mathcal{F}$ such that $\left\|f - \hat{f}\right\|_\infty \leq \varepsilon$. Since $\hat{q}(\cdot,\cdot) = \min\{\hat{f}(\cdot,\cdot) + w(\cdot,\cdot), H\} \in \mathcal{Q}$ and

$$|q(s,a) - \hat{q}(s,a)| = |\min\{f(s,a) + w(s,a), H\} - \min\{\hat{f}(s,a) + w(s,a), H\}|$$
$$\leq |f(s,a) - \hat{f}(s,a)| \leq \varepsilon$$

for all $(s,a) \in \mathcal{S} \times \mathcal{A}$, we have that

$$\log \mathcal{N}(\mathcal{Q}, \varepsilon)$$
$$\leq \log \mathcal{N}(\mathcal{F}, \varepsilon) + \log |\mathcal{W}|$$
$$= \log \mathcal{N}(\mathcal{F}, \varepsilon)$$
$$+ C_{\mathcal{W}} \log(T\mathcal{N}(\mathcal{F}, \delta/T^2)/\delta) \cdot \dim_E(\mathcal{F}, 1/T) \cdot \log^2 T \cdot \log\left(\mathcal{N}(\mathcal{S} \times \mathcal{A}, \delta/T^2) \cdot T/\delta\right).$$

On the other hand, Lemma F.12 directly implies

$$\log \mathcal{N}(\mathcal{V}, \varepsilon) \leq \log \mathcal{N}(\mathcal{Q}, \varepsilon/2) + HN_g \log \mathcal{N}(\mathcal{Q}, \varepsilon^2/(16\alpha N_g H^3)).$$

$\square$

**Lemma D.9.** *Let $\mathcal{C}_h^k$ be a confidence set defined as*

$$\mathcal{C}_h^k = \left\{f \in \mathcal{F} : \left\|f - f_h^k\right\|_{\mathcal{Z}_h^k}^2 \leq \beta/100\right\}.$$

*Then with probability at least $1 - \delta/4$, for all $(k,h) \in [K] \times [H]$,*

$$r_h(\cdot,\cdot) + P_h V_{h+1}^k(\cdot,\cdot) \in \mathcal{C}_h^k,$$

*provided*

$$\beta = c' \cdot H^2 \left(\log(T/\delta) + \log \mathcal{N}(\mathcal{V}, 1/T)\right) + \log \mathcal{N}(\mathcal{F}, 1/T))$$

*for some absolute constant $c'$ and the function class $\mathcal{V}$ defined in Lemma D.8.*

*Proof.* We condition on the event in Lemma D.1 and $\bigcap_{(k,h)\in[K]\times[H]}\bigcap_{V\in\mathcal{C}(\mathcal{V},1/T)}\mathcal{E}_{V,\delta/(8\mathcal{N}(\mathcal{V},1/T))T}$ where $\mathcal{E}_{V,\delta}$ is defined in Lemma F.13. The events occur with probability at least $1-\delta/4$. For any $V\in\mathcal{V}$, there exists $V'\in\mathcal{C}(\mathcal{V},1/T)$ such that $\|V-V'\|_\infty\leq 1/T$. Therefore, it holds that

$$\left\|\hat{f}_V(\cdot,\cdot)-r_h(\cdot,\cdot)-P_hV(\cdot,\cdot)\right\|_{\mathcal{Z}_h^k}\leq\left\|\hat{f}_{V'}(\cdot,\cdot)-r_h(\cdot,\cdot)-P_hV'(\cdot,\cdot)\right\|_{\mathcal{Z}_h^k}+|\mathcal{Z}_h^k|/T$$
$$\lesssim H\sqrt{\log(8\mathcal{N}(\mathcal{V},1/T)T/\delta)+\log\mathcal{N}(\mathcal{F},1/T)}$$
$$\lesssim H\sqrt{\log(T/\delta)+\log\mathcal{N}(\mathcal{V},1/T)+\log\mathcal{N}(\mathcal{F},1/T)}$$

where the second inequality holds due to $\mathcal{E}_{V',\delta/(8\mathcal{N}(\mathcal{V},1/T))T}$. $\qquad\square$

## D.3 PROOF OF LEMMA 4.3

*Proof of Lemma 4.3.* We condition on the event in Lemma D.9. Since Lemma D.9 condition on the event in Lemma D.1, the results in Lemma D.1 also hold. Define $\bar{f}_h^k(\cdot,\cdot):=r_h(\cdot,\cdot)+P_hV_{h+1}^k(\cdot,\cdot)$. Since $\left\|f_h^k-\bar{f}_h^k\right\|_{\mathcal{Z}_h^k}^2\leq\beta/100$, i.e. $(f_h^k,\bar{f}_h^k)\in\underline{\mathcal{B}}_h^k(\beta)$, it follows that $|(f_h^k(\cdot,\cdot)-\bar{f}_h^k(\cdot,\cdot))|\leq\sup_{f_1,f_2\in\underline{\mathcal{B}}_h^k(\beta)}|f_1(\cdot,\cdot)-f_2(\cdot,\cdot)|=\underline{b}_h^k(\cdot,\cdot)$. Since conditioned on $\mathcal{E}_h^k$, we have $|(f_h^k(\cdot,\cdot)-\bar{f}_h^k(\cdot,\cdot))|\leq b_h^k(\cdot,\cdot)$.

Therefore, for all $(k,h)\in[K]\times[H]$ and for all $(s,a)\in\mathcal{S}\times\mathcal{A}$,

$$\begin{aligned}\xi_h^k(s,a)&=\bar{f}_h^k(s,a)-Q_h^k(s,a)\\&=\bar{f}_h^k(s,a)-\min\left\{f_h^k(s,a)+b_h^k(s,a),H\right\}\\&=\max\left\{\bar{f}_h^k(s,a)-f_h^k(s,a)-b_h^k(s,a),\bar{f}_h^k(s,a)-H\right\}\\&\leq\max\{0,0\}=0\end{aligned}$$

Similarly, it holds that

$$\begin{aligned}-\xi_h^k(s,a)&=Q_h^k(s,a)-\bar{f}_h^k(s,a)\\&=\min\left\{f_h^k(s,a)+b_h^k(s,a),H\right\}-\bar{f}_h^k(s,a)\\&\leq\min\left\{f_h^k(s,a)+b_h^k(s,a)-\bar{f}_h^k(s,a),H-\bar{f}_h^k(s,a)\right\}\\&\leq\min\{2b_h^k(s,a),H\}\leq 2b_h^k(s,a)\end{aligned}$$

$\qquad\square$

## D.4 REGRET BOUND

The following two lemmas from Wang et al. (2020); Kong et al. (2021) bounds the sum of bonuses by utilizing the property of the eluder dimension.

**Lemma D.10.** *With probability at least $1-\delta/8$, for any $\varepsilon>0$, for all $h\in[H]$, it holds that*

$$\sum_{k=1}^K\mathbf{1}\left\{b_h^k(s_h^k,a_h^k)>\varepsilon\right\}\leq\left(\frac{100\beta}{\varepsilon^2}+1\right)\cdot dim_E\left(\mathcal{F},\varepsilon\right).$$

*Proof.* We condition on the event defined in Lemma D.1, which happens with probability at least $1-\delta/8$.

Let $\mathcal{L}_h=\{(s_h^k,a_h^k):b_h^k(s_h^k,a_h^k)>\varepsilon\}$ with $|\mathcal{L}_h|=L_h$. Consider a fixed $h\in[H]$. We show that there exists $(s_h^k,a_h^k)\in\mathcal{L}_h$ such that $(s_h^k,a_h^k)$ is $\varepsilon$-dependent on at least $N_h=L_h/dim_E(\mathcal{F},\varepsilon)-1$ disjoint subsequences in $\mathcal{L}_h$. We decompose $\mathcal{L}_h$ into $N_h+1$ disjoint subsets $\mathcal{L}_h=\bigcup_{j=1}^{N_h+1}\mathcal{L}_h^j$ using the following procedure.

First, initialize $N_h+1$ sets $\mathcal{L}_h^1,\ldots,\mathcal{L}_h^{N_h+1}$ to be empty sets. We consider $z_h^k\in\mathcal{L}_h$ sequentially. For each $z_h^k\in\mathcal{L}_h$, we find the smallest $1\leq j\leq N_h$ such that $z_h^k$ is $\varepsilon$-independent on $\mathcal{L}_h^j$ with respect to $\mathcal{F}$, and add $z_h^k$ into $\mathcal{L}_h^j$. If such $j$ does not exist, add $z_h^k$ into $L_h^{N_h+1}$. By the definition of

the eluder dimension, $|\mathcal{L}_h^j| \leq \dim_E(\mathcal{F}, \varepsilon)$. Thus $\mathcal{L}_h^{N_h+1}$ is nonempty at the end of the procedure, and each $z_h^k \in \mathcal{L}_h^{N_h+1}$ is $\varepsilon$-dependent on at least $N_h = L_h/\dim_E(\mathcal{F}, \varepsilon) - 1$ disjoint subsequences in $\mathcal{Z}_h^k \cap \mathcal{L}_h$.

On the other hand, since $(s_h^k, a_h^k) \in \mathcal{L}_h$, there exist $f_1, f_2 \in \mathcal{B}_h^k(\beta) \subseteq \overline{\mathcal{B}}_h^k(\beta)$ such that $|f_1(s_h^k, a_h^k) - f_2(s_h^k, a_h^k)| > \varepsilon$ and $\|f_1 - f_2\|_{\mathcal{Z}_h^k} \leq 100\beta$. By the definition of $\varepsilon$-independence, it holds that

$$(L_h/\dim_E(\mathcal{F}, \varepsilon) - 1)\varepsilon^2 \leq \|f_1 - f_2\|_{\mathcal{Z}_h^k \cap \mathcal{L}_h} \leq \|f_1 - f_2\|_{\mathcal{Z}_h^k} \leq 100\beta$$

which implies

$$L_h \leq \left(\frac{100\beta}{\varepsilon^2} + 1\right) \cdot \dim_E(\mathcal{F}, \varepsilon).$$

$\square$

**Lemma D.11.** *With probability at least $1 - \delta/8$, it holds that*

$$\sum_{k=1}^{K}\sum_{h=1}^{H} b_h^k(s_h^k, a_h^k) \leq H + H(H+1)dim_E(\mathcal{F}, 1/T) + c'\sqrt{dim_E(\mathcal{F}, 1/T) \cdot TH \cdot \beta}$$

*for some absolute constant $c'$.*

*Proof.* We condition on the event defined in Lemma D.10. Fix $h \in [H]$, then let $b_1 \geq b_2 \geq \cdots \geq b_K$ be a permutation of $\{b_h^k(s_h^k, a_h^k)\}_{k \in [K]}$. For any $b_t \geq 1/T$, by Lemma D.10, we have

$$t \leq \left(\frac{100\beta}{b_t^2} + 1\right) \cdot \dim_E(\mathcal{F}, b_t) \leq \left(\frac{100\beta}{b_t^2} + 1\right) \cdot \dim_E(\mathcal{F}, 1/T),$$

which implies

$$b_t \leq \left(\frac{t}{\dim_E(\mathcal{F}, 1/T)} - 1\right) \cdot \sqrt{100\beta}.$$

Moreover, we have $b_t \leq H + 1$ by definition. Therefore, it holds that

$$\sum_{k=1}^{K} b_h^k(s_h^k, a_h^k)$$
$$\leq \sum_{b_k \leq 1/T} b_k + \sum_{k \leq \dim_E(\mathcal{F}, 1/T)} b_k + \sum_{k > \dim_E(\mathcal{F}, 1/T)} b_k$$
$$\leq \frac{1}{T} \cdot T + (H+1) \cdot \dim_E(\mathcal{F}, 1/T) + \sum_{k > \dim_E(\mathcal{F}, 1/T)} \left(\frac{t}{\dim_E(\mathcal{F}, 1/T)} - 1\right) \cdot \sqrt{100\beta}$$
$$\leq 1 + (H+1)\dim_E(\mathcal{F}, 1/T) + c'\sqrt{\dim_E(\mathcal{F}, 1/T) \cdot K \cdot \beta}$$

where the last inequality holds due to the fact that $\sum_{i=1}^{n} 1/\sqrt{i} \lesssim \sqrt{n}$. Taking the summation over $h \in [H]$ gives

$$\sum_{k=1}^{K}\sum_{h=1}^{H} b_h^k(s_h^k, a_h^k) \leq H + H(H+1)\dim_E(\mathcal{F}, 1/T) + c'\sqrt{\dim_E(\mathcal{F}, 1/T) \cdot TH \cdot \beta}$$

$\square$

As done in the proof of Theorem 1, we can bound the sum of martingale difference sequences.

**Lemma D.12.** *For $\delta > 0$, it holds with probability at least $1 - \delta/2$ that*

$$\sum_{k=1}^{K}\sum_{h=1}^{H} \mathcal{M}_{1,h}^k + \sum_{k=1}^{K}\sum_{h=1}^{H} \mathcal{M}_{2,h}^k \leq 2\sqrt{2H^2 T \log(4/\delta)}$$

*where $\mathcal{M}_{1,h}^k$ and $\mathcal{M}_{2,h}^k$ are defined in Lemma F.3.*

*Proof.* The proof is almost identical to the proof of Lemma C.2, except that $\mathbb{F}_{1,h}^k$ and $\mathbb{F}_{2,h}^k$ are replaced with $\mathbb{G}_{1,h}^k$ and $\mathbb{G}_{2,h}^k$. $\qquad\square$

Now we are ready to prove Theorem 4.1.

*Proof of Theorem 4.1.* We condition on the event defined in Lemma 4.2, Lemma D.12, and Lemma D.11. Using Lemma F.3, with probability at least $1 - \delta$, it holds that

$$
\begin{aligned}
\text{Regret}(K) = & \sum_{k=1}^{K} V_1^*(s_1^k) - V_1^{\pi^k}(s_1^k) \\
= & \sum_{k=1}^{K} \sum_{h=1}^{H} \mathbb{E}_{\pi^*}[\langle Q_h^k(s_h, \cdot), \pi_h^*(\cdot \mid s_h) - \pi_h^k(\cdot \mid s_h)\rangle \mid s_1 = s_1^k] \\
& + \sum_{k=1}^{K} \sum_{h=1}^{H} \mathcal{M}_{1,h}^k + \sum_{k=1}^{K} \sum_{h=1}^{H} \mathcal{M}_{2,h}^k \\
& + \sum_{k=1}^{K} \sum_{h=1}^{H} (\mathbb{E}_{\pi^*}[\xi_h^k(s_h, a_h) \mid s_1 = s_1^k] - \xi_h^k(s_h, a_h)) \\
\leq & \alpha^{-1} H K \log |\mathcal{A}| + 2\sqrt{2H^2 T \log(4/\delta)} + \sum_{k=1}^{K} \sum_{h=1}^{H} 2b_h^k(s_h^k, a_h^k) \\
\leq & \alpha^{-1} H K \log |\mathcal{A}| + 2\sqrt{2H^2 T \log(4/\delta)} \\
& + 2(H + H(H+1)\dim_E(\mathcal{F}, 1/T) + c'\sqrt{\dim_E(\mathcal{F}, 1/T) \cdot TH \cdot \beta}).
\end{aligned}
$$

By Lemma D.8, we have

$$
\begin{aligned}
\log \mathcal{N}(\mathcal{V}, 1/T) \leq & \log \mathcal{N}(\mathcal{Q}, 1/2T) + HN_g \log \mathcal{N}(\mathcal{Q}, 1/(16\alpha N_g T^2 H^3)), \\
\log \mathcal{N}(\mathcal{Q}, 1/2T) \leq & \log \mathcal{N}(\mathcal{F}, 1/2T) + C_\mathcal{W} \log(T\mathcal{N}(\mathcal{F}, \delta/T^2)/\delta) \\
& \cdot \dim_E(\mathcal{F}, 1/T) \cdot \log^2 T \cdot \log\left(\mathcal{N}(\mathcal{S} \times \mathcal{A}, \delta/T^2) \cdot T/\delta\right), \\
\log \mathcal{N}(\mathcal{Q}, 1/(16\alpha N_g T^2 H^3)) \leq & \log \mathcal{N}(\mathcal{F}, 1/(16\alpha N_g T^2 H^3)) + C_\mathcal{W} \log(T\mathcal{N}(\mathcal{F}, \delta/T^2)/\delta) \\
& \cdot \dim_E(\mathcal{F}, 1/T) \cdot \log^2 T \cdot \log\left(\mathcal{N}(\mathcal{S} \times \mathcal{A}, \delta/T^2) \cdot T/\delta\right), \\
N_g = & C_{sw} \cdot \log(T\mathcal{N}(\mathcal{F}, \sqrt{\delta/64T^3}/\delta)) \cdot \dim_E(\mathcal{F}, 1/T) \cdot \log^2 T \\
& \lesssim \log(T\mathcal{N}(\mathcal{F}, \delta/T^2)/\delta) \cdot \dim_E(\mathcal{F}, 1/T) \cdot \log^2 T
\end{aligned}
$$

For sufficiently large T. Hence, it follows that Lemma D.9 holds for $\beta$ such that

$$
\begin{aligned}
\beta \lesssim & H^2 \left(\log(T/\delta) + \log \mathcal{N}(\mathcal{V}, 1/T) + \log \mathcal{N}(\mathcal{F}, 1/T)\right) \\
\lesssim & H^3 \cdot \log(T\mathcal{N}(\mathcal{F}, \delta/T^2)/\delta) \cdot \dim_E(\mathcal{F}, 1/T) \cdot \log^2 T \cdot [\log \mathcal{N}(\mathcal{F}, 1/(16\alpha N_g T^2 H^3)) \\
& + \log(T\mathcal{N}(\mathcal{F}, \delta/T^2)/\delta) \cdot \dim_E(\mathcal{F}, 1/T) \cdot \log^2 T \cdot \log\left(\mathcal{N}(\mathcal{S} \times \mathcal{A}, \delta/T^2) \cdot T/\delta\right)].
\end{aligned}
$$

Setting $\alpha \geq \Omega(\sqrt{T} \log |\mathcal{A}|)$, we have

$$
\begin{aligned}
\text{Regret}(K) \leq & \alpha^{-1} H K \log |\mathcal{A}| + 2\sqrt{2H^2 T \log(4/\delta)} \\
& + 2\left(H + H(H+1)\dim_E(\mathcal{F}, 1/T) + \sqrt{\dim_E(\mathcal{F}, 1/T) \cdot TH \cdot \beta}\right) \\
\lesssim & H\sqrt{T} + \sqrt{H^2 T \log(1/\delta)} + 2\sqrt{2H^2 T \log(4/\delta)} + \sqrt{\dim_E(\mathcal{F}, 1/T) \cdot TH \cdot \beta} \\
\lesssim & \sqrt{\iota \cdot H^4 T}
\end{aligned}
$$

where

$$
\begin{aligned}
\iota = & \log(T\mathcal{N}(\mathcal{F}, \delta/T^2)/\delta) \cdot \dim_E(\mathcal{F}, 1/T)^2 \cdot \log^2 T \cdot [\log \mathcal{N}(\mathcal{F}, 1/(16\alpha N_g T^2 H^3)) \\
& + \log(T\mathcal{N}(\mathcal{F}, \delta/T^2)/\delta) \cdot \dim_E(\mathcal{F}, 1/T) \cdot \log^2 T \cdot \log\left(\mathcal{N}(\mathcal{S} \times \mathcal{A}, \delta/T^2) \cdot T/\delta\right)]
\end{aligned}
$$

$\qquad\square$

# E  SAMPLING-BASED OPORS

In this section, we present a sampling-based variant of OPORS mentioned in Section 4.2. The algorithm implements $O(\log T)$ policy switching by maintaining the approximation of Gram matrices.

## E.1  ALGORITHM AND REGRET BOUND

---

**Algorithm 4** OPORS-v2

---

1: **Input:** Failure probability $\delta \in (0, 1)$, stepsize $\alpha > 0$, confidence radius $\beta$, regularization parameter $\lambda > 0$, sampling precision parameter $\epsilon \in (0, 1)$, sampling probability parameter $c > 0$
2: **Initialize:** Set $\{\pi_h^0(\cdot \mid \cdot)\}_{h \in [H]}$ as uniform policy on $\mathcal{A}$, $\Lambda_h^0, \hat{\Lambda}_h^0 \leftarrow \lambda I$ for $\forall h \in [H+1]$, $\bar{k} = 1$
3: **for** episode $k = 1, \cdots, K$ **do**
4:     **for** step $h = 1, \ldots, H$ **do**
5:         $l_h^k \leftarrow \min\{(1+\epsilon)\phi(s_h^{k-1}, a_h^{k-1})^T(\hat{\Lambda}_h^{k-1})^{-1}\phi(s_h^{k-1}, a_h^{k-1}), 1\}$
6:         $p_h^k \leftarrow \min\{cl_h^k, 1\}$
7:         Set $\hat{\Lambda}_h^k \leftarrow \begin{cases} \hat{\Lambda}_h^{k-1} + \phi(s_h^k, a_h^k)\phi(s_h^{k-1}, a_h^{k-1})^T/p_h^k & \text{with probability } p_k \\ \hat{\Lambda}_{h-1}^k & \text{otherwise} \end{cases}$
8:     **end for**
9:     **if** $k = 1$ **or** $\exists h \in [H]$ $\hat{\Lambda}_h^k \neq \hat{\Lambda}_h^{k-1}$ **then**
10:         **for** step $h = H, \ldots, 1$ **do**
11:             $\Lambda_h^k \leftarrow \sum_{i=1}^{k-1} \phi(s_h^i, a_h^i)\phi(s_h^i, a_h^i)^T + \lambda I$
12:             $\hat{w}_h^k \leftarrow (\Lambda_h^k)^{-1} \sum_{i=1}^{k-1} \phi(s_h^i, a_h^i) \left[r_h(s_h^i, a_h^i) + V_{h+1}^k(s_{h+1}^i)\right]$
13:             $b_h^k(s, a) \leftarrow \beta \|\phi(s, a)\|_{(\hat{\Lambda}_h^k)^{-1}}$
14:             $Q_h^k(s, a) \leftarrow [\phi(s, a)^T \hat{w}_h^k + b_h^k(s, a)]_{[0, H-h+1]}$
15:             Update the policy by $\pi_h^k(a \mid s) \propto \pi_h^{k-1}(a \mid s)\exp(\alpha Q_h^k(s, a))$
16:             $V_h^k(s) \leftarrow \langle Q_h^k(s, \cdot), \pi_h^k(\cdot \mid s)\rangle$
17:         **end for**
18:         $\bar{k} \leftarrow k$
19:     **else**
20:         **for** step $h = 1, \ldots, H$ **do**
21:             $Q_h^k(s, a) \leftarrow Q_h^{k-1}(s, a), \pi_h^k(a \mid s) \leftarrow \pi_h^{k-1}(a \mid s), V_h^k(s) \leftarrow V_h^{k-1}(s)$
22:         **end for**
23:     **end if**
24:     **for** step $h = 1, \ldots, H$ **do**
25:         Take an action $a_h^k \sim \pi_h^k(\cdot \mid s_h^k)$ and observe $s_{h+1}^k$
26:     **end for**
27: **end for**

---

**Sampling.** The sensitivity sampling procedure of Algorithm 3 is computationally expensive, as it requires optimization on the function space $\mathcal{F}$ and rounding $z$ into a finite cover of $\mathcal{S} \times \mathcal{A}$. However, we can efficiently sample and compute the bonus function using the online spectral approximation algorithm presented in Cohen et al. (2016) (Lines 4-8 of Algorithm 4). Starting with $\hat{\Lambda}_h^0 = \lambda I$ for all $h \in [H]$, we maintain the approximate Gram matrices $\{\hat{\Lambda}_h^k\}$. At the $k$-th episode, we compute the online leverage score:

$$l_h^k := \min\{(1+\epsilon)\phi(s_h^{k-1}, a_h^{k-1})^T(\hat{\Lambda}_h^{k-1})^{-1}\phi(s_h^{k-1}, a_h^{k-1}), 1\}.$$

based on this score, we sample $\phi(s_h^{k-1}, a_h^{k-1})$ with probability $p_h^k := \min\{cl_h^k, 1\}$. If $\phi(s_h^{k-1}, a_h^{k-1})$ is sampled, update the approximate Gram matrix by $\hat{\Lambda}_h^k = \hat{\Lambda}_h^{k-1} + \phi(s_h^k, a_h^k)\phi(s_h^{k-1}, a_h^{k-1})^T/p_h^k$. We will prove that with high probability, for all $(k, h) \in [K] \times [H]$, $\hat{\Lambda}_h^k$ is a good approximation of $\Lambda_h^k$ in that

$$(1-\epsilon)\Lambda_h^k \preceq \hat{\Lambda}_h^k \preceq (1+\epsilon)\Lambda_h^k.$$

**Value Function Estimation and Policy Update.** OPORS-v2 switches the policy only if $\{\hat{\Lambda}_h^k\}_{h \in [H]}$ has changed. The value function estimation and policy update procedure (Lines 10-23) are almost identical to that of OPORS (Algorithm 1), except that OPORS-v2 constructs the bonus function with $\{\hat{\Lambda}_h^k\}_{h \in [H]}$.

We have the following regret upper bound for OPORS-v2.

**Theorem E.1.** *Suppose Assumption 1 holds. There exists a constant $\tilde{C}_l > 0$ such that, if we set $\lambda = 1$, $c = 4/\epsilon^2 \cdot \log(4dT/\delta)$, $\alpha = \text{poly}(T, 1/\delta, \log |\mathcal{A}|, d) \geq \Omega(\sqrt{K} \log |\mathcal{A}|)$, and $\beta = \tilde{C}_l \cdot d^{3/2} H^{3/2} \tilde{\chi}_l (1 + \epsilon)/\epsilon$ with $\tilde{\chi}_l = \sqrt{\log^2\left(\frac{dT}{\lambda}\right) \log\left(\frac{dT \log |\mathcal{A}|}{\delta \lambda \epsilon}\right)}$, then with probability at least $1 - \delta$, the regret of Algorithm 4 is upper bounded by*

$$\text{Regret}(K) \leq \tilde{O}\left(\frac{1}{\epsilon(1 - \epsilon)} d^2 H^2 \sqrt{T}\right).$$

### E.2 PROOF OF THEOREM E.1

This section proves Theorem E.1. We define the following filtrations.

$$\mathbb{H}^k := \sigma\left(\{(s_h^\tau, a_h^\tau, r_h^\tau)\}_{(\tau, h) \in [k] \times [H]} \cup \hat{\Lambda}_h^k\right)$$

$$\mathbb{H}_{1,h}^k := \sigma\left(\mathbb{H}^{k-1} \cup \hat{\Lambda}_h^k \cup \{(s_j^k, a_j^k, r_j^k)\}_{j \in [h]}\right)$$

$$\mathbb{H}_{2,h}^k := \sigma\left(\mathbb{H}^{k-1} \cup \hat{\Lambda}_h^k \cup \{(s_j^k, a_j^k, r_j^k)\}_{j \in [h-1]} \cup s_h^k\right)$$

where $\sigma(A)$ is the $\sigma$-algebra generated by $A$. We use $\mathbb{E}_k$ to denote the expectation conditioned on $\mathbb{H}^k$.

The following lemma guarantees that $\hat{\Lambda}_h^k$ is a good approximation of $\Lambda_h^k$, with high probability. Our approach is inspired by Cohen et al. (2016).

**Lemma E.2.** *If we set $c = 4/\epsilon^2 \cdot \log(4dT/\delta)$, then with probability at least $1 - \delta/4$, for all $(k, h) \in [K] \times [H]$, it holds that*

$$(1 - \epsilon)\Lambda_h^k \preceq \hat{\Lambda}_h^k \preceq (1 + \epsilon)\Lambda_h^k$$

*Proof.* Consider a fixed pair $(k, h) \in [K] \times [H]$, then define

$$u_i := (\Lambda_h^k)^{-1/2} \phi(s_h^i, a_h^i), \quad i = 1, \dots, k$$

We construct a matrix martingale $\mathbf{Y}_0, \dots, \mathbf{Y}_k \in \mathbb{R}^{d \times d}$ with the difference sequence $\mathbf{X}_1, \dots, \mathbf{X}_k$. Set $\mathbf{Y} = \mathbf{0}$. For $i = 1, \dots, k$, if $\|\mathbf{Y}_{i-1}\|_2 \geq \epsilon$, we set $\mathbf{X}_i = \mathbf{0}$. Otherwise, let

$$\mathbf{X}_i := \begin{cases} (1/p_h^i - 1) u_i u_i^T & \text{if } \phi(s_h^i, a_h^i) \text{ is sampled into } \hat{\Lambda}_h^i \\ -u_i u_i^T & \text{otherwise.} \end{cases}$$

Note that in this case, we have

$$\mathbf{Y}_{i-1} = (\Lambda_h^k)^{-1/2} (\hat{\Lambda}_h^{i-1} - \Lambda_h^{i-1}) (\Lambda_h^k)^{-1/2}.$$

Since $\|\mathbf{Y}_{i-1}\|_2 < \epsilon$, we have

$$\begin{aligned} l_i &= \min\{(1 + \epsilon) \phi(s_h^i, a_h^i)^T (\hat{\Lambda}_h^{i-1})^{-1} \phi(s_h^i, a_h^i), 1\} \\ &\geq \min\{(1 + \epsilon) \phi(s_h^i, a_h^i)^T (\Lambda_h^{i-1} + \epsilon \Lambda_h^k)^{-1} \phi(s_h^i, a_h^i), 1\} \\ &\geq \min\{(1 + \epsilon) \phi(s_h^i, a_h^i)^T ((1 + \epsilon) \Lambda_h^k)^{-1} \phi(s_h^i, a_h^i), 1\} \\ &= \phi(s_h^i, a_h^i)^T (\Lambda_h^k)^{-1} \phi(s_h^i, a_h^i) \\ &= u_i^T u_i \end{aligned}$$

where the first inequality holds due to the condition $\|\mathbf{Y}_{i-1}\|_2 = \left\|(\Lambda_h^k)^{-1/2} (\hat{\Lambda}_h^{i-1} - \Lambda_h^{i-1}) (\Lambda_h^k)^{-1/2}\right\|_2 < \epsilon$. Thus, we further have $p_h^i = \min\{c l_h^i, 1\} \geq$

$\min\{cu_i^T u_i, 1\}$. If $p_i = 1$, then $\mathbf{X}_i = \mathbf{0}$. Otherwise, we have $1 > p_h^i \geq \min\{cu_i^T u_i, 1\} = cu_i^T u_i$. Hence, it follows that

$$
\begin{aligned}
\|\mathbf{X}_i\|_2 &\leq \max\{(1/p_h^i - 1)\left\|u_i u_i^T\right\|_2, \left\|u_i u_i^T\right\|_2\} \\
&\leq \max\{(1/cu_i^T u_i - 1)\cdot\left\|u_i u_i^T\right\|_2, p_h^i/c\} \leq 1/c
\end{aligned}
$$

and

$$
\begin{aligned}
\mathbb{E}_{i-1}[\mathbf{X}_i^2] &\preceq p_h^i\cdot(1/p_h^i - 1)^2(u_i u_i^T)^2 + (1 - p_h^i)\cdot(u_i u_i^T)^2 \\
&= (u_i u_i^T)^2/p_h^i \preceq (u_i u_i^T)^2/c.
\end{aligned}
$$

For the predictable quadratic variation process $\mathbf{W}_i := \sum_{\tau=1}^i \mathbb{E}_{\tau-1}[\mathbf{X}_\tau^2]$ of the martingale $\{\mathbf{Y}_i\}$, we have

$$
\|\mathbf{W}_i\|_2 \leq \left\|\sum_{\tau=1}^i (u_i u_i^T)^2/c\right\|_2 \leq 1/c.
$$

Therefore, by Lemma F.2, it follows that

$$
\begin{aligned}
\mathbb{P}\left(\|\mathbf{Y}_k\|_2 \geq \epsilon\right) &\leq \mathbb{P}\left(\exists i\, \|\mathbf{Y}_i\|_2 \geq \epsilon\right) \\
&= \mathbb{P}\left(\exists i\, \|\mathbf{Y}_i\|_2 \geq \epsilon \text{ and } \|\mathbf{W}_i\|_2 \leq 1/c\right) \\
&\leq d\cdot\exp\left(-\frac{\epsilon^2/2}{1/c + \epsilon/3c}\right) \\
&\leq d\cdot\exp\left(-c\epsilon^2/4\right) = \delta/4T,
\end{aligned}
$$

if we set $c = 4/\epsilon^2\cdot\log(4dT/\delta)$. This implies that with probability at least $1 - \delta/4T$, it holds that

$$
\|\mathbf{Y}_k\|_2 = \left\|(\Lambda_h^k)^{-1/2}\hat\Lambda_h^k(\Lambda_h^k)^{-1/2} - \mathbf{I}\right\|_2 \leq \epsilon,
$$

and therefore

$$
(1 - \epsilon)\Lambda_h^k \preceq \hat\Lambda_h^k \preceq (1 + \epsilon)\Lambda_h^k.
$$

Taking a union bound for all $(k, h) \in [K]\times[H]$, we get the desired result. $\qquad\square$

Now we bound the number of policy switches. We use the technique presented in Cohen et al. (2016).

**Lemma E.3.** *Conditioned on the event defined in Lemma E.2, with probability at least $1 - \delta/4$, the number of policy switches of Algorithm 4 is bounded by*

$$
\tilde N_l = \tilde C_{sw}\cdot dH\cdot\log(1 + K/\lambda)\log(dT/\lambda))/\epsilon^2
$$

*for some absolute constant $\tilde C_{sw}$.*

*Proof.* For any fixed $h \in [H]$, define

$$
\delta_h^k = \log\det(\hat\Lambda_h^k) - \log\det(\hat\Lambda_h^{k-1}).
$$

Since we know that $\det(\boldsymbol{A} + \boldsymbol{x}\boldsymbol{x}^T) = \det(\boldsymbol{A})(1 + \boldsymbol{x}^T\boldsymbol{A}^{-1}\boldsymbol{x})$ for any vector $\boldsymbol{x}$ and any invertible matrix $\boldsymbol{A}$, it holds that

$$
\begin{aligned}
&\mathbb{E}_{i-1}[\exp(l_h^i/8 - \delta_h^i)] \\
&= p_h^i\cdot e^{l_h^i/8}(1 + \phi(s_h^i, a_h^i)^T(\hat\Lambda_h^{i-1})^{-1}\phi(s_h^i, a_h^i)/p_h^i)^{-1} + (1 - p_h^i)\cdot e^{l_h^i/8} \\
&\leq p_h^i\cdot(1 + l_h^i/4)(1 + \phi(s_h^i, a_h^i)^T(\hat\Lambda_h^{i-1})^{-1}\phi(s_h^i, a_h^i)/p_h^i)^{-1} + (1 - p_h^i)\cdot(1 + l_h^i/4).
\end{aligned}
$$

where the inequality holds due to the fact that $e^x \leq 1 + 2x$ for $x \in [0, 1]$. If $cl_h^i < 1$, we have $p_h^i = cl_h^i$ and $l_h^i = (1 + \epsilon)\phi(s_h^i, a_h^i)^T(\hat\Lambda_h^{i-1})^{-1}\phi(s_h^i, a_h^i)$. Thus, it holds that

$$
\begin{aligned}
\mathbb{E}_{i-1}[\exp(l_h^i/8 - \delta_h^i)] &\leq cl_h^i\cdot(1 + l_h^i/4)(1 + 1/(1 + \epsilon)c)^{-1} + (1 - cl_h^i)\cdot(1 + l_h^i/4) \\
&= (1 + l_h^i/4)(cl_h^i(1 + 1/(1 + \epsilon)c)^{-1} + 1 - cl_h^i) \\
&\leq (1 + l_h^i/4)(1 + cl_h^i(1 - 1/4c - 1)) \leq 1
\end{aligned}
$$

where the second inequality holds due to the fact that $(1 + x)^{-1} \leq 1 - x/2$ for $x \in [0, 1]$ with $0 < 1/(1 + \epsilon)c < 1$. Otherwise, we have $p_i = 1$ and

$$
\begin{aligned}
\mathbb{E}_{i-1}[\exp(l_h^i/8 - \delta_h^i)] &\leq (1 + l_h^i/4)(1 + \phi(s_h^i, a_h^i)^T (\hat{\Lambda}_h^{i-1})^{-1} \phi(s_h^i, a_h^i)/p_h^i)^{-1} \\
&\leq (1 + l_h^i/4)(1 + l_h^i)^{-1} \leq 1
\end{aligned}
$$

On the other hand, for $k \geq 1$, we have

$$
\begin{aligned}
\mathbb{E}\left[\exp\left(\sum_{i=1}^k l_h^i/8 - \delta_h^i\right)\right] &= \mathbb{E}\left[\exp\left(\sum_{i=1}^{k-1} l_h^i/8 - \delta_h^i\right) \mathbb{E}_{k-1}[\exp(l_h^k/8 - \delta_h^k)]\right] \\
&\leq \mathbb{E}\left[\exp\left(\sum_{i=1}^{k-1} l_h^i/8 - \delta_h^i\right)\right].
\end{aligned}
$$

Proceeding recursively, we further have

$$
\mathbb{E}\left[\exp\left(\sum_{i=1}^k l_h^i/8 - \delta_h^i\right)\right] \leq 1.
$$

Hence, Markov's inequality implies

$$
\mathbb{P}\left(\sum_{k=1}^K l_h^k > 8d + 8\sum_{k=1}^K \delta_h^k\right) \leq e^{-d}.
$$

Since we conditioned on the event defined in Lemma E.2, we have

$$
\hat{\Lambda}_h^K \preceq (1 + \epsilon)\Lambda_h^K,
$$

which implies

$$
\log \det(\hat{\Lambda}_h^K) \leq d \log\left((1 + \epsilon)\left\|\Lambda_h^K\right\|_2^2\right) \leq d(1 + 2\log\left\|\Lambda_h^K\right\|_2).
$$

Hence, it holds that

$$
\begin{aligned}
\sum_{k=1}^K l_h^k \leq 8d + 8\sum_h^K \delta_h^k &= 8d + 8(\log \det(\hat{\Lambda}_h^K) - d\log \lambda) \\
&\leq 16d + 8d \log(\left\|\Lambda_h^K\right\|_2^2/\lambda)) \\
&\leq 16d + 8d \log(1 + K/\lambda).
\end{aligned}
$$

For any $(k, h) \in [K] \times [H]$, define an event

$$
\tilde{\mathcal{E}}_h^k = \{(1 - \epsilon)\Lambda_h^k \preceq \hat{\Lambda}_h^k \preceq (1 + \epsilon)\Lambda_h^k\}.
$$

and

$$
Z_h^k := \begin{cases} \mathbf{1}\{\tilde{\mathcal{E}}_h^k\} & \text{if } \phi(s_h^i, a_h^i) \text{ is sampled into } \hat{\Lambda}_h^i \\ 0 & \text{otherwise.} \end{cases}
$$

By Lemma E.2, we have

$$
\sum_{k=1}^K Z_h^k \leq \sum_{k=1}^K p_h^k \leq \sum_{k=1}^K c l_h^k \leq \frac{\tilde{C}_{sw}}{3} d \log(1 + K/\lambda) \log(dT/\lambda))/\epsilon^2 = \tilde{N}_l/3H
$$

for sufficiently large $\tilde{C}_{sw}$. Note that $Z_h^k$ is adapted to $\mathbb{H}^k$ and $|Z_h^k| \leq 1$. Moreover, we have

$$
\mathbb{E}_{k-1}[Z_h^k] = p_h^k,
$$
$$
\mathbb{E}_{k-1}[(Z_h^k - \mathbb{E}_{k-1}[Z_h^k])^2] = p_h^k(1 - p_h^k).
$$

Using Freedman's inequality(Lemma F.1), we obtain

$$\mathbb{P}\left(\sum_{k=1}^{K} Z_h^k \geq \tilde{N}_l/H\right) \leq \mathbb{P}\left(\left|\sum_{k=1}^{K} Z_h^k - \sum_{k=1}^{K} \mathbb{E}_{k-1}[Z_h^k]\right| \geq 2\tilde{N}_l/3H\right)$$

$$\leq 2\exp\left(-\frac{(2\tilde{N}_l/3H)^2/2}{\tilde{N}_l/3H + 2N_3/9H}\right)$$

$$= 2\exp\left(-\frac{(\tilde{C}_{sw}d\log(1+K/\lambda)\log(dT/\lambda))/9\epsilon^2}{1/3 + 2/9}\right)$$

$$\leq \delta/4H$$

Taking a union bound for all $h \in [H]$, with probability at least $1 - \delta/4$,for all $h \in [H]$, we have

$$\sum_{k=1}^{K} Z_h^k \leq \tilde{N}_l/H.$$

Thus it holds that

$$\sum_{k=1}^{K}\sum_{h=1}^{H} Z_h^k \leq \tilde{N}_l.$$

Conditioned on $\bigcap_{k=1}^{K}\bigcap_{h=1}^{H}\tilde{\mathcal{E}}_h^k$, i.e. the event defined in Lemma E.2, the number of policy switches of Algorithm 4 is less than or equal to $\sum_{k=1}^{K}\sum_{h=1}^{H} Z_h^k$. This completes the proof. $\qquad\square$

The following lemma bound the error due to the least square value iteration.

**Lemma E.4.** *With probability at least $1 - \delta/2$, for all $k \in [K], h \in [H], (s,a) \in \mathcal{S} \times \mathcal{A}$, it holds that*

$$|\phi(s,a)^T \hat{w}_h^k - r_h(s,a) - P_h V_{h+1}^k(s,a)| \leq \frac{\beta}{1+\epsilon}\|\phi(s,a)\|_{(\Lambda_h^k)^{-1}}$$

*where $\beta$ is defined by $\beta = \tilde{C}_l d^{3/2} H^{3/2}\tilde{\chi}_l(1+\epsilon)/\epsilon$ with $\tilde{\chi}_l = \sqrt{\log^2\left(\frac{dT}{\lambda}\right)\log\left(\frac{dT\log|\mathcal{A}|}{\delta\lambda\epsilon}\right)}$ for some constant $\tilde{C}_l$.*

*Proof.* By Lemma F.4, we can find $w_h^k$ such that $\phi(s,a)^T w_h^k = r_h(s,a) + P_h V_{h+1}^k(s,a)$ for all $(s,a) \in \mathcal{S} \times \mathcal{A}$. Therefore, we have

$$\phi(s,a)^T \hat{w}_h^k - r_h(s,a) - P_h V_{h+1}^k(s,a)$$

$$= \phi(s,a)^T (\Lambda_h^k)^{-1}\sum_{i=1}^{k-1}\phi(s_h^i, a_h^i)[r_h(s_h^i, a_h^i) + V_{h+1}^k(s_{h+1}^i)] - \phi(s,a)^T w_h^k$$

$$= \phi(s,a)^T (\Lambda_h^k)^{-1}\sum_{i=1}^{k-1}\phi(s_h^i, a_h^i)[r_h(s_h^i, a_h^i) + V_{h+1}^k(s_{h+1}^i)]$$

$$\quad - \phi(s,a)^T (\Lambda_h^k)^{-1}\left(\sum_{i=1}^{k-1}\phi(s_h^i, a_h^i)\phi(s_h^i, a_h^i)^T w_h^k + \lambda w_h^k\right)$$

$$= \phi(s,a)^T (\Lambda_h^k)^{-1}\sum_{i=1}^{k-1}\phi(s_h^i, a_h^i)\left[V_{h+1}^k(s_{h+1}^i) - P_h V_{h+1}^k(s_h^i, a_h^i)\right] - \lambda\phi(s,a)^T (\Lambda_h^k)^{-1}w_h^k$$

where the second last equality uses the definition of $\Lambda_h^k$.

Now we condition on the event defined in Lemma E.3. Since Lemma E.3 states that the number of policy switches is bounded by $\tilde{N}_l = \tilde{C}_{sw}dH\log(1+K/\lambda)\log(dT/\lambda)/\epsilon^2$, Using Lemma F.12

with $M = \tilde{N}_l$, Lemma F.8, and Lemma F.6, we can bound the covering number of the class of $V$-functions:

$$
\begin{aligned}
\log \mathcal{N}(\mathcal{V}, \varepsilon) &\leq \log \mathcal{N}(\mathcal{Q}, \varepsilon/2) + \log \mathcal{N}(\Pi, \varepsilon/2H) \\
&\leq \log \mathcal{N}(\mathcal{Q}, \varepsilon/2) + \tilde{N}_l \cdot \log \mathcal{N}(\mathcal{Q}, \varepsilon/(16\alpha M H^2)) \\
&\leq \left[ d \log(1 + 16H\sqrt{dK/\lambda}/\varepsilon) + d^2 \log(1 + 32\sqrt{d}\beta^2/(\lambda\varepsilon^2)) \right] \\
&\quad + \tilde{N}_l \left[ d \log(1 + 128\alpha M H^3 \sqrt{dK/\lambda}/\varepsilon) + d^2 \log(1 + 2048\alpha^2\sqrt{d}\beta^2 M^2 H^4/(\lambda\varepsilon^2)) \right] \\
&\lesssim \frac{d^3 H}{\epsilon^2} \log\left(1 + \frac{K}{\lambda}\right) \log\left(\frac{dT}{\lambda}\right) \log\left(1 + \frac{\alpha\beta dT}{\lambda\varepsilon^2}\right)
\end{aligned}
$$

Applying this bound to Lemma F.7 with $\varepsilon = \frac{dH}{K}$ and $\alpha = \text{poly}(T, 1/\delta, \log|\mathcal{A}|, d)$, it holds with probability at least $1 - \delta/4$ that

$$
\begin{aligned}
&|\phi(s,a)^T (\Lambda_h^k)^{-1} \sum_{i=1}^{k-1} \phi(s_h^i, a_h^i) \left[ V_{h+1}^i(s_{h+1}^i) - P_h V_{h+1}^k(s_h^i, a_h^i) \right] | \\
&\leq \left\| \sum_{i=1}^{k-1} \phi(s_h^i, a_h^i) \left[ V_{h+1}^i(s_{h+1}^i) - P_h V_{h+1}^k(s_h^i, a_h^i) \right] \right\|_{(\Lambda_h^k)^{-1}} \|\phi(s,a)\|_{(\Lambda_h^k)^{-1}} \\
&\lesssim \frac{d^{3/2} H^{3/2}}{\epsilon} \sqrt{\log^2\left(\frac{dT}{\lambda}\right) \log\left(\frac{\beta dHK \log|\mathcal{A}|}{\delta\lambda\epsilon}\right)} \|\phi(s,a)\|_{(\Lambda_h^k)^{-1}} .
\end{aligned}
$$

On the other hand, by the Cauchy-Schwarz inequality and Lemma F.4, it holds that

$$
|\lambda\phi(s,a)^T (\Lambda_h^k)^{-1} w_h^k| \leq \lambda \left\| w_h^k \right\|_{(\Lambda_h^k)^{-1}} \|\phi(s,a)\|_{(\Lambda_h^k)^{-1}} \leq 2H\sqrt{d} \|\phi(s,a)\|_{(\Lambda_h^k)^{-1}}
$$

where we used the fact that $(\Lambda_h^k)^{-1} \preceq \frac{1}{\lambda} I$. Combining the results above, it follows that

$$
\begin{aligned}
&|\phi(s,a)^T \hat{w}_h^k - r_h(s,a) - P_h V_{h+1}^k(s,a)| \\
&\leq |\phi(s,a)^T (\Lambda_h^k)^{-1} \sum_{i=1}^{k-1} \phi(s_h^i, a_h^i) \left[ V_{h+1}^i(s_{h+1}^i) - P_h V_{h+1}^k(s_h^i, a_h^i) \right] | + |\lambda\phi(s,a)^T (\Lambda_h^k)^{-1} w_h^k| \\
&\leq c' \cdot \frac{d^{3/2} H^{3/2}}{\epsilon} \sqrt{\log^2\left(\frac{dT}{\lambda}\right) \log\left(\frac{\beta dHK \log|\mathcal{A}|}{\delta\lambda\epsilon}\right)} \|\phi(s,a)\|_{(\Lambda_h^k)^{-1}}
\end{aligned}
$$

where $c'$ is some absolute constant. Now the proof is complete if the following inequality holds:

$$
\begin{aligned}
c' \cdot \frac{d^{3/2} H^{3/2}}{\epsilon} \sqrt{\log^2\left(\frac{dT}{\lambda}\right) \log\left(\frac{(\tilde{C}_l d^{3/2} H^{3/2} \tilde{\chi}_l (1+\epsilon)/\epsilon) dHK \log(|\mathcal{A}|)}{\delta\lambda\epsilon}\right)} \|\phi(s,a)\|_{(\Lambda_h^k)^{-1}} \\
\leq \frac{\beta}{1+\epsilon} = \tilde{C}_l d^{3/2} H^{3/2} \tilde{\chi}_l/\epsilon
\end{aligned}
$$

Since $c'$ is independent of $\tilde{C}_l$, we can find an absolute constant $\tilde{C}_l$ satisfying this inequality. This completes the proof. $\qquad \square$

**Lemma E.5.** *With probability at least $1 - \delta/2$, for all $k \in [K], h \in [H], (s,a) \in \mathcal{S} \times \mathcal{A}$, it holds that*

$$
-2\beta \|\phi(s,a)\|_{(\hat{\Lambda}_h^{\bar{k}})^{-1}} \leq \xi_h^k(s,a) \leq 0
$$

*where $\bar{k}$ is the largest index $k' \leq k$ on which the policy is switched.*

*Proof.* By Lemma E.4, with probability at least $1 - \delta/2$, for all $k \in [K], h \in [H]$, we have

$$
|\phi(s,a)^T \hat{w}_h^k - r_h(s,a) P_h V_{h+1}^k(s,a)| \leq \frac{\beta}{1+\epsilon} \|\phi(s,a)\|_{(\Lambda_h^k)^{-1}} .
$$

Note that Lemma E.4 implies the results of Lemma E.2 and Lemma E.3. Therefore, for any $(s, a) \in \mathcal{S} \times \mathcal{A}$, it holds that

$$
\begin{aligned}
\xi_h^k(s, a) &= r_h(s, a) + P_h V_{h+1}^k(s, a) - Q_h^k(s, a) \\
&= r_h(s, a) + P_h V_{h+1}^{\bar{k}}(s, a) - Q_h^{\bar{k}}(s, a) \\
&= r_h(s, a) + P_h V_{h+1}^{\bar{k}}(s, a) - \left[ \phi(s, a)^T \hat{w}_h^{\bar{k}} + \beta \left\| \phi(s, a) \right\|_{(\hat{\Lambda}_h^{\bar{k}})^{-1}} \right]_{[0, H-h+1]} \\
&\leq r_h(s, a) + P_h V_{h+1}^{\bar{k}}(s, a) - \phi(s, a)^T \hat{w}_h^{\bar{k}} - \beta \left\| \phi(s, a) \right\|_{(\hat{\Lambda}_h^{\bar{k}})^{-1}} \\
&\leq \frac{\beta}{1 + \epsilon} \left\| \phi(s, a) \right\|_{(\Lambda_h^{\bar{k}})^{-1}} - \beta \left\| \phi(s, a) \right\|_{(\hat{\Lambda}_h^{\bar{k}})^{-1}} \\
&\leq \frac{\beta}{1 + \epsilon} \left\| \phi(s, a) \right\|_{(\hat{\Lambda}_h^{\bar{k}})^{-1}} - \frac{\beta}{1 + \epsilon} \left\| \phi(s, a) \right\|_{(\Lambda_h^{\bar{k}})^{-1}} = 0
\end{aligned}
$$

where the first equality uses the definition of $\bar{k}$, the first inequality uses the fact that $r_h(s, a) + P_h V_{h+1}^k(s, a) \in [0, H - h + 1]$, and the second last inequality holds due to Lemma E.2. The other direction can be shown similarly:

$$
\begin{aligned}
-\xi_h^k(s, a) &= Q_h^k(s, a) - r_h(s, a) - P_h V_{h+1}^k(s, a) \\
&= Q_h^{\bar{k}}(s, a) - r_h(s, a) - P_h V_{h+1}^{\bar{k}}(s, a) \\
&= \left[ \phi(s, a)^T \hat{w}_h^{\bar{k}} + \beta \left\| \phi(s, a) \right\|_{(\hat{\Lambda}_h^{\bar{k}})^{-1}} \right]_{[0, H-h+1]} - r_h(s, a) - P_h V_{h+1}^{\bar{k}}(s, a) \\
&\leq \phi(s, a)^T \hat{w}_h^{\bar{k}} + \beta \left\| \phi(s, a) \right\|_{(\hat{\Lambda}_h^{\bar{k}})^{-1}} - r_h(s, a) - P_h V_{h+1}^{\bar{k}}(s, a) \\
&\leq \beta \left\| \phi(s, a) \right\|_{(\hat{\Lambda}_h^{\bar{k}})^{-1}} + \frac{\beta}{1 + \epsilon} \left\| \phi(s, a) \right\|_{(\Lambda_h^{\bar{k}})^{-1}} \\
&\leq \beta \left\| \phi(s, a) \right\|_{(\hat{\Lambda}_h^{\bar{k}})^{-1}} + \beta \left\| \phi(s, a) \right\|_{(\hat{\Lambda}_h^{\bar{k}})^{-1}} = 2\beta \left\| \phi(s, a) \right\|_{(\hat{\Lambda}_h^{\bar{k}})^{-1}}
\end{aligned}
$$

$\square$

**Lemma E.6.** *With probability at least $1 - \delta/2$ it holds that*

$$
\sum_{k=1}^K \sum_{h=1}^H \mathcal{M}_{1,h}^k + \sum_{k=1}^K \sum_{h=1}^H \mathcal{M}_{2,h}^k \leq 2\sqrt{2H^2 T \log(4/\delta)}
$$

*where $\mathcal{M}_{1,h}^k$ and $\mathcal{M}_{2,h}^k$ are defined in Lemma F.3.*

*Proof.* The proof is almost identical to the proof of Lemma C.2, except that $\mathbb{F}_{1,h}^k$ and $\mathbb{F}_{2,h}^k$ are replaced with $\mathbb{H}_{1,h}^k$ and $\mathbb{H}_{2,h}^k$. $\square$

Combining these lemmas, we prove Theorem E.1.

*Proof of Theorem E.1.* By Lemma F.3, we have

$$
\begin{aligned}
\text{Regret}(K) &= \sum_{k=1}^K V_1^*(s_1^k) - V_1^{\pi^k}(s_1^k) \\
&= \sum_{k=1}^K \sum_{h=1}^H \mathbb{E}_{\pi^*} \left[ \langle Q_h^k(s_h, \cdot), \pi_h^*(\cdot \mid s_h) - \pi_h^k(\cdot \mid s_h) \rangle \mid s_1 = s_1^k \right] \\
&\quad + \sum_{k=1}^K \sum_{h=1}^H \mathcal{M}_{1,h}^k + \sum_{k=1}^K \sum_{h=1}^H \mathcal{M}_{2,h}^k \\
&\quad + \sum_{k=1}^K \sum_{h=1}^H \left( \mathbb{E}_{\pi^*} [\xi_h^k(s_h, a_h) \mid s_1 = s_1^k] - \xi_h^k(s_h, a_h) \right).
\end{aligned}
$$

Lemma 3.4 with $\alpha \geq \Omega(\sqrt{K} \log(|\mathcal{A}|))$ gives

$$\sum_{k=1}^{K} \sum_{h=1}^{H} \mathbb{E}_{\pi^*}[\langle Q_h^k(s_h, \cdot), \pi_h^*(\cdot \mid s_h) - \pi_h^k(\cdot \mid s_h)\rangle \mid s_1 = s_1^k] \leq H\sqrt{K}.$$

By Lemma E.6, with probability at least $1 - \delta/2$, we have

$$\sum_{k=1}^{K} \sum_{h=1}^{H} \mathcal{M}_{1,h}^k + \sum_{k=1}^{K} \sum_{h=1}^{H} \mathcal{M}_{2,h}^k \leq 2\sqrt{2H^2 T \log(4/\delta)}.$$

On the other hand, Lemma E.5 implies that, with probability at least $1 - \delta/2$, we have

$$\sum_{k=1}^{K} \sum_{h=1}^{H} (\mathbb{E}_{\pi^*}[\xi_h^k(s_h, a_h) \mid s_1 = s_1^k] - \xi_h^k(s_h, a_h))$$

$$\leq 0 + \sum_{k=1}^{K} \sum_{h=1}^{H} 2\beta \left\| \phi(s_h^k, a_h^k) \right\|_{(\hat{\Lambda}_h^{\bar{k}})^{-1}} = \sum_{k=1}^{K} \sum_{h=1}^{H} 2\beta \left\| \phi(s_h^k, a_h^k) \right\|_{(\hat{\Lambda}_h^k)^{-1}}$$

$$\leq \sum_{k=1}^{K} \sum_{h=1}^{H} \frac{2\beta}{1 - \epsilon} \left\| \phi(s_h^k, a_h^k) \right\|_{(\Lambda_h^k)^{-1}}$$

where the equality holds due to the definition of $\hat{\Lambda}_h^k$, and the last inequality holds due to Lemma E.2. Further, it follows that

$$\sum_{k=1}^{K} \sum_{h=1}^{H} \frac{2\beta}{1 - \epsilon} \left\| \phi(s_h^k, a_h^k) \right\|_{(\Lambda_h^k)^{-1}} \leq \frac{2\beta}{1 - \epsilon} \sum_{h=1}^{H} \sqrt{K} \sqrt{\sum_{k=1}^{K} \left\| \phi(s_h^k, a_h^k) \right\|_{(\Lambda_h^k)^{-1}}}$$

$$\leq \frac{2\beta}{1 - \epsilon} \sqrt{K} \sum_{h=1}^{H} \sqrt{2 \log \frac{\det(\Lambda_h^K)}{\det(\Lambda_h^1)}} \leq \frac{2\beta}{1 - \epsilon} H\sqrt{K} \sqrt{2d \log\left(1 + \frac{K}{\lambda d}\right)}$$

where the first inequality uses the Cauchy-Schwartz inequality, the second inequality holds due to Lemma F.9, and the last inequality holds due to Lemma F.11. Combining the inequalities above, with probability at least $1 - \delta$, we have

$$Regret(K) \leq H\sqrt{K} + 2\sqrt{2H^2 T \log(4/\delta)} + \frac{2\beta}{1 - \epsilon} H\sqrt{K} \sqrt{2d \log\left(1 + \frac{K}{\lambda d}\right)}$$

$$\lesssim \frac{d^2 H^2 \sqrt{T}}{\epsilon(1 - \epsilon)} \sqrt{\log^3\left(\frac{dT}{\lambda}\right) \log\left(\frac{dT \log(|\mathcal{A}|)}{\delta \lambda \epsilon}\right)}$$

since we set $\beta = \tilde{C}_l d^{3/2} H^{3/2} \sqrt{\log^2\left(\frac{dT}{\lambda}\right) \log\left(\frac{dT \log|\mathcal{A}|}{\delta \lambda \epsilon}\right)} (1 + \epsilon)/\epsilon$.

$\square$

## F    SUPPORTING LEMMAS

**Lemma F.1** (Freedman's inequality, Freedman (1975)). *Let $\{Y_k\}_{k \in \mathbb{N}}$ be a real-valued martingale with difference sequence $\{X_k\}_{k \in \mathbb{N}}$. Assume that the difference sequence is uniformly bounded:*

$$|X_k| \leq R \text{ almost surely for } \forall k \in \mathbb{N}$$

*For a fixed $n \in \mathbb{N}$, assume that*

$$\sum_{k=1}^{n} \mathbb{E}_k[X_k^2] \leq \sigma^2$$

*for some $\sigma > 0$ almost surely. Then for all $t \geq 0$, it holds that*

$$\mathbb{P}(|Y_n - Y_0| \geq t) \leq 2 \exp\left(-\frac{t^2/2}{\sigma^2 + Rt/3}\right)$$

**Lemma F.2** (Matrix Freedman's inequality, Tropp (2011)). *Consider a matrix martingale* $\{\mathbf{Y}_k\}_{k\in\mathbb{N}\cup\{0\}}$ *whose values are self-adjoint matrices with dimension* $d$, *and let* $\{\mathbf{X}_k\}_{k\in\mathbb{N}}$ *be the difference sequence. Assume that the difference sequence is uniformly bounded in the sense that*

$$\|\mathbf{X}_k\|_2 \le R \text{ almost surely for } k = 1, \ldots, n.$$

*Define the predictable quadratic variation process of the martingale:*

$$\mathbf{W}_k := \sum_{j=1}^{k} \mathbb{E}_{j-1}[X_j^2], \text{ for } k = 1, \ldots, n.$$

*Then, for all* $\epsilon > 0$ *and* $\sigma^2 > 0$,

$$\mathbb{P}\left(\exists k \ge 0 : \|\mathbf{Y}_k\|_2 \ge t \text{ and } \|\mathbf{W}_k\|_2 \le \sigma^2\right) \le d \cdot \exp\left(-\frac{t^2/2}{\sigma^2 + Rt/3}\right)$$

**Lemma F.3** (Lemma 4.2 in Cai et al. (2020)). *It holds that*

$$\text{Regret}(K) = \sum_{k=1}^{K} V_1^*(s_1^k) - V_1^{\pi^k}(s_1^k)$$

$$= \underbrace{\sum_{k=1}^{K} \sum_{h=1}^{H} \mathbb{E}_{\pi^*}[\langle Q_h^k(s_h, \cdot), \pi_h^*(\cdot \mid s_h) - \pi_h^k(\cdot \mid s_h)\rangle \mid s_1 = s_1^k]}_{\textit{Policy optimization error}}$$

$$+ \underbrace{\sum_{k=1}^{K} \sum_{h=1}^{H} \mathcal{M}_{1,h}^k + \sum_{k=1}^{K} \sum_{h=1}^{H} \mathcal{M}_{2,h}^k}_{\textit{Sum of martingale difference sequences}} + \underbrace{\sum_{k=1}^{K} \sum_{h=1}^{H} (\mathbb{E}_{\pi^*}[\xi_h^k(s_h, a_h) \mid s_1 = s_1^k] - \xi_h^k(s_h, a_h))}_{\textit{Statistical error}}$$

*where*

$$\mathcal{M}_{1,h}^k := P_h[V_{h+1}^k - V_{h+1}^{\pi^k}](s_h^k, a_h^k) - [V_{h+1}^k - V_{h+1}^{\pi^k}](s_{h+1}^k),$$

$$\mathcal{M}_{2,h}^k := \langle [Q_h^k - Q_h^{\pi^k}](s_h^k, \cdot), \pi_h^k(\cdot \mid s_h^k)\rangle - [Q_h^k - Q_h^{\pi^k}](s_h^k, a_h^k),$$

$$\xi_h^k(\cdot, \cdot) := r_h(\cdot, \cdot) + P_h V_{h+1}^k(\cdot, \cdot) - Q_h^k(\cdot, \cdot).$$

**Lemma F.4.** *For any* $(k, h) \in [K] \times [H]$, *there exists* $w_h^k \in \mathbb{R}^d$ *such that* $\phi(s, a)^T w_h^k = r_h(s, a) + P_h V_{h+1}^k(s, a)$ *for all* $(s, a) \in \mathcal{S} \times \mathcal{A}$. *Furthermore,* $\|w_h^k\|_2 \le H\sqrt{d}$.

*Proof.* The proof is almost identical to that of Lemma B.1 in Jin et al. (2020). By Assumption 1, we know

$$r_h(s, a) + P_h V_{h+1}^k(s, a) = \phi(s, a)^T \boldsymbol{\theta}_h + \int_{\mathcal{S}} V_{h+1}^k(s)\phi(s, a)^T d\boldsymbol{\mu}_h(s) = \phi(s, a)^T \left(\boldsymbol{\theta}_h + \int_{\mathcal{S}} V_{h+1}^k(s)d\boldsymbol{\mu}_h(s)\right)$$

which implies $w_h^k = \boldsymbol{\theta}_h + \int_{\mathcal{S}} V_{h+1}^k(s)d\boldsymbol{\mu}_h(s)$. The boundedness assumption gives $\|w_h^k\|_2 = \|\boldsymbol{\theta}_h + \int_{\mathcal{S}} V_{h+1}^k(s)d\boldsymbol{\mu}_h(s)\|_2 \le \|\boldsymbol{\theta}_h\|_2 + \|\int_{\mathcal{S}} V_{h+1}^k(s)d\boldsymbol{\mu}_h(s)\|_2 \le \sqrt{d} + H\sqrt{d} \le 2H\sqrt{d}$ □

**Lemma F.5** (Lemma D.1 in Jin et al. (2020)). *Let* $\Lambda_t = \lambda I + \sum_{i=1}^{t} \phi_i \phi_i^T$ *where* $\phi_i \in \mathbb{R}^d$ *and* $\lambda > 0$. *Then we have* $\sum_{i=1}^{t} \phi_i^T (\Lambda_t)^{-1} \phi_i \le d$.

**Lemma F.6.** *For any* $(k, h) \in [K] \times [H]$, *the estimator* $\hat{w}_h^k$ *in Algorithm 1 and Algorithm 4 satisfies* $\|\hat{w}_h^k\|_2 \le 2H\sqrt{dk/\lambda}$

*Proof.* The proof is almost identical to that of Lemma B.2 in Jin et al. (2020). For any vector $x \in \mathbb{R}^d$, we have

$$|x^T \hat{w}_h^k| = |x^T (\Lambda_h^k)^{-1} \sum_{i=1}^{k-1} \phi(s_h^i, a_h^i)[r_h(s_h^i, a_h^i) + V_{h+1}^k(s_{h+1}^i)]|$$

$$\le \sum_{i=1}^{k-1} |x^T (\Lambda_h^k)^{-1} \phi(s_h^i, a_h^i)| \cdot 2H \le 2H \sqrt{\left[\sum_{i=1}^{k-1} x^T (\Lambda_h^k)^{-1} x\right] \cdot \left[\sum_{i=1}^{k-1} \phi(s_h^i, a_h^i)^T (\Lambda_h^k)^{-1} \phi(s_h^i, a_h^i)\right]}$$

$$\le 2H \|x\|_2 \sqrt{dk/\lambda}$$

where the last inequality holds due to Lemma F.5. Since $x$ is arbitrary, the proof is complete. $\qquad\square$

**Lemma F.7** (Lemma D.4 in Jin et al. (2020)). *Let $\{x_t\}_{t=1}^\infty \in \mathcal{S}$ and $\{\phi_t\}_{t=1}^\infty \in \mathbb{R}^d$ be stochastic processes adapted to filtration $\{\mathcal{F}_t\}_{t=1}^\infty$ and $\|\phi_t\|_2 \leq 1$ for all $t$. Let $\Lambda_k = \lambda I + \sum_{t=1}^k \phi_t\phi_t^T$. Then for $0 < \delta < 1$, with probability at least $1 - \delta$, for all $k \geq 1$, and for any $V \in \mathcal{V}$ satisfying $\sup_x |V(x)| \leq H$, it holds that*

$$\left\| \sum_{t=1}^k \phi_t \{V(x_{t+1}) - \mathbb{E}[V(x_{t+1}) \mid \mathcal{F}_t]\} \right\|_{\Lambda_k^{-1}}^2 \leq 4H^2 \left[ \frac{d}{2}\log\left(1 + \frac{k}{\lambda}\right) + \log\frac{\mathcal{N}(\mathcal{V},\varepsilon)}{\delta} \right] + \frac{8k^2\varepsilon^2}{\lambda}$$

*where $\mathcal{N}(\mathcal{V},\varepsilon)$ is the $\varepsilon$-covering number of $\mathcal{V}$ with respect to the distance $dist(V,V') = \|V - V'\|_\infty$.*

**Lemma F.8** (Lemma D.6 in Jin et al. (2020)). *Define a function class*

$$\mathcal{Q} = \left\{ Q(\cdot,\cdot) = \left[ w^T\phi(\cdot,\cdot) + \beta\sqrt{\phi(s,a)^T\Lambda^{-1}\phi(s,a)} \right]_{[0,H-h+1]} : \|w\|_2 \leq L, \beta \in [0,B], \lambda_{min}(\Lambda) \geq \lambda \right\}$$

*where $\lambda_{min}(A)$ is the minimum eigenvalue of $A$. Assuming $\|\phi(s,a)\|_2 \leq 1$ for all $(s,a)$, the $\varepsilon$-covering number $\mathcal{N}(\mathcal{Q},\varepsilon)$ of the function class $\mathcal{Q}$ with respect to the distance $dist(Q,Q') = \|Q - Q'\|_\infty$ satisfies*

$$\log\mathcal{N}(\mathcal{Q},\varepsilon) \leq d\log(1 + 4L/\varepsilon) + d^2\log(1 + 8\sqrt{d}B^2/(\lambda\varepsilon^2)).$$

**Lemma F.9** (Lemma 11 in Abbasi-Yadkori et al. (2011)). *Let $\{X_t\}_{t=1}^\infty$ be a sequence in $\mathbb{R}^d$, and let $V$ a $d \times d$ positive definite matrix and define $\bar{V}_t = V + \sum_{s=1}^t X_s X_s^T$. Assuming $\|X_t\|_2 \leq 1$ for all $t$, then*

$$\log\frac{\det(\bar{V}_n)}{\det(V)} \leq \sum_{t=1}^n \|X_t\|_{\bar{V}_{t-1}^{-1}}^2 \leq 2\log\frac{\det(\bar{V}_n)}{\det(V)}$$

**Lemma F.10** (Lemma 12 in Abbasi-Yadkori et al. (2011)). *Suppose $A, B \in \mathbb{R}^{d\times d}$ are two positive definite matrices such that $A \succ B$. Then for any $x \in \mathbb{R}^d$, $\|x\|_A \leq \|x\|_B \cdot \sqrt{\det(A)/\det(B)}$*

**Lemma F.11** (Lemma C.1 in Wang et al. (2021)). *Let $\Lambda = \lambda I + \sum_{k=1}^K \phi_k\phi_k^T$ with $\|\phi_k\|_2 \leq 1, \forall k \in [K]$. Then $\det(\Lambda) \leq (\lambda + K/d)^d$.*

**Lemma F.12** (Covering number, Zhong & Zhang (2023)). *Define a policy class*

$$\Pi = \left\{ \pi(\cdot \mid \cdot) = \frac{\exp\left(\sum_{i=1}^M \alpha Q_i(\cdot,\cdot)\right)}{\sum_{a\in\mathcal{A}} \exp\left(\sum_{i=1}^M \alpha Q_i(\cdot,a)\right)} : Q_i \in \mathcal{Q} \right\}$$

*where $\mathcal{Q}$ is the class of Q-functions. Based on the policy class, define a class of V-functions*

$$\mathcal{V} = \{V(\cdot) = \langle Q(\cdot,\cdot), \pi(\cdot \mid \cdot)\rangle : Q \in \mathcal{Q}, \pi \in \Pi\}.$$

*Then it holds that*

$$\mathcal{N}(\mathcal{V},\varepsilon) \leq \mathcal{N}(\mathcal{Q},\varepsilon/2) \cdot \mathcal{N}(\Pi,\varepsilon/(2H)).$$

*where the covering number of $\Pi$ is with respect to the distance $dist(\pi,\pi') = \sup_s \|\pi(\cdot \mid s) - \pi'(\cdot \mid s)\|_1$. Furthermore, we can bound the covering number of $\Pi$ by*

$$\mathcal{N}(\Pi,\varepsilon/(2H)) \leq \left(\mathcal{N}(\mathcal{Q},\varepsilon^2/(16\alpha MH^2))\right)^M$$

*Proof.* Given any $V(\cdot) = \langle Q(\cdot,\cdot), \pi(\cdot \mid \cdot)\rangle \in \mathcal{V}$, there exist $(Q',\pi') \in \mathcal{C}(\mathcal{Q},\varepsilon/2) \times \mathcal{C}(\Pi,\varepsilon/2H)$ such that

$$\|Q - Q'\|_\infty \leq \varepsilon/2, \ \sup_{s\in\mathcal{S}} \|\pi(\cdot \mid s) - \pi'(\cdot \mid s)\|_1 \leq \varepsilon/(2H).$$

Hence, for $V'(\cdot) = \langle Q(\cdot, \cdot), \pi(\cdot \mid \cdot) \rangle$, we have

$$
\begin{aligned}
\|V - V'\|_\infty &= \sup_{s \in \mathcal{S}} |\langle Q(s, \cdot), \pi(\cdot \mid s) \rangle - \langle Q'(s, \cdot), \pi'(\cdot \mid s) \rangle| \\
&\leq \sup_{s \in \mathcal{S}} |\langle Q(s, \cdot) - Q'(s, \cdot), \pi(\cdot \mid s) \rangle| + \sup_{s \in \mathcal{S}} |\langle Q'(s, \cdot), \pi(\cdot \mid s) - \pi'(\cdot \mid s) \rangle| \\
&\leq \sup_{s \in \mathcal{S}} \|Q(s, \cdot) - Q'(s, \cdot)\|_\infty + \sup_{s \in \mathcal{S}} H \|\pi(\cdot \mid s) - \pi'(\cdot \mid s)\|_1 \\
&\leq \varepsilon/2 + H \cdot \varepsilon/(2H) = \varepsilon
\end{aligned}
$$

where the first inequality holds due to the triangle inequality and $\sup_x f(x) + g(x) \leq \sup_x f(x) + \sup_x g(x)$, the second inequality uses Hölder's inequality with the fact that $\|Q(s, \cdot)\|_\infty \leq H$ for all $s \in \mathcal{S}$. Therefore, $\mathcal{C}(\mathcal{Q}, \varepsilon/2) \times \mathcal{C}(\Pi, \varepsilon/2H)$ is a $\varepsilon$-covering net of $\mathcal{V}$, which implies

$$
\mathcal{N}(\mathcal{V}, \varepsilon) \leq \mathcal{N}(\mathcal{Q}, \varepsilon/2) \cdot \mathcal{N}(\Pi, \varepsilon/2H).
$$

Now we bound the covering number of $\Pi$. For any $\pi \in \Pi$, $\pi$ takes the form

$$
\pi(\cdot \mid \cdot) = \frac{\exp\left(\sum_{i=1}^M \alpha Q_i(\cdot, \cdot)\right)}{\sum_{a \in \mathcal{A}} \exp\left(\sum_{i=1}^M \alpha Q_i(\cdot, a)\right)}
$$

where $Q_i \in \mathcal{Q}$ for $i \in [M]$. We can find $Q'_i \in \mathcal{C}(\mathcal{Q}, \varepsilon^2/(16\alpha M H^2))$ such that $\|Q_i - Q'_i\|_\infty \leq \varepsilon^2/(16\alpha M H^2)$ for $i \in [M]$, and define

$$
\pi'(\cdot \mid \cdot) = \frac{\exp\left(\sum_{i=1}^M \alpha Q'_i(\cdot, \cdot)\right)}{\sum_{a \in \mathcal{A}} \exp\left(\sum_{i=1}^M \alpha Q'_i(\cdot, a)\right)}.
$$

On the other hand, for any probability distributions $\pi, \pi'$ such that $\pi(\cdot) \propto \exp(Q(\cdot))$ and $\pi'(\cdot) \propto \exp(Q'(\cdot))$ for some $Q, Q' : \mathcal{A} \to \mathbb{R}^+$, we have

$$
\begin{aligned}
\|\pi - \pi'\|_1 &\leq \sqrt{2 D_{KL}(\pi \| \pi')} = \sqrt{2 \sum_{a \in \mathcal{A}} \pi(a) \log \frac{\pi(a)}{\pi'(a)}} \\
&= \sqrt{2 \sum_{a \in \mathcal{A}} \pi(a) \log \left[\exp(Q(a) - Q'(a)) \frac{\sum_{a' \in \mathcal{A}} \exp(Q'(a'))}{\sum_{a' \in \mathcal{A}} \exp(Q(a'))}\right]} \\
&\leq \sqrt{2 \sum_{a \in \mathcal{A}} \pi(a) \log \left[\exp(\|Q - Q'\|_\infty) \frac{\exp(\|Q - Q'\|_\infty) \sum_{a' \in \mathcal{A}} \exp(Q(a'))}{\sum_{a' \in \mathcal{A}} \exp(Q(a'))}\right]} \\
&= \sqrt{2 \sum_{a \in \mathcal{A}} \pi(a) 2 \|Q - Q'\|_\infty} = 2\sqrt{\|Q - Q'\|_\infty}
\end{aligned}
$$

where the first inequality holds due to Pinsker's inequality, and the second inequality holds due to the fact that

$$
\frac{\sum_{a' \in \mathcal{A}} \exp(Q'(a'))}{\sum_{a' \in \mathcal{A}} \exp(Q(a'))} = \frac{\sum_{a' \in \mathcal{A}} \exp(Q'(a') - Q(a')) \exp(Q(a'))}{\sum_{a' \in \mathcal{A}} \exp(Q(a'))} \leq \frac{\exp(\|Q - Q'\|_\infty) \sum_{a' \in \mathcal{A}} \exp(Q(a'))}{\sum_{a' \in \mathcal{A}} \exp(Q(a'))}.
$$

Combining the results, we have

$$
\begin{aligned}
\sup_{s \in \mathcal{S}} \|\pi(\cdot \mid s) - \pi'(\cdot \mid s)\|_1 &\leq \sup_{s \in \mathcal{S}} 2\sqrt{\left\|\sum_{i=1}^M \alpha Q_i(s, \cdot) - \sum_{i=1}^M \alpha Q'_i(s, \cdot)\right\|_\infty} \\
&\leq \sup_{s \in \mathcal{S}} 2\sqrt{\alpha \sum_{i=1}^M \|Q_i(s, \cdot) - Q'_i(s, \cdot)\|_\infty} \\
&\leq \sup_{s \in \mathcal{S}} 2\sqrt{\alpha \sum_{i=1}^M \varepsilon^2/(16\alpha M H^2)} \leq \varepsilon/(2H).
\end{aligned}
$$

where the first inequality holds due to the inequality we derived above. Hence, we can construct a $\mathcal{C}(\Pi, \varepsilon/2H)$ from $\prod_{i=1}^{M} \mathcal{C}(\mathcal{Q}, \varepsilon^2/(16\alpha MH^2))$, which implies

$$\mathcal{N}(\Pi, \varepsilon/(2H)) \leq \left(\mathcal{N}(\mathcal{Q}, \varepsilon^2/(16\alpha MH^2))\right)^M.$$

$\square$

**Lemma F.13** (Lemma 7 in Kong et al. (2021)). *Consider a fixed pair* $(k, h) \in [K] \times [H]$. *Define*

$$\mathcal{Z}_h^k = \{(s_h^\tau, a_h^\tau)\}_{\tau \in [k-1]},$$

*and for any* $V \to [0, H]$, *define*

$$\mathcal{D}_h^k(V) = \{(s_h^\tau, a_h^\tau, r_h^\tau + V(s_{h+1}^\tau))\}_{\tau \in [k-1]}$$

*and*

$$\hat{f}_V := \arg\min_{f \in \mathcal{F}} \|f\|_{\mathcal{D}_h^k}^2.$$

*For any* $V \to [0, H]$ *and* $\delta \in (0, 1)$, *there is an event* $\mathcal{E}_{V,\delta}$ *which holds with probability at least* $1 - \delta$, *such that conditioned on* $\mathcal{E}_{V,\delta}$, *for any* $V' \to [0, H]$ *with* $\|V' - V\|_\infty \leq 1/T$, *we have*

$$\left\| \hat{f}_V(\cdot, \cdot) - r_h(\cdot, \cdot) - \sum_{s' \in \mathcal{S}} P_h(s' \mid \cdot, \cdot)V'(s') \right\|_{\mathcal{Z}_h^k} \leq c' \cdot H\sqrt{\log(1/\delta) + \log \mathcal{N}(\mathcal{F}, 1/T)}$$

*for some absolute constant* $c'$.

## G EXPERIMENT DETAILS

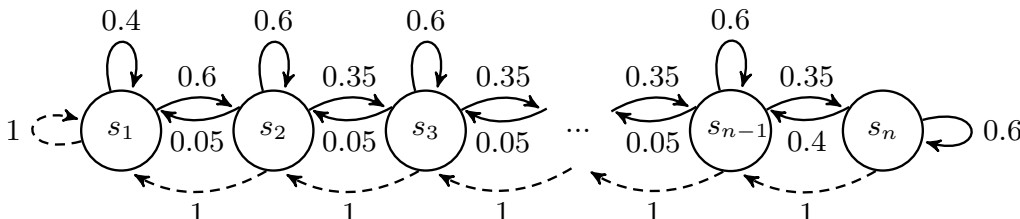

Figure 3: The "RiverSwim" environment with $n$ states

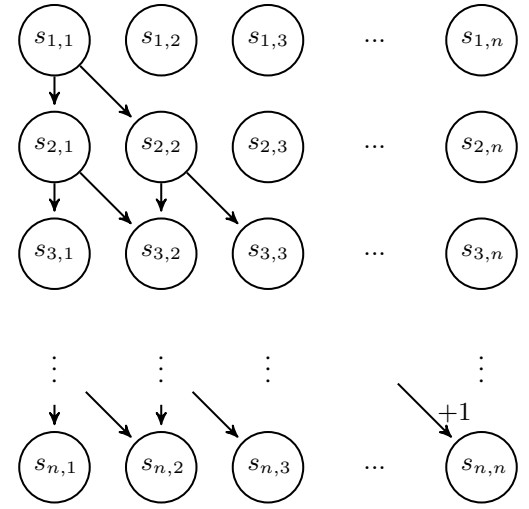

Figure 4: The "DeepSea" environment with $n \times n$ states

Figure 3 shows a diagram of the RiverSwim environment. $|\mathcal{S}| = n$ states are lined up in a chain, and the agent starts from the leftmost state $s_1$. In each state, the agent can swim to the left or to the right. If the agent swims to the left, along the current, it deterministically moves to the left (dotted arrows). However, swimming to the right causes stochastic transitions, as the agent swims against the current (solid arrows). If the agent swims to the left at $s_1$, it receives a small reward of $0.05$. To maximize return, the agent should reach the rightmost state $s_n$ and then swim to the right, where it receives a large reward of $1$.

The DeepSea environment in Figrue 4 is an $n \times n$ grid of states. The agent starts in the top left corner $s_{1,1}$, and in each state, the agent can choose to just move down or move down and right. Moving down and right gives a reward of $-0.01/n$, and moving down gives zero reward. The episode terminates after $n$ steps, and if the agent reaches the bottom right corner $s_{n,n}$, it receives a large reward of $+1$. Our implementation is slightly different from Osband et al. (2019), in that the 'moving down' action leads to just moving down instead of moving down and left.

It is easy to verify that our theoretical guarantees are valid if the stepsize at episode $k$ (say $\alpha_k$) is a random variable. Hence, we set $\alpha_k = \alpha_0 \cdot (k - \bar{k})$ for the experiments, where $\bar{k}$ is the largest episode index $\bar{k} < k$ on which the policy is switched. To tune the hyperparameters of OPORS, we sweep over $\alpha_0$ and $\beta$, while fixing $\eta = 2$. For LSVI-UCB, we sweep over the confidence radius $\beta$. We sweep over the noise level $\sigma$ for LSVI-PHE while setting the number of sampling $M$ as suggested in Ishfaq et al. (2021). For OPPO+, we sweep over the stepsize $\alpha$ and the confidence radius $\beta$, while fixing the batch size $B$ as suggested in Zhong & Zhang (2023).

