# OpenReview forum: "Provably Efficient Policy Optimization with Rare Policy Switches"
_ICLR.cc/2024/Conference — ICLR 2024 Conference Withdrawn Submission_

### Official Review · Reviewer_cKeF · 2023-10-30

**Soundness:** 3 good
**Presentation:** 3 good
**Contribution:** 2 fair
**Rating:** 5
**Confidence:** 3

**Summary:**

The authors address the gap by proposing a new provably efficient policy optimization algorithm that incorporates optimistic value estimation and rare policy switches. Their algorithm achieves the sharpest regret bound of a policy optimization algorithm for linear MDPs. Furthermore, the authors extend their algorithm to general function approximation, and establish a regret bound of $\tilde{O}(\sqrt{T})$. This is the first regret guarantee of a policy optimization algorithm with general function approximation. The authors also provide numerical experiments to demonstrate that their algorithm has competitive regret performances compared to existing RL algorithms while being computationally efficient.

**Strengths:**

1.	The studied problem, i.e., policy optimization algorithms for MDPs with linear and general function approximation, is important in the RL literature.
2.	The authors provide both computationally-efficient and sample-efficient algorithms. The achieved regret bounds are tighter than those of existing policy optimization algorithms.
3.	The authors also conduct experiments to demonstrate the competitive performances of their algorithms in practice.

**Weaknesses:**

1.	It seems that the proposed algorithms use some existing techniques, e.g., rare switches, to improve the regret bounds of policy optimization algorithms. However, the idea of such lazy update or doubling trick is well-known in the literature. The authors should discuss more on their technical novelty.
2.	In the last paragraph of Section 3.1, the authors highlight the difference on the policy update rule, i.e., using $Q^k_h(s,\cdot)$ instead of $Q^{k-1}_h(s,\cdot)$, compared to existing policy optimization algorithms. Is this important? It seems that it just depends on whether you update the policy or the Q-value function first and the order that you assign the superscript of the Q-value function. The authors should explain more on how this helps improve the regret bound.
3.	The authors should elaborate more on the optimalities of the proposed algorithms for linear and general function approximation with respect to problem parameters such as $d$ and $H$.

**Questions:**

Please see the weaknesses above.

---

### Official Review · Reviewer_KS5n · 2023-10-31

**Soundness:** 2 fair
**Presentation:** 3 good
**Contribution:** 3 good
**Rating:** 6
**Confidence:** 2

**Summary:**

The paper delves into the realm of policy optimization algorithms within Markov decision processes with function approximation. Notably, the authors propose a policy-based algorithm that attains a $\sqrt{T}$ regret for linear MDPs. This algorithm is characterized by its distinctive design, incorporating elements such as optimistic value estimation and rare policy switches. Beyond the scope of linear function approximation, their findings have a broader reach, encompassing general function approximation settings, thus expanding the applicability of their contributions. To substantiate the superiority of their proposed algorithms, the authors undertake a series of numerical experiments, offering compelling empirical evidence regarding the practical effectiveness of these methods in the domains of reinforcement learning.

**Strengths:**

The policy optimization algorithm studied in this paper hold significant interest within the realm of reinforcement learning theory. The attainment of a $\sqrt{T}$ regret for policy-based algorithms in the context of linear MDPs, and even low eluder MDPs, is a noteworthy achievement. Additionally, the authors bolster their theoretical contributions with empirical evidence, a commendable practice. Furthermore, the paper's overall clarity and effective communication make it accessible, allowing readers to readily grasp the core message.

**Weaknesses:**

I am content with the results presented in this paper. My primary concern pertains to the correctness of these results. Due to time constraints, I have not thoroughly reviewed the entire proof. I will try to check the proof recently, and I believe resolving the following issues will enhance my comprehension of this paper:

- It appears that Algorithm 1 closely resembles the OPPO+ algorithm introduced by [Zhong and Zhang, 2023], with the main difference being the policy switching update. I am curious whether this straightforward modification alone leads to the achievement of $\sqrt{T}$ regret, or if the authors have introduced novel proof techniques to support their claims.

- The algorithm design and proof techniques applied in this paper seem notably distinct from the concurrent work presented by [Sherman et al., 2023]. I am interested in understanding how these two works independently derive the $\sqrt{T}$ regret, and would appreciate the authors' insights on this matter.

- If Algorithm OPORS bears a strong resemblance to OPPO or OPPO+, it raises questions regarding its significant outperformance compared to OPPO+ empirically. In prior research, OPPO from [Cai et al., 2020] was found to be on par or slightly superior to LSVI-UCB. However, the absence of the provided source code prevents me from verifying whether the authors have properly tuned the hyperparameters of both algorithms to ensure a fair comparison. It would be valuable if the authors could briefly explain this phenomenon and, if feasible, include the source code in the supplementary materials for transparency.

**Questions:**

The main questions are provided in the weakness section. Another question is that whether this paper can handle adversarial linear MDPs with full-information feedback. If so, it would further strengthen the paper.

---

### Official Review · Reviewer_p4hn · 2023-10-31

**Soundness:** 2 fair
**Presentation:** 3 good
**Contribution:** 2 fair
**Rating:** 3
**Confidence:** 5

**Summary:**

This paper's main focused on the realm of policy optimization for learning linear Markov Decision Processes (MDPs). In contrast to prior frameworks centered around adversarial reward scenarios, the author introduces the OPORS algorithm, specifically designed to handle stochastic rewards. This novel algorithm attains a regret guarantee of $O(d^2H^2\sqrt{T})$. The author also extends these findings to encompass general function approximation within finite eluder dimension. Moreover, the empirical results solidly affirm the efficacy of the proposed algorithm. However, there exist a technique flaw in the proof of Lemma 3.4 and the main results may not correct.

**Strengths:**

1. The OPORS algorithm, as proposed, initially attains a regret guarantee of $O(\sqrt{T})$ for the task of learning linear Markov Decision Processes using policy optimization. In comparison to the concurrent work by Sherman et al. in 2023, OPORS distinguishes itself by not necessitating a reward-free exploration process for warm-up and also by offering a superior regret guarantee with respect to the factors of d and H.

2. The author further extends the results to encompass general function approximation within the constraints of finite eluder dimension, marking the introduction of the inaugural policy optimization algorithm tailored for general function approximation.

3. The empirical results provide strong confirmation of the effectiveness of the proposed algorithm.

**Weaknesses:**

1. In this study, the proposed OPORS algorithm is specifically designed to address scenarios with stochastic rewards. It's important to note that tackling adversarial rewards, as accomplished by previous policy-optimization methods, presents a more challenging task. Therefore, the contribution of the OPORS algorithm is limited. Furthermore, even in the context of stochastic rewards, a comparison with the optimal algorithm in He et al. 2023 indicates that the OPORS algorithm has a similar computational cost, albeit with the additional dependencies on d and H in the regret guarantee.

[1] Rate-optimal policy optimization or linear markov decision processes

[2] Nearly minimax optimal reinforcement learning for linear Markov decision processes.
the regret guarantee has extra d,H dependency

2. The proof of key Lemma 3.4 is incorrect. At the beginning of the page 16, the proof appears to have a technique flaw in the step where it uses the fact that $0 \leq k_{i+1} - k_i \leq K$ to establish $(k_{i+1} - k_i) \cdot \text{gap}(D_{KL}) \leq K \cdot \text{gap}(D_{KL}).$ This indeed assumes that $\text{gap}(D_{KL})$ is greater than zero, which has not been adequately proven in the context of the proof.

3. Regarding computational cost, it's essential to consider that Wang et al. 2021 [1] and He et al. 2023 [2] have also introduced the rare-switching policy technique, reducing the frequency of policy updates to $O(dH\log T)$. Consequently, it's reasonable to expect that these algorithms might exhibit a similar computational cost to the OPORS algorithm. Therefore, it is advisable to conduct a comprehensive computational complexity analysis and experiments, comparing these algorithms, before asserting a "significantly reduced computation cost."

[1] Provably efficient reinforcement learning with linear function approximation under adaptivity constraints.

[2] Nearly minimax optimal reinforcement learning for linear Markov decision processes.

4. In the context of policy optimization, it's worth noting that the policy is updated using the Exp3 method (as indicated in Line 10 of OPORS), and the optimistic value function is computed with respect to the policy (Line 11 in OPORS). While this approach may pose additional computational challenges compared to UCB-type algorithms, it raises questions about the computational complexity of OPORS when dealing with larger state-action spaces. To address these concerns, it would be beneficial if the author could provide a detailed computational analysis of the policy updating process or conduct simulations with larger state-action spaces to assess the algorithm's performance in such scenarios.

**Questions:**

1. Providing more insights into how to eliminate the additional $\log |A|$ factor in the regret guarantee as compared to Sherman et al. 2023 would indeed be valuable. This factor is attributed to policy optimization error (2), as indicated by the regret decomposition in Cai et al. 2020 [1] and Lemma 3.4. It is noteworthy that most policy optimization methods for linear function approximation exhibit a $\log |A|$ dependency. Addressing this issue could be of independent interest for researchers aiming to remove this factor.

[1] Provably efficient exploration in policy optimization


2. The computation cost analysis and experiments in the paper predominantly focus on the linear function approximation (OPORS algorithm). It is important for the author to explicitly state whether the same analysis applies to the general function approximation $F-OPORS$ algorithm in the introduction.

3. There is an error in the additional related works section concerning He et al. (2023) [1]. The reference should be corrected, as the work in question does not use the policy optimization method and instead combines Bernstein-type concentration inequality, rare-switching policy, and a monotonic variance estimator to achieve optimal regret in linear MDPs, which has already been discussed in the RL with linear function approximation section. The correct reference for adapting the Bernstein-type concentration inequality from Zhou et al. (2021) is He et al. (2022) [2].

[1] Nearly minimax optimal reinforcement learning for linear markov decision processes

[2] Near-optimal policy optimization algorithms for learning adversarial linear mixture mdps.

---

### Official Review · Reviewer_1GBb · 2023-11-08

**Soundness:** 3 good
**Presentation:** 3 good
**Contribution:** 3 good
**Rating:** 8
**Confidence:** 4

**Summary:**

The paper addresses the limited theoretical analysis of policy optimization algorithms in reinforcement learning compared to value-based algorithms. It introduces a new policy optimization algorithm that is provably efficient and incorporates optimistic value estimation and rare policy switches. For linear Markov decision processes (MDP), the algorithm achieves the sharpest regret bound among policy optimization algorithms. It is also extended to general function approximation and establishes the first regret guarantee for policy optimization with general function approximation. Numerical experiments confirm that the algorithm is competitive with existing RL algorithms and computationally efficient, supporting the theoretical claims.

**Strengths:**

1. The paper is fairly well organized and the results provided are impressive.

2. The theoretical guarantee is comprehensive (from linear setting to general setting) and solid. The paper also demonstrates the effectiveness of the algorithm in practice.

**Weaknesses:**

The idea of using rare policy switches and the related techniques are not very new.

**Questions:**

1. Is the rare policy switches technique the main reason that OPORS gives the regret bound of $O(\sqrt{K})$, instead of $O(K^{\frac{3}{4}})$ in [1]? In my view, the rare policy switches can effectively reduce the log covering number of the value function class, and [1] fails to get the optimal bound since they use the multi-batched updating, and the batch $l = O(\sqrt{K})$, which means there are still $O(\sqrt{K})$ different policies and this increases the covering number of policy class.

2. Can authors give some statistical intuition of improving the regret bound of [2]?

[1] Han Zhong and Tong Zhang. A theoretical analysis of optimistic proximal policy optimization in
linear markov decision processes. arXiv preprint arXiv:2305.08841, 2023.


[2] Uri Sherman, Alon Cohen, Tomer Koren, and Yishay Mansour. Rate-optimal policy optimization
for linear markov decision processes. arXiv preprint arXiv:2308.14642, 2023.